# DKP: Semantic Consistency Distillation via Key-layer Pre-alignment with Large Vision-language Models for Cross-Modal Retrieval

## Abstract

Recent retrieval solutions based on large vision-language models (VLMs) have shown promising performance by aligning vision and language representations for image-text retrieval (ITR). However, most methods rely solely on final-layer features, overlooking the rich semantic patterns embedded in intermediate layers. In this paper, we propose Semantic Consistency **D**istillation via **K**ey-layer **P**re-alignment (termed as DKP), a novel paradigm that enhances cross-modal retrieval by leveraging intermediate knowledge of VLMs. Specifically, we introduce (i) Key-layer Pre-alignment (KPA) to identify and align the most semantically meaningful intermediate features across modalities, and (ii) Semantic Consistency Distillation (SCD) to regularize cross-modal learning via intra-modal structure. Extensive experiments on Flickr30K and MS-COCO validate DKP significantly boosts retrieval performance while requiring over 60% fewer learnable parameters and significantly less computational cost, without introducing additional supervision or external knowledge. The anonymous code is available at: https://anonymous.4open.science/r/DKP.

## 1 Introduction

In recent years, large vision-language models (VLMs) such as CLIP Radford et al. (2021) have significantly advanced the field of cross-modal image-text retrieval (ITR) Wang et al. (2024); Xie et al. (2024); Zhang et al. (2025); Fu et al. (2024); Zhang et al. (2024), which is a fundamental task that aims to align visual and textual modalities within a shared semantic space. This task plays a pivotal role in a wide range of real-world applications, including cross-modal search Liu et al. (2023); Ziems et al. (2023), image captioning Dessì et al. (2023); Li et al. (2025a), and visual question answering Lee et al. (2025); Jiang et al. (2024); Cao et al. (2025). The core challenge of ITR lies in learning discriminative and modality-invariant representations that can capture fine-grained semantics and relational structures across images and natural language descriptions.

Among traditional ITR models, most approaches adopt independent visual and textual encoders, followed by a similarity computation operation to align cross-modal representations. For instance, SGRAF Diao et al. (2021) proposed a semantic graph reasoning framework leveraging Graph Convolutional Networks (GCNs) and attention filtering to suppress noisy alignments and enhance fine-grained matching. TVRN Pang et al. (2024) further improved alignment by enriching textual context and reducing visual ambiguity. Although effective, these approaches typically rely on task-specific supervision and hand-crafted alignment strategies, limiting their generalization, especially in zero-shot or low-resource scenarios. To overcome these limitations, recent advances have shifted toward large-scale vision-language pretraining paradigms. Notably, CLIP Radford et al. (2021) aligns vision and language representations via contrastive learning over hundreds of millions of image-text pairs, demonstrating remarkable generalization across diverse retrieval benchmarks. Despite their success, most existing CLIP-based methods focus on global-level alignment and depend heavily on large-scale data and computationally intensive backbones. In practical settings, it is often desirable to maintain high retrieval performance under limited supervision and restricted training resources. This tension between performance and efficiency highlights an important yet under-explored question: **how can we retain the high performance of VLMs while significantly reducing training**

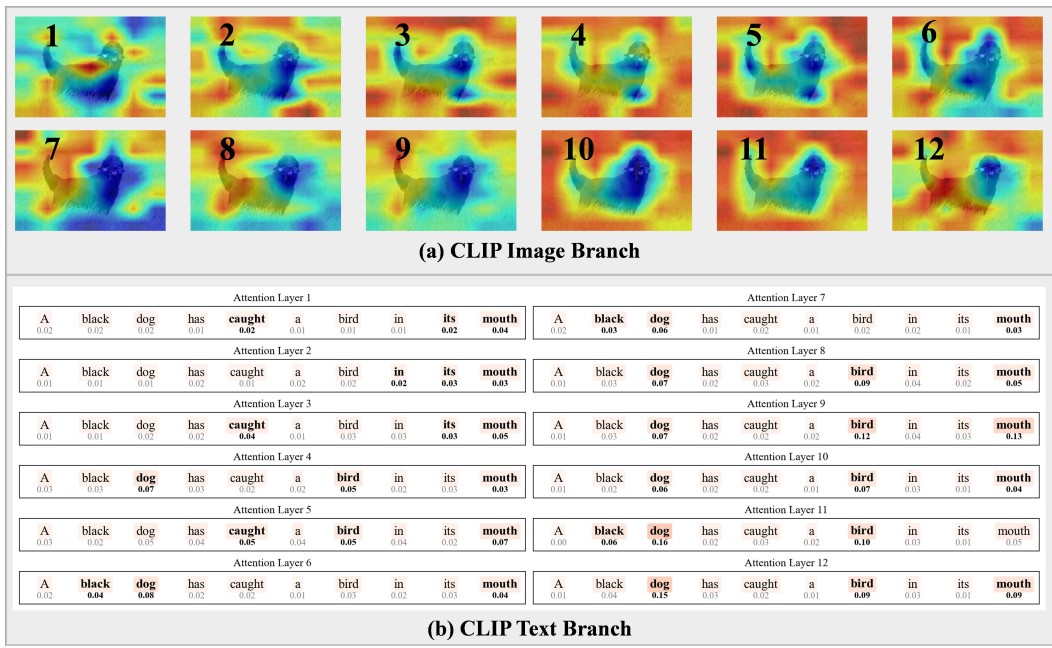

Figure 1: Attention maps from CLIP's image and text encoders reveal a semantic crystallization phenomenon beyond layer 8, where attention consistently focuses on salient regions and keywords.

**cost and parameter overhead?** To this end, gaining a deeper understanding of where and how vision-language alignment emerges within pretrained backbones like CLIP becomes crucial.

However, most existing CLIP-based ITR methods Huang et al. (2024); Zhao et al. (2024) exclusively align the final-layer representations from the vision and language encoders, overlooking the rich intermediate semantics that are progressively formed throughout the network. As a result, they tend to underexploit structural or localized cues captured in earlier layers, and the alignment becomes overly dependent on the final-stage features. Moreover, fine-tuning the entire backbone leads to substantial training costs in terms of parameters and computation time. This raises a natural and compelling question: **can we identify and leverage specific intermediate layers that are semantically expressive enough to complement or even substitute final-layer alignment, thereby reducing training overhead while preserving cross-modal semantic consistency?**

To explore this question, we conduct an empirical analysis of CLIP's hierarchical attention behavior across its transformer layers. Taking an image with its corresponding text caption "A black dog has caught a bird in its mouth" as an example, as illustrated in Figure 1(a), the image encoder (ViT-B/32) exhibits a clear progression in semantic focus: attention maps in early layers (Layers 1–7) are diffuse and unstable, whereas from Layer 8 onward, attention consistently converges on semantically salient regions, such as the dog's head and the bird in its mouth. This convergence pattern is mirrored in the text encoder, as shown in Figure 1(b), where attention gradually shifts from uniformly distributed weights to concentrated focus on key tokens like "dog", "bird", and "mouth". Notably, the attention weight for "dog" peaks at 0.16 in Layer 11, indicating a strong semantic crystallization beyond Layer 8.

These observations reveal a crucial yet under-explored phenomenon: although semantic focus emerges in deeper layers, the learned representations remain susceptible to **semantic drift**, where subtle yet meaningful intra-modal relationships are overlooked or diluted during alignment. This insight motivates our second line of investigation. While existing retrieval paradigms primarily emphasize cross-modal alignment, that is, matching visual and textual representations, they often ignore **intra-modal semantic structures**, such as the similarity relationships within images or within sentences themselves. Recent studies Wang et al. (2023) have shown that maintaining semantic consistency within modalities can provide auxiliary supervision to optimize cross-modal matching. Therefore, **how to preserve semantic consistency both across modalities and within individual modalities has become an important research focus in current cross-modal retrieval.** Rep-

resentative solutions like CUSA Huang et al. (2024) rely on additional external knowledge from monomodal large models to guide the model's internal knowledge. On the one hand, this approach requires external knowledge assistance, and on the other hand, such "fixed" external knowledge would introduce semantic biases, failing to adapt in real time to the semantic changes that occur during training.

To address the above challenges, we propose Semantic Consistency **D**istillation via **K**ey-layer **P**re-alignment with Large VLMs for cross-modal retrieval (termed as **DKP**). DKP reduces training parameters by over 60% compared to full parameter fine-tuning, effectively addressing the issue of semantic consistency within key-layers and intra-modalities. (1) We propose Key-layer Pre-alignment (KPA), a parameter-efficient dual-phase alignment strategy that optimizes cross-modal correspondence at the semantic-crystallizing Layer 8 while synergistically co-training the output layer, achieving >60% parameter reduction without compromising CLIP's representational capacity. (2) To preserve semantic consistency across and within modalities during alignment, we propose Semantic Consistency Distillation (SCD), a novel self-supervised method that combats semantic drift by leveraging intra-modal relationships inherent in the model to preserve cross-modal semantic consistency without external knowledge. The main contributions of this paper can be summarized as follows.

- **New perspective.** We conduct a comprehensive analysis of attention dynamics within CLIP, revealing a key convergence phenomenon where semantically meaningful representations consistently emerge at intermediate transformer layers.

- **Novel method.** We propose a dual-component architecture DKP, which introduces a *Key-layer Pre-alignment (KPA)* strategy to exploit semantically expressive intermediate layers and a *Semantic Consistency Distillation (SCD)* mechanism to regularize cross-modal alignment by leveraging intra-modal structure without requiring external knowledge or annotations.

- **High performance.** Extensive experiments on Flickr30K and MS-COCO demonstrate that our DKP achieves superior performance compared to existing state-of-the-art (SOTA) solutions, while greatly reducing the model complexity including training parameters and time cost.

## 2 RELATED WORKS

### 2.1 CROSS-MODAL IMAGE-TEXT RETRIEVAL

The core challenge of cross-modal image-text retrieval (ITR) lies in aligning the semantic information of heterogeneous modalities (e.g., image and text). Early research mainly employed non-pre-trained models. For instance, VSE++ Faghri et al. (2018) utilized a Convolutional Neural Network (CNN) to extract image features and align them with text embeddings. However, such methods struggled to capture high-level semantics due to their reliance on local pixel features. Subsequent work introduced object detectors (e.g., Faster R-CNN Ren et al. (2017)) to extract image region features and combined them with attention mechanisms (e.g., SCAN Lee et al. (2018)) to achieve fine-grained alignment. CHAN Pan et al. (2023) effectively mines the most informative region-word pairs by introducing a hard allocation coding scheme and rejects redundant or irrelevant alignments, thereby achieving significant improvements in both accuracy and efficiency. KGEMT Zheng et al. (2024) improves retrieval performance by combining coarse-grained and fine-grained features with a multi-modal knowledge graph. TSVC Lyu et al. (2025) further introduces a topology-aware self-adaptive consistency strategy that constructs intra-image topological graphs to enhance the robustness of cross-modal alignment. Due to the limited generalization ability and scalability of such traditional or non-pretrained methods, recent research has increasingly shifted towards leveraging large VLMs to address cross-modal retrieval tasks.

### 2.2 LARGE VISION-LANGUAGE MODELS

Large VLMs achieve unified cross-modal semantic embedding representations by pre-training on massive image-text paired data. Among them, CLIP Radford et al. (2021), as a representative of the

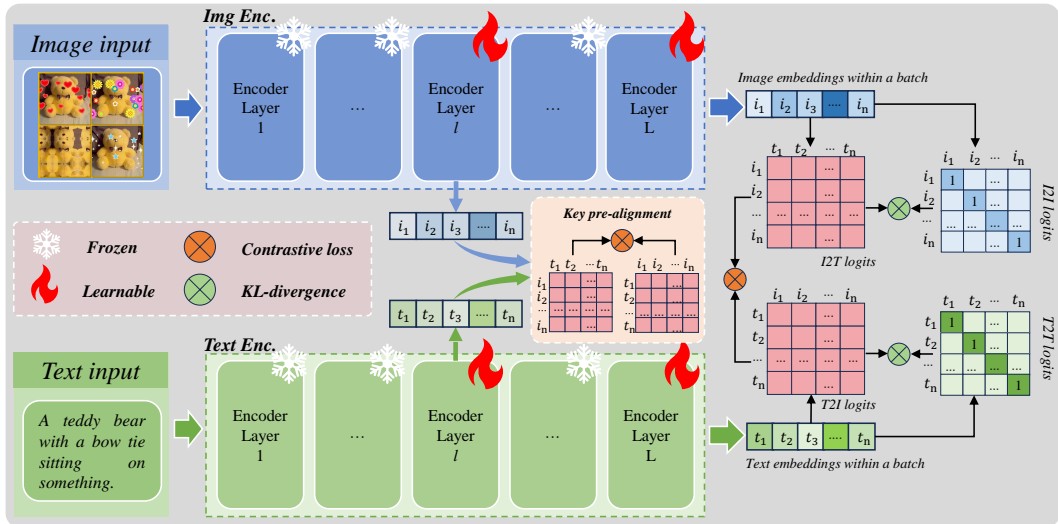

Figure 2: Overall architecture of DKP. We freeze lower layers while selectively fine-tuning top layers and introduce two complementary objectives: Key-layer Pre-Alignment (KPA) at the semantic crystallization layer and Semantic Consistency Distillation (SCD) across modalities.

dual-stream architecture, employs separate encoders for images and text and aligns modal features in a shared embedding space through contrastive learning. CLIP is trained on 400 million image-text pairs, and its core idea is to maximize the similarity of positive sample pairs while minimizing the similarity of negative sample pairs. PyramidCLIP Gao et al. (2022) alleviates the strict constraints during the pre-training phase by softening the loss of negative samples, thereby reducing the risk of the model being forced to distinguish between compatible negative sample pairs. FLIP Li et al. (2023) achieves a better trade-off between accuracy and training time by masking, enabling it to process more sample pairs within the same timeframe and compare a greater number of sample pairs in each iteration. SoftCLIP Gao et al. (2024b) enhances cross-modal interaction between vision and language by introducing soft targets based on fine-grained intra-modal self-similarity and decoupling negative samples within the distribution. Recently, MV-VSE++ Li et al. (2025b) explores multi-view soft alignment to enhance intra-modal and cross-modal representation learning, yet it still relies on full fine-tuning and does not explicitly address the hierarchical semantic emergence within VLMs. How to leverage the high performance of VLMs for ITR while reducing their complexity has become a current development trend.

## 3 PROPOSED METHODOLOGY

### 3.1 OVERVIEW ARCHITECTURE

Our solution (i.e., DKP) is plug-and-play and can be deployed on any VLM's architecture similar to CLIP for efficient and robust ITR. Taking CLIP as an example, given a batch of image-text pairs $\{(i_k, t_k)\}_{k=1}^N$, we first extract intermediate and final-layer features from the image encoder $E^I$ and the text encoder $E^T$. Instead of fully fine-tuning all layers, DKP identifies a semantically expressive intermediate layer, denoted as Layer $l$ (e.g., Layer 8), as the **key-layer**. We freeze the shallow and middle layers (marked with white snow in Figure 2) while updating only the final few transformer layers (marked with red fire), significantly reducing the number of trainable parameters. We then perform dual-stage alignment. (1) A contrastive loss is applied at the final-layer to ensure global cross-modal matching. (2) A pre-alignment loss is imposed at the key-layer to enhance intermediate semantic compatibility across modalities. Furthermore, we incorporate a Semantic Consistency Distillation objective that distills intra-modal relational structures (e.g., image-to-image and text-to-text similarities) into the cross-modal similarity.

## 3.2 KEY-LAYER PRE-ALIGNMENT

Traditional ITR methods rely exclusively on the final-layer outputs of VLMs. However, our empirical analysis in Section 1 reveals that rich localized and structural semantics emerge earlier around Layer $l$. To exploit this, we introduce a "Key-layer Pre-alignment" strategy that directly applies a contrastive loss at this intermediate layer. Let $Q_l^i, Q_l^t \in R^{N \times d}$ be the $l$-th layer features for image and text, respectively. We define the contrastive loss at this layer:

$$
\mathcal{L}_{i2t}^{(l)} = -\frac{1}{N} \sum_{j=1}^{N} \log \frac{\exp((Q_j^{i,l})^\top Q_j^{t,l}/\tau)}{\sum_{k=1}^{N} \exp((Q_j^{i,l})^\top Q_k^{t,l}/\tau)},
$$

$$
\mathcal{L}_{t2i}^{(l)} = -\frac{1}{N} \sum_{j=1}^{N} \log \frac{\exp((Q_j^{t,l})^\top Q_j^{i,l}/\tau)}{\sum_{k=1}^{N} \exp((Q_j^{t,l})^\top Q_k^{i,l}/\tau)}, \tag{1}
$$

$$
\mathcal{L}_{\text{KPA}} = \frac{1}{2}(\mathcal{L}_{i2t}^{(l)} + \mathcal{L}_{t2i}^{(l)}),
$$

where $\tau$ is a learnable temperature parameter. This loss complements the standard contrastive loss at the final layer (see Equation (4)), allowing for "semantic alignment at multiple depths".

## 3.3 SEMANTIC CONSISTENCY DISTILLATION

While contrastive learning excels at aligning paired image-text instances, it largely neglects latent semantic structures within each modality, such as relationships among visually similar images or semantically related captions. This often leads to an over-reliance on instance-level supervision and insufficient modeling of intra-modal regularities, which can impair robustness and generalization. To address this issue, we propose a Semantic Consistency Distillation (SCD) mechanism that regularizes cross-modal matching by preserving intra-modal semantic coherence through a self-distillation operation.

Specifically, we begin by computing intra-modal similarity matrices within a batch. Let $F^i \in R^{N \times d}$ and $F^t \in R^{N \times d}$ denote the normalized image and text features extracted from the final encoder layer. The intra-modal similarities $S^{\text{i2i}} \in R^{N \times N}$ and $S^{\text{t2t}} \in R^{N \times N}$ are calculated using cosine similarity:

$$
S_{jk}^{\text{i2i}} = \cos(F_j^i, F_k^i), \quad S_{jk}^{\text{t2t}} = \cos(F_j^t, F_k^t). \tag{2}
$$

These matrices are then row-wise normalized via softmax to obtain "probabilistic intra-modal semantic distributions", denoted by $F_j^{\text{i2i}} = \text{softmax}(S_j^{\text{i2i}})$ and $F_j^{\text{t2t}} = \text{softmax}(S_j^{\text{t2t}})$, which act as structure-aware guidance signals.

Meanwhile, we compute cross-modal similarity distributions using the same batch-wise contrastive embeddings. The image-to-text similarity $S^{\text{i2t}}$ and text-to-image similarity $S^{\text{t2i}}$ are similarly computed and softmax-normalized to yield predicted distributions $P_j^{\text{i2t}}$ and $P_j^{\text{t2i}}$, respectively.

To transfer intra-modal structure to cross-modal space, we minimize the KL-divergence between the intra-modal targets and the corresponding cross-modal predictions. The final SCD loss is defined as:

$$
\mathcal{L}_{\text{SCD}} = \frac{1}{2N} \sum_{j=1}^{N} (\text{KL}(F_j^{\text{i2i}} \parallel P_j^{\text{i2t}}) + \text{KL}(F_j^{\text{t2t}} \parallel P_j^{\text{t2i}})). \tag{3}
$$

By enforcing semantic consistency between intra-modal and cross-modal similarity structures, SCD effectively mitigates "semantic drift" during training. It encourages the model to learn "relationally aligned embeddings", where similar visual or textual inputs reinforce each other across modalities even in the absence of direct supervision. This serves as a form of structure-preserving regularization, enhancing both retrieval robustness and fine-grained matching fidelity.

## 3.4 TRAINING LOSS

The overall training objective integrates three components: (1) the original image-text contrastive loss at the final layer, denoted as $\mathcal{L}_{\text{Original}}$, (2) the intermediate-level alignment via our proposed

Key-layer Pre-alignment (KPA) loss $\mathcal{L}_{\text{KPA}}$, and (3) the Semantic Consistency Distillation (SCD) loss $\mathcal{L}_{\text{SCD}}$. Together, they enable multi-level and structure-aware alignment across and within modalities.

The original contrastive loss operates on the final-layer representations $F^i, F^t \in R^{N \times d}$ from the image and text encoders, respectively. It follows a symmetric InfoNCE formulation and is defined as:

$$\mathcal{L}_{i2t}^{(L)} = -\frac{1}{N} \sum_{j=1}^{N} \log \frac{\exp((F_j^i)^\top F_j^t / \tau)}{\sum_{k=1}^{N} \exp((F_j^i)^\top F_k^t / \tau)},$$

$$\mathcal{L}_{t2i}^{(L)} = -\frac{1}{N} \sum_{j=1}^{N} \log \frac{\exp((F_j^t)^\top F_j^i / \tau)}{\sum_{k=1}^{N} \exp((F_j^t)^\top F_k^i / \tau)}, \tag{4}$$

$$\mathcal{L}_{\text{Original}} = \frac{1}{2} \left( \mathcal{L}_{i2t}^{(L)} + \mathcal{L}_{t2i}^{(L)} \right),$$

where $\tau$ is a learnable temperature parameter, and $L$ denotes the final encoder layer. The final training objective combines all three components as follows:

$$\mathcal{L}_{\text{Total}} = \mathcal{L}_{\text{Original}} + \alpha \cdot \mathcal{L}_{\text{KPA}} + \beta \cdot \mathcal{L}_{\text{SCD}}, \tag{5}$$

where $\alpha$ and $\beta$ are learnable hyperparameters that balance the contributions of key-layer pre-alignment and semantic consistency distillation. By jointly optimizing $\mathcal{L}_{\text{Original}}$, $\mathcal{L}_{\text{KPA}}$, and $\mathcal{L}_{\text{SCD}}$, our approach benefits from both hierarchical depth-aware alignment and structure-aware regularization. This unified objective enables more accurate and robust cross-modal retrieval, especially under low-resource or fine-tuning-constrained scenarios.

## 4 Experiments

### 4.1 Experimental Settings

**Datasets and evaluation metrics.** To evaluate the effectiveness of our proposed method, we conduct experiments on two widely adopted ITR benchmarks: MS-COCO Lin et al. (2014) and Flickr30K Young et al. (2014). MS-COCO contains a total of 123,287 images, each paired with five human-annotated captions, resulting in 616,435 text descriptions. Following the standard split, we use 113,287 images for training and allocate two separate subsets of 5,000 images each for validation and testing, respectively. Flickr30K consists of 31,783 images, also annotated with five captions per image, leading to 158,915 textual descriptions. Out of these, 29,783 images are used for training, while the remaining 2,000 images are evenly divided into 1,000 each for validation and testing. For evaluation, we adopt the widely used Recall@K (R@K) metric, where K $\in \{1, 5, 10\}$. This metric measures the percentage of correct matches found within the top-K retrieved results. To provide an overall assessment of retrieval performance, we compute the RSUM score, which is the sum of all six Recall@K values (i.e., R@1, R@5, and R@10 for both image-to-text and text-to-image retrieval).

**Baselines.** To thoroughly evaluate the effectiveness of our proposed DKP, we conduct comprehensive comparisons with a wide range of SOTA ITR methods. These baselines include both non-CLIP-based approaches, such as VSE++ Faghri et al. (2018), SCAN Lee et al. (2018), VSE$\infty$ Chen et al. (2021), SGRAF Diao et al. (2021), NAAF Zhang et al. (2022), CHAN Pan et al. (2023), LAPS Fu et al. (2024), TVRN Pang et al. (2024), and KGEMT Zheng et al. (2024), DSRLN Wu et al. (2025), as well as CLIP-based solutions including the official CLIP Radford et al. (2021), RLCF Zhao et al. (2024), and CUSA Huang et al. (2024). These methods cover a diverse set of backbones and design paradigms, ranging from region-based feature fusion to global contrastive learning.

**Implementation details.** All experiments are implemented in PyTorch using $8 \times 24$GB NVIDIA RTX 3090 GPUs. We employ the Adam optimizer with an initial learning rate of 8e-6 and a weight decay of 1e-5 for both Flickr30K and MS-COCO datasets. A cosine learning rate scheduler is used with a minimum learning rate of 1e-6 and a decay rate of 1, without warm-up or cooldown phases. The total training epochs are set to 30 on Flickr30K and 20 on MS-COCO. We adopt different batch sizes depending on the backbone: 128 for ViT-B/32, and 64 for ViT-B/16. More details of this project have been released at: https://anonymous.4open.science/r/DKP.

Table 1: Performance (%) comparisons on Flickr30K and MS-COCO. Note that † denotes CLIP without fine-tuning and ‡ denotes CLIP with full-parameter fine-tuning.

| Methods | Backbone | Learnable parameters (M) | Flickr30K (1K test set) | | | | | | | MS-COCO (5K test set) | | | | | | |
| | | | I2T | | | T2I | | | RSUM | I2T | | | T2I | | | RSUM |
| | | | R@1 | R@5 | R@10 | R@1 | R@5 | R@10 | | R@1 | R@5 | R@10 | R@1 | R@5 | R@10 | |
| *Non-CLIP-based methods* | | | | | | | | | | | | | | | | |
| *VSE++* | *ResNet-152, BiGRU* | 15.8 | 71.8 | 92.8 | 96.5 | 59.4 | 84.7 | 90.9 | 496.1 | 54.9 | 82.8 | 90.4 | 42.4 | 72.4 | 82.8 | 425.7 |
| *SCAN* | *Faster R-CNN, BiGRU* | 13.7 | 67.4 | 90.3 | 95.8 | 48.6 | 77.7 | 85.2 | 465.0 | 50.4 | 82.2 | 90.0 | 38.6 | 69.3 | 80.4 | 410.9 |
| *VSE∞* | *Faster R-CNN, BERT* | 113.9 | 81.7 | 95.4 | 97.6 | 61.4 | 85.9 | 91.5 | 513.5 | 58.3 | 85.3 | 92.3 | 42.4 | 72.7 | 83.2 | 434.2 |
| *SGRAF* | *Faster R-CNN, BiGRU* | 18.1 | 78.4 | 94.6 | 97.5 | 58.2 | 83.0 | 89.1 | 500.8 | 58.8 | 84.8 | 92.1 | 41.6 | 70.9 | 81.5 | 429.7 |
| *NAAF* | *Faster R-CNN, BiGRU* | 13.6 | 81.9 | 96.1 | 98.3 | 61.0 | 85.3 | 90.6 | 513.2 | 58.9 | 85.2 | 92 | 42.5 | 70.9 | 81.4 | 430.9 |
| *CHAN* | *Faster R-CNN, BiGRU* | 13.8 | 79.7 | 96.7 | 98.7 | 63.8 | 90.4 | 95.8 | 525.1 | 60.2 | 85.9 | 92.4 | 41.7 | 71.5 | 81.7 | 433.4 |
| *LAPS* | *ViT-Base-224, BERT* | 196.7 | 74.0 | 93.4 | 97.4 | 62.5 | 87.3 | 92.7 | 507.3 | 57.5 | 84.0 | 90.8 | 44.5 | 74.0 | 83.6 | 434.4 |
| *TVRN* | *Faster R-CNN, BiGRU* | 37.3 | 79.8 | 95.4 | 97.6 | 62.3 | 85.3 | 90.6 | 511.0 | 59.2 | 84.6 | 91.6 | 42.5 | 71.8 | 82.1 | 431.8 |
| *KGEMT* | *Faster R-CNN, BERT* | >100 | 79.8 | 97.1 | 99.0 | 66.7 | 92.0 | 97.3 | 531.9 | 59.6 | 86.0 | 93.1 | 45.1 | 75.1 | 85.0 | 443.9 |
| *DSRLN* | *Faster R-CNN, BiGRU* | >25 | 82.7 | 96.9 | 98.7 | 63.7 | 88.9 | 93.3 | 524.2 | 60.8 | 86.9 | 93.3 | 43.9 | 73.6 | 84.0 | 442.5 |
| *CLIP-based methods* | | | | | | | | | | | | | | | | |
| *CLIP†* | *CLIP-ViT (ViT-B/32)* | 0 | 77.8 | 94.5 | 98.3 | 59.08 | 83.54 | 90.10 | 503.32 | 50.04 | 74.70 | 83.02 | 30.18 | 55.63 | 66.69 | 360.26 |
| | *CLIP-ViT (ViT-B/16)* | 0 | 81.5 | 96.2 | 98.4 | 62.10 | 85.66 | 91.76 | 515.62 | 52.20 | 76.48 | 84.38 | 32.99 | 58.16 | 68.83 | 373.04 |
| *CLIP‡* | *CLIP-ViT (ViT-B/32)* | 151.80 | 78.7 | 95.4 | 98.0 | 66.30 | 88.60 | 93.10 | 520.10 | 56.30 | 81.70 | 89.40 | 42.80 | 71.20 | 81.10 | 422.50 |
| | *CLIP-ViT (ViT-B/16)* | 150.15 | 86.3 | 97.5 | 98.9 | 72.50 | 92.10 | 95.50 | 542.88 | 62.56 | 86.16 | 92.16 | 47.60 | 75.53 | 84.53 | 448.50 |
| *RLCF* | *CLIP-ViT (ViT-B/16)* | 86.19 | 68.5 | 90.2 | 93.7 | 88.10 | 97.70 | 98.90 | 537.10 | 38.40 | 63.50 | 72.60 | 60.80 | 80.50 | 87.60 | 403.40 |
| *CUSA* | *CLIP-ViT (ViT-B/32)* | 152.30 | 82.1 | 95.3 | 97.9 | 67.50 | 89.60 | 93.90 | 526.30 | 57.30 | 83.10 | 90.30 | 44.20 | 72.70 | 82.10 | 429.70 |
| | *CLIP-ViT(ViT-B/16)* | 150.68 | 88.5 | 97.8 | 99.1 | 75.38 | 92.98 | 96.26 | 550.02 | 63.22 | 85.96 | 92.44 | 48.78 | 76.48 | 85.11 | 451.99 |
| *Ours* | | | | | | | | | | | | | | | | |
| *DKP* | *CLIP-ViT (ViT-B/32)* | 21.01 | 81.2 | 96.3 | 98.9 | 68.18 | 90.76 | 94.86 | 530.20 | 58.28 | 83.54 | 90.24 | 44.12 | 72.17 | 82.46 | 430.81 |
| *DKP* | *CLIP-ViT (ViT-B/16)* | 21.01 | 89.4 | 98.6 | 99.4 | 77.22 | 94.20 | 97.06 | 555.88 | 63.74 | 87.12 | 92.86 | 48.90 | 76.66 | 85.11 | 454.39 |

Table 2: Cross-dataset retrieval performance between Flickr30K and MS-COCO. Models are trained on one dataset and tested on the other.

| Training Dataset | Test Dataset | Method | Image-to-Text | | | Text-to-Image | | | RSUM |
| | | | R@1 | R@5 | R@10 | R@1 | R@5 | R@10 | |
| Flickr30K | MS-COCO | *CLIP‡* | 35.16 | 59.60 | 70.54 | 25.03 | 48.67 | 59.84 | 298.84 |
| Flickr30K | MS-COCO | *CUSA* | 37.28 | 63.20 | 72.98 | 26.57 | 50.29 | 61.35 | 311.67 |
| Flickr30K | MS-COCO | *LAPS* | 27.10 | 50.50 | 62.00 | 19.00 | 40.30 | 51.50 | 250.40 |
| Flickr30K | MS-COCO | *DKP* | 41.92 | 67.86 | 78.06 | 31.53 | 57.15 | 68.31 | 344.83 |
| MS-COCO | Flickr30K | *CLIP‡* | 72.10 | 90.80 | 95.90 | 57.52 | 83.28 | 89.76 | 489.36 |
| MS-COCO | Flickr30K | *CUSA* | 74.30 | 91.70 | 95.60 | 60.12 | 85.14 | 90.66 | 497.52 |
| MS-COCO | Flickr30K | *LAPS* | 63.70 | 89.60 | 94.90 | 54.70 | 80.90 | 88.50 | 472.30 |
| MS-COCO | Flickr30K | *DKP* | 76.70 | 93.80 | 97.50 | 65.82 | 88.70 | 93.46 | 515.98 |

## 4.2 PERFORMANCE COMPARISONS

We report the performance of our proposed DKP on two widely-used ITR benchmarks, Flickr30K and MS-COCO, and compare it with a range of both traditional and CLIP-based methods (see Table 1). On Flickr30K, DKP consistently outperforms most existing baselines across all Recall@K metrics for both image-to-text (I2T) and text-to-image (T2I) retrieval. Notably, with the ViT-B/16 backbone, DKP achieves 89.4% R@1 for I2T and 77.22% R@1 for T2I, yielding a RSUM of 555.88, which surpasses the full-parameter fine-tuning CLIP‡ and CUSA by a notable margin. On MS-COCO, our approach maintains competitive performance, achieving 63.74% and 48.9% in R@1 for I2T and T2I respectively, and the overall highest RSUM of 454.39, demonstrating the robustness of our method across datasets and backbones. These results validate the effectiveness of DKP in improving fine-grained alignment without relying on full backbone tuning, and confirm that the combination of key-layer pre-alignment and semantic consistency distillation significantly enhances retrieval accuracy across different architecture variants. In Appendix A.9, our DKP consistently presents parameter efficiency and performance advantages on larger CLIP (ViT-L/14@336px) architecture.

## 4.3 CROSS-DATASET EVALUATION

We further conducted cross-dataset evaluations to analyze the model's generalization capability with ViT-B/32 architecture. First, we evaluated the performance of the model when it was trained on

Table 3: Training and inference cost across different architectures. DKP reduces both per-epoch training time and inference latency compared to full-parameter fine-tuning.

| | ViT-B/32 | | | | |
|---|---|---|---|---|---|
| Methods | Training | | Inference | | |
| | Time (minute) | Memory (MB) | Time (second) | Memory (MB) | FLOPs (GB) |
| *CLIP‡* | 4.96/epoch | 13932.07 | 6 | 2631.29 | |
| *CUSA* | 6.32/epoch | 13963.72 | 7 | 2635.30 | 4.89 |
| ***DKP*** | **3.97/epoch** | **6640.93** | **4** | **1296.40** | |
| | ViT-B/16 | | | | |
| Methods | Training | | Inference | | |
| | Time (minute) | Memory (MB) | Time (second) | Memory (MB) | FLOPs (GB) |
| *CLIP‡* | 12.28/epoch | 15802.81 | 8 | 3262.40 | |
| *CUSA* | 12.73/epoch | 15824.12 | 8 | 3266.43 | 13.21 |
| ***DKP*** | **11.07/epoch** | **8239.49** | **7** | **1934.01** | |

Flickr30K and tested on MS-COCO, as well as the opposite scenario. The results are shown in Table 2. When the model was trained on Flickr30K and tested on MS-COCO, DKP achieved an RSUM score of 344.83, which was significantly better than that of CLIP‡ (298.84) and CUSA (311.67). Similarly, when the model was trained on MS-COCO and tested on Flickr30K, DKP obtained an RSUM score of 515.98, while CLIP‡ and CUSA scored 489.36 and 497.52, respectively. In addition, we found that VLMs based approaches generally outperform methods based on strong dual-stream independent encoders (i.e., LAPS). This reflects that pre-trained cross-modal knowledge also plays a significant role in cross-domain retrieval tasks. These results indicate that DKP possesses excellent generalization capabilities when used for cross-dataset retrieval across different datasets. This is attributed to its key-layer pre-alignment mechanism and semantic-consistency distillation techniques, which contribute to the creation of more robust and cross-modal representations.

## 4.4 MODEL COMPLEXITY ANALYSIS

As reported in Table 1, DKP requires only **21.0M** learnable parameters for both ViT-B/32 and ViT-B/16. This is a $\geq$ **60%** reduction relative to full-parameter fine-tuning CLIP‡ (151.8M for ViT-B/32, 150.2M for ViT-B/16) and an even larger saving compared with CUSA (152.3M / 150.7M). Despite this lightweight footprint, DKP still yields the highest RSUM with each backbone, confirming that key-layer pre-alignment delivers strong retrieval accuracy without the cost of exhaustive fine-tuning.

To comprehensively evaluate the efficiency of our proposed method, we compare DKP with full-parameter fine-tuning (CLIP‡) and the state-of-the-art CUSA method in terms of training time, memory consumption, and inference overhead. The results for ViT-B/32 and ViT-B/16 backbones are summarized in Table 3. For ViT-B/32, DKP requires only 3.97 minutes per training epoch, which is **20%** faster than CLIP‡ (4.96 minutes) and **37%** faster than CUSA (6.32 minutes). More notably, DKP reduces training memory usage by over **52%**, consuming only 6640.93 MB compared to 13932.07 MB for CLIP‡ and 13963.72 MB for CUSA. During inference, DKP achieves a latency of 4 seconds, outperforming both CLIP‡ (6 seconds) and CUSA (7 seconds), while using less than half the inference memory (1296.40 MB vs. 2631.29 MB). Similar advantages are observed with ViT-B/16. DKP completes each training epoch in 11.07 minutes, compared to 12.28 minutes for CLIP‡ and 12.73 minutes for CUSA. Training memory is reduced by **48%** (8239.49 MB vs. 15802.81 MB), and inference memory is cut by **41%** (1934.01 MB vs. 3262.40 MB). Although FLOPs remain identical across methods for each backbone (4.89 GB for ViT-B/32 and 13.21 GB for ViT-B/16) due to the shared architecture, DKP's parameter-efficient design enables faster optimization and lower resource utilization without compromising computational throughput.

## 4.5 ABLATION STUDIES

**Influence of key components.** To investigate the individual contribution of each proposed component, we conduct a series of ablation experiments on Flickr30K with ViT-B/32. As shown in Table 4, the vanilla contrastive loss at the output layer yields an RSUM of 520.10. Adding our key-layer pre-alignment term ($\mathcal{L}_{\text{KPA}}$) lifts RSUM by +7.22, confirming the benefit of aligning features at the semantically rich Layer 8. Introducing semantic-consistency distillation ($\mathcal{L}_{\text{SCD}}$) alone

Table 4: Ablation study on different loss components. Adding KPA or SCD individually boosts RSUM relative to the baseline contrastive loss, while combining all three terms yields the best performance.

| $\mathcal{L}_{\text{Original}}$ | $\mathcal{L}_{\text{KPA}}$ | $\mathcal{L}_{\text{SCD}}$ | Image-to-Text | | | Text-to-Image | | | RSUM |
|---|---|---|---|---|---|---|---|---|---|
| | | | R@1 | R@5 | R@10 | R@1 | R@5 | R@10 | |
| ✓ | ✗ | ✗ | 78.7 | 95.4 | 98.0 | 66.30 | 88.60 | 93.10 | 520.10 |
| ✓ | ✓ | ✗ | 79.7 | 96.3 | 98.1 | 67.74 | 90.58 | 94.90 | 527.32 |
| ✓ | ✗ | ✓ | 81.7 | 95.6 | 98.2 | 68.40 | 90.64 | 95.20 | 529.74 |
| ✓ | ✓ | ✓ | 81.2 | 96.3 | 98.9 | 68.18 | 90.76 | 94.86 | **530.22** |

improves RSUM by +9.64, evidencing the value of preserving intra-modal structure. Combining all three objectives delivers the best overall score (530.22), indicating that hierarchical alignment and structure-aware regularisation are complementary and jointly responsible for DKP's performance gains.

**Influence of key layer.** To explore the semantic emergence process within CLIP, we investigate how selecting different layers as the key-layer impacts retrieval performance. As shown in Figure 3, we report the retrieval scores on MS-COCO using ViT-B/32 and ViT-B/16 backbones, where the x-axis denotes the transformer layer index used for Key-layer Pre-alignment. We observe a clear turning point at Layer 8: performance in earlier layers (Layers 1–7) is unstable and significantly lower, suggesting that semantic representations in these stages are still immature. Starting from Layer 8, retrieval accuracy becomes more consistent and steadily improves, al-

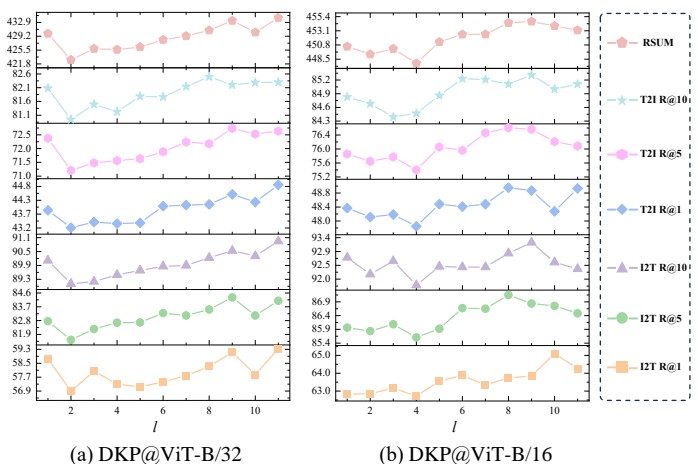

(a) DKP@ViT-B/32     (b) DKP@ViT-B/16

Figure 3: Influence of key-layer pre-alignment. Retrieval accuracy is reported when choosing different transformer layers for alignment on MS-COCO with ViT-B/32 and ViT-B/16.

ways yielding the highest performance when performing pre-alignment at one layer between 8-11. This observation validates our empirical finding that Layer 8 marks the onset of semantic crystallization across both modalities, serving as an effective and efficient alignment point. Our design choice to perform pre-alignment at this layer is thus empirically grounded and essential for preserving semantic fidelity while reducing parameter overhead.

**Sensitivity analysis of $\alpha$ and $\beta$.**
To evaluate the impact of the weighting coefficients $\alpha$ and $\beta$ in the overall loss function $\mathcal{L}_{\text{Total}}$ (see Equation (5)), we conduct a grid search on fixed values of $\alpha, \beta \in \{0.2, 0.4, 0.6, 0.8\}$ as well as a version where both weights are set to 1.0. As shown in Table 5, we observe that moderately increasing $\alpha$ (i.e., increasing the contribution of key-layer align-

Table 5: Sensitivity analysis of $\alpha$ and $\beta$. Moderate values such as $\alpha = 0.6, \beta = 0.4$ improve RSUM, while extreme settings can harm performance.

| $\alpha$ | $\beta$ | Image-to-Text | | | Text-to-Image | | | RSUM |
|---|---|---|---|---|---|---|---|---|
| | | R@1 | R@5 | R@10 | R@1 | R@5 | R@10 | |
| 0.2 | 0.8 | 81.2 | 95.7 | 98.0 | 68.48 | 90.46 | 95.01 | 528.94 |
| 0.4 | 0.6 | 81.2 | 95.7 | 98.2 | 68.38 | 90.38 | 94.94 | 528.80 |
| 0.6 | 0.4 | 82.2 | 95.8 | 98.4 | 68.28 | 90.46 | 94.66 | 529.80 |
| 0.8 | 0.2 | 81.5 | 96.0 | 98.4 | 68.50 | 90.38 | 94.58 | 529.36 |
| 1.0 | 1.0 | 81.1 | 95.7 | 98.4 | 67.82 | 90.10 | 94.78 | 527.90 |
| **Learnable** | | 81.2 | 96.3 | 98.9 | 68.18 | 90.76 | 94.86 | **530.22** |

ment) generally leads to better performance, with the setting $\alpha = 0.6, \beta = 0.4$ achieving a strong RSUM of 529.80. However, overly favoring one term, such as $\beta = 0.8$ or both $\alpha = \beta = 1.0$, can lead to slight performance drops, possibly due to over-regularization or loss imbalance. Notably, when we allow $\alpha$ and $\beta$ to be learnable during training, our model achieves the best performance across all metrics, with an RSUM of 530.22. This result highlights the importance of dynamically

balancing intermediate alignment and intra-modal semantic consistency, and suggests that adaptive weight learning can effectively stabilize training and maximize performance.

## 4.6 VISUALIZATION

**Intra-modal cohesion.** We visualize the intra-modal geometry of vanilla CLIP and our DKP in Figure 4. We randomly sample 1,000 image–text pairs from MS-COCO validation split and extract ViT-B/16 features for each modality. Cluster labels are obtained from DKP's features (right) and color mirrored back to CLIP (left), so each dot carries the same hue across models. Under DKP (b,d), the colored islands are markedly tighter and more separated (e.g., the cyan and magenta groups), whereas the same colors are dispersed in CLIP (a,c). This indicates that samples with similar semantics are pulled closer together only when Semantic Consistency Distillation (SCD) is applied. The visual trend matches the quantitative RSUM gains reported in Table 4 and highlights SCD's role in reducing intra-modal noise, which ultimately benefits cross-modal retrieval.

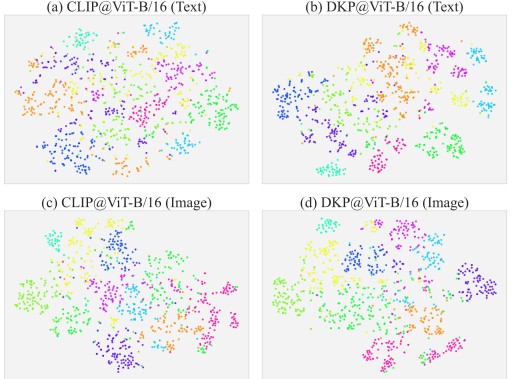

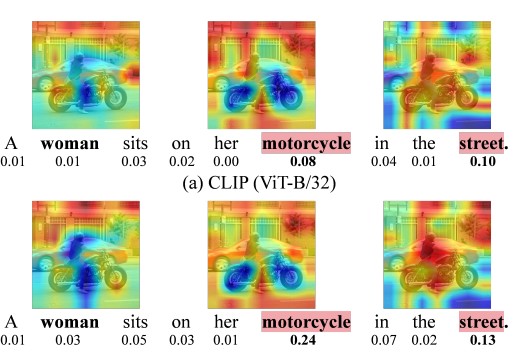

Figure 4: Qualitative comparison of intra-modal geometry for CLIP (left) and our DKP (right). DKP embeddings form denser and more coherent clusters in both image and text spaces, indicating stronger intra-modal cohesion.

Figure 5: Qualitative comparison of token–image attention for CLIP (top) and our DKP (bottom). DKP assigns higher weights to discriminative tokens such as "motorcycle" and produces sharper heatmaps focused on relevant image regions.

**Token-image attention.** We visualize the token-image attention in Figure 5. Taking an image with its caption "A woman sits on her motorcycle in the street." as an example, standard CLIP assigns only modest weight to the key object words (*motorcycle* (0.08) and *street* (0.10)) and the heat maps remain diffuse, attending to background as often as to the target vehicle. In contrast, our DKP amplifies the same tokens to 0.24 and 0.13, and its image maps concentrate on the bike's chassis and the road surface, producing a crisper, more semantically aligned focus on both the primary subject (*motorcycle*) and its context (*street*). The gains on verbs (*sits*: 0.05 vs 0.03) and agent (*woman*: 0.03 vs 0.01) further indicate that DKP preserves relational cues, yielding a richer cross-modal correspondence than vanilla CLIP while using far fewer trainable parameters.

## 5 CONCLUSION AND FUTURE WORKS

In this paper, we propose a lightweight yet effective paradigm named DKP for ITR, which tackles the challenges of parameter inefficiency and semantic misalignment in CLIP-based VLMs. By identifying a semantically crystallized key layer through empirical attention analysis, we introduce a KPA strategy that substantially reduces training overhead while preserving cross-modal alignment quality. Furthermore, to mitigate semantic drift and strengthen intra-modal coherence, we design a self-supervised SCD mechanism that leverages the model's inherent relational structure without relying on external knowledge. Extensive experiments on Flickr30K and MS-COCO demonstrate that DKP achieves competitive or superior performance compared to SOTA methods, while requiring over 60% fewer learnable parameters and significantly less computational cost. In the future, we plan to explore an input-adaptive dynamic key-layer selection mechanism, combined with parameter-efficient strategies such as prompt learning.

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

# A APPENDIX

This appendix is organized as follows:

- **Section A.1 presents the training procedure of DKP.**
- **Section A.2 analyzes the dynamic evolution of $\alpha$ and $\beta$.**
- **Section A.3 explores cross-modal alignment on more key-layers.**
- **Section A.4 provides the visualization results of image-text retrieval.**
- **Section A.5 illustrates more visualization results of token-image attention.**
- **Section A.6 illustrates more visualization results of intra-modal cohesion.**
- **Section A.7 explores pure prompt learning for image-text retrieval.**
- **Section A.8 discusses the architecture design with representative parameter-efficient fine-tuning (PEFT) methods.**
- **Section A.9 conducts layer-wise attention analysis and performance comparisons on asymmetric backbone with CLIP (ViT-L/14@336px).**
- **Section A.10 explores strong robustness on diverse datasets.**
- **Section A.11 conducts extensive cross-domain retrieval evaluations.**
- **Section A.12 analyzes the performance change with data pattern.**
- **Section A.13 investigates additional experiments analyzing attention patterns across both simple and complex scenarios.**
- **Section A.14 extends our layer-wise attention analysis to BLIP and ALBEF.**
- **Section A.15 explores the sensitivity analysis of temperature parameter on SCD.**
- **Section A.16 shows the spread of deviations from the mean.**
- **Section A.17 talks about the limitations of DKP and future research directions.**

## A.1 TRAINING PROCEDURE OF DKP

To provide a clearer overview of the training process of our proposed DKP, we present the detailed iterative training procedure in Algorithm 1. This algorithm outlines the key steps including forward propagation through the encoders, computation of the three loss components (original contrastive loss, key-layer pre-alignment loss, and semantic consistency distillation loss), and the backward update of trainable parameters. We highlight the efficiency and simplicity of DKP, which requires only a fraction of the parameters compared to full fine-tuning while achieving superior retrieval performance.

## A.2 DYNAMIC EVOLUTION OF $\alpha$ AND $\beta$

Figure 6 and Figure 7 track the two learnable coefficients that weight Key-layer Pre-Alignment ($\alpha$) and Semantic Consistency Distillation ($\beta$) during training. All runs start from an uninformed value of 0.5. On the smaller Flickr30K (30 epochs), both parameters decay slowly and almost in lock-step: from 0.50 to roughly 0.44 for ViT-B/32 and to 0.37 for the deeper ViT-B/16. This gentle decline suggests that KPA and SCD remain influential throughout, acting as sustained regularisers in a data-scarce regime.

In contrast, on the larger MS-COCO (20 epochs) the drop is steeper. For ViT-B/32, the pair stabilises near 0.31, while for ViT-B/16, they fall below 0.12 within the same wall-time budget. The model therefore learns to lean heavily on the original contrastive loss once the cross-modal signal becomes reliable, relegating the auxiliary terms to a refining role. Interestingly, $\alpha$ remains marginally higher than $\beta$ on all settings, hinting that key-layer alignment is slightly more critical than intra-modal distillation, yet the gap never exceeds 0.02–0.03—evidence that the two objectives are largely complementary.

Overall, the monotonic yet data-dependent decay pattern corroborates our design intuition: KPA and SCD are most valuable during the early, noisy stages of optimization, after which their influence is

---

**Algorithm 1** Training Procedure of DKP

---

**Require:** Batch of image-text pairs $(i_k, t_k)_{k=1}^{N}$; Image encoder $E^I$, text encoder $E^T$; Key layer index $l$ (e.g., $l = 8$); Hyperparameters $\alpha, \beta$ (learnable or fixed).
**Ensure:** Trained model parameters $\theta_{\text{trainable}}$.
 1: Freeze all layers except the last few transformer layers (red blocks in Figure 2).
 2: **for** each training iteration **do**
 3:     Extract intermediate features $Q_l^I, Q_l^T$ from layer $l$ of $E^I$ and $E^T$.
 4:     Extract final features $F^I, F^T$ from output of $E^I$ and $E^T$.
 5:     Compute $\mathcal{L}_{\text{Original}}$ using Equation (4) with $F^I, F^T$.
 6:     Compute $\mathcal{L}_{\text{KPA}}$ using Equation (1) with $Q_l^I, Q_l^T$.
 7:     Compute intra-modal similarities $S^{i2i}, S^{t2t}$ via Equation (2).
 8:     Compute cross-modal similarities $S^{i2t}, S^{t2i}$.
 9:     Normalize via softmax to get $F^{i2i}, F^{t2t}, P^{i2t}, P^{t2i}$.
 10:    Compute $\mathcal{L}_{\text{SCD}}$ via Equation (3).
 11:    $\mathcal{L}_{\text{Total}} \leftarrow \mathcal{L}_{\text{Original}} + \alpha \cdot \mathcal{L}_{\text{KPA}} + \beta \cdot \mathcal{L}_{\text{SCD}}$
 12:    Update $\theta_{\text{trainable}}$ via backpropagation with $\mathcal{L}_{\text{Total}}$.
 13: **end for**=0

---

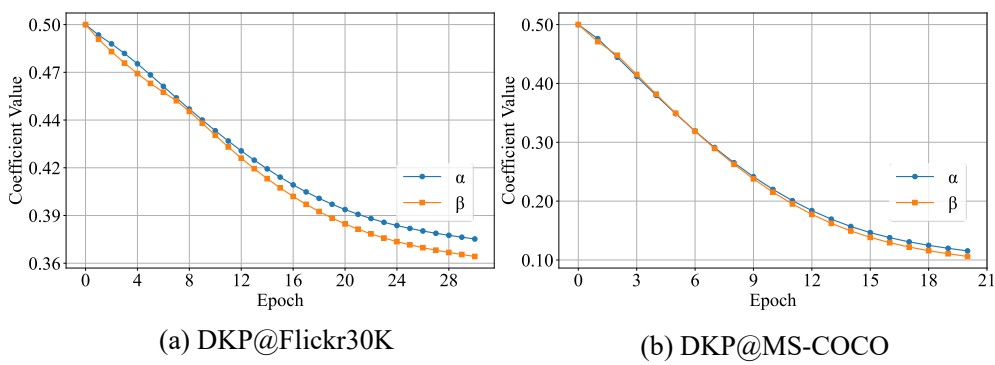

(a) DKP@Flickr30K      (b) DKP@MS-COCO

Figure 6: Dynamic evolution of loss weights $\alpha$ and $\beta$ for DKP on Flickr30K and MS-COCO using ViT-B/16. Both coefficients start from 0.5 and gradually decay, showing how the model adaptively balances KPA and SCD contributions throughout training.

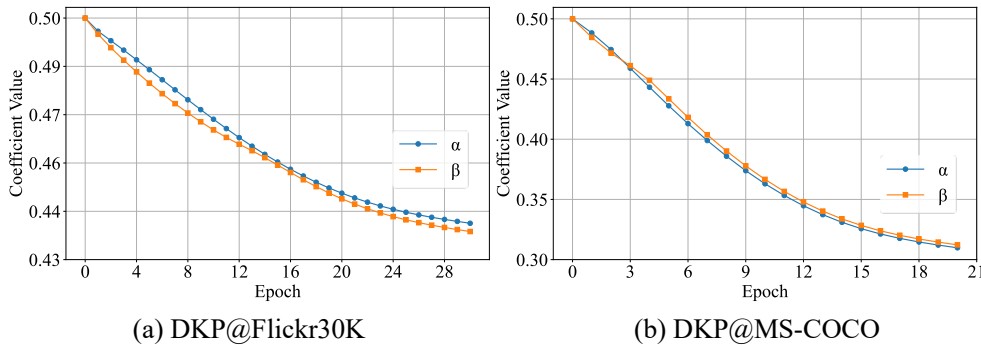

(a) DKP@Flickr30K      (b) DKP@MS-COCO

Figure 7: Dynamic evolution of $\alpha$ and $\beta$ for DKP on Flickr30K and MS-COCO using ViT-B/32. The faster decay compared to ViT-B/16 suggests that larger models increasingly rely on the original contrastive loss as alignment matures.

adaptively reduced, allowing the model to converge to a stable, high-precision alignment without manual re-tuning of loss weights.

Table 6: Performance change on more key-layer alignment. Relative increase columns explicitly show how multi-layer strategies inflate parameters and training time, highlighting the practical advantages of our proposed single-layer DKP alignment.

| Key-layer | Image-to-Text | | | Text-to-Image | | | RSUM | Learnable parameters (M) | Training time (minute) | Relative increase | |
|---|---|---|---|---|---|---|---|---|---|---|---|
| | R@1 | R@5 | R@10 | R@1 | R@5 | R@10 | | | | Params (%) | Time (%) |
| *ViT-B/32* | | | | | | | | | | | |
| 8,9 | 79.8 | 95.5 | 97.3 | 67.38 | 89.20 | 94.22 | 523.40 | 31.25 | 5.92/epoch | +48.7% | +49.1% |
| 8,10 | 80.7 | 96.2 | 98.6 | 67.24 | 89.68 | 94.44 | 526.86 | | | | |
| 8,9,10 | 79.2 | 95.5 | 97.9 | 66.32 | 89.16 | 93.90 | 521.98 | 41.49 | 7.91/epoch | +97.5% | +99.2% |
| 8 | 81.2 | 96.3 | 98.9 | 68.18 | 90.76 | 94.86 | 530.20 | 21.01 | 3.97/epoch | - | - |
| *ViT-B/16* | | | | | | | | | | | |
| 8,9 | 88.7 | 98.6 | 99.7 | 76.22 | 94.24 | 97.00 | 554.46 | 31.25 | 16.40/epoch | +48.7% | +48.1% |
| 8,10 | 89.1 | 98.3 | 99.5 | 76.98 | 94.18 | 97.16 | 555.22 | | | | |
| 8,9,10 | 88.2 | 98.5 | 99.7 | 76.02 | 93.70 | 96.86 | 552.98 | 41.49 | 22.23/epoch | +97.5% | +100.8% |
| 8 | 89.4 | 98.6 | 99.4 | 77.22 | 94.20 | 97.06 | 555.88 | 21.01 | 11.07/epoch | - | - |

## A.3 ALIGNMENT ON MORE KEY-LAYERS

In the main paper, we demonstrate the effectiveness of selecting a single intermediate transformer layer (Layer 8) as the key alignment layer for DKP. To further investigate whether aligning multiple intermediate layers simultaneously could yield additional performance benefits, we conduct extended experiments involving combinations of Layers 8, 9, and 10. The detailed retrieval results, training overhead, and parameter counts are summarized in Table 6.

We observe that introducing multiple key-layer alignments ({8,9}, {8,10}, and {8,9,10}) indeed increases the number of learnable parameters, ranging from 31.25M (two-layer alignment) to 41.49M (three-layer alignment). However, this also significantly increases the computational cost. For example, on ViT-B/32, training time per epoch nearly doubles from 3.97 minutes (single key-layer alignment at Layer 8) to 7.91 minutes when aligning Layers 8, 9, and 10 simultaneously. Similar patterns emerge on ViT-B/16, indicating a clear trade-off between alignment depth and computational efficiency.

Interestingly, despite increased complexity, aligning multiple layers does not yield performance improvements. Specifically, on ViT-B/32, multi-layer alignments ({8,9}, {8,10}, {8,9,10}) obtain RSUM values of 523.40, 526.86, and 521.98, respectively, all of which are consistently lower than the single-layer alignment strategy at Layer 8 (530.20). A similar trend is revealed with ViT-B/16, where the highest RSUM (555.88) is also achieved by the single-layer alignment approach. This suggests that performing alignment at multiple layers introduces redundancy or interference, potentially harming the discriminative capacity of intermediate features.

In conclusion, while exploring additional intermediate alignments intuitively seems beneficial, our comprehensive evaluation indicates that a single carefully chosen intermediate key-layer (Layer 8, based on our empirical and layer-wise attention analysis) strikes the optimal balance between retrieval performance, parameter efficiency, and computational overhead. This result further validates the effectiveness and practical superiority of the proposed DKP paradigm. Furthermore, given that existing VLMs are all pre-trained on massive image-text datasets, they should all possess certain key layers to capture cross-modal semantic correspondences. Once these key layers are identified, our proposed approach can be directly deployed on these VLMs for efficient ITR.

## A.4 VISUALIZATION OF IMAGE-TEXT RETRIEVAL

To better illustrate the qualitative effectiveness of our proposed method, we present several examples of bi-directional retrieval results in Figure 8. In the image-to-text scenario, our model accurately retrieves the most semantically aligned captions for each image, achieving perfect top-3 retrievals in both cases shown. For text-to-image retrieval, our method successfully ranks the ground-truth image as the top-1 result, highlighted in green boxes. Notably, it also retrieves visually and contextually similar images in the top ranks (in orange boxes), indicating strong fine-grained alignment and robust generalization beyond exact matches. These results demonstrate that our approach can effectively capture nuanced visual-textual correspondences and yield semantically coherent rankings across modalities.

**(a) Image-to-text**

① a bowl of apples and a bowl of oranges.

② a counter top with a bowl of oranges next to a bowl of apples and a lamp.

③ a bowl has apples and another has oranges in it.

① a group of people walking down a rain soaked street holding umbrellas.

② people carrying colorful umbrellas walking down sidewalk in the rain.

③ people walking down a wet sidewalk holding umbrellas.

**(b) Text-to-image**

a very small bathroom with a pair of shoes.

a snowboarder in a black jacket is doing a trick.

Figure 8: Visualization of bi-directional retrieval examples. In image-to-text, DKP retrieves the correct captions consistently within top-3 results, while in text-to-image, it ranks the ground-truth as top-1.

## A.5 VISUALIZATION OF TOKEN-IMAGE ATTENTION

Figure 9 contrasts the last-layer token-to-patch attention for vanilla CLIP and DKP on four randomly chosen MS-COCO samples. Each caption token is shown with its normalised weight (subscripts), and the corresponding attention map overlays the image. We find several key insights: (1) Saliency amplification. Across both backbones, DKP elevates the weights of truly discriminative tokens. For example, motorcycle 0.18 vs 0.08 and helmets 0.11 vs 0.05 in the first row; giraffe 0.25 vs 0.16 in the second. The same trend appears for sheep 0.17 vs 0.11 and laptop 0.19 vs 0.09 with the deeper ViT-B/16. (2) Sharper spatial focus. Higher token scores translate into heat-maps that lock onto the relevant regions (handlebars and rider, giraffe torso, grazing flock, laptop screen) while suppressing diffuse background activation present in CLIP. (3) Backbone consistency. Improvements are evident on both ViT-B/32 (rows (a)–(b)) and ViT-B/16 (rows (c)–(d)), confirming that Key-layer Pre-Alignment plus Semantic Consistency Distillation sharpen cross-modal correspondence independently of model depth. These qualitative results corroborate the quantitative gains reported in Table 1 and Table 2 of the main paper: DKP not only retrieves correct matches more often, but does so by forming cleaner, semantically faithful alignments between language tokens and visual evidence.

## A.6 VISUALIZATION OF INTRA-MODAL COHESION

Figure 10 juxtaposes t-SNE embeddings for CLIP and DKP on two datasets (MS-COCO, Flickr30K) and two backbones (ViT-B/32 and ViT-B/16). For each setting, we randomly sample 1000 image-text pairs and extract L2-normalized final layer features.

Across all four panels, the pattern is clear. In the DKP plots (b,d), colored islands are markedly denser and more isolated. For instance, the magenta and cyan blobs in COCO-Text, or the yellow–orange cluster in Flickr-Image, whereas the same hues disperse under vanilla CLIP (a,c), form elongated or fractured shapes. The effect holds for both modalities: textual embeddings contract around shared semantics (verbs, nouns, attributes) and visual embeddings around coherent object layouts. Importantly, the improvement persists when moving from the shallower ViT-B/32 to the deeper ViT-B/16, confirming that Semantic Consistency Distillation sharpens intra-modal geometry

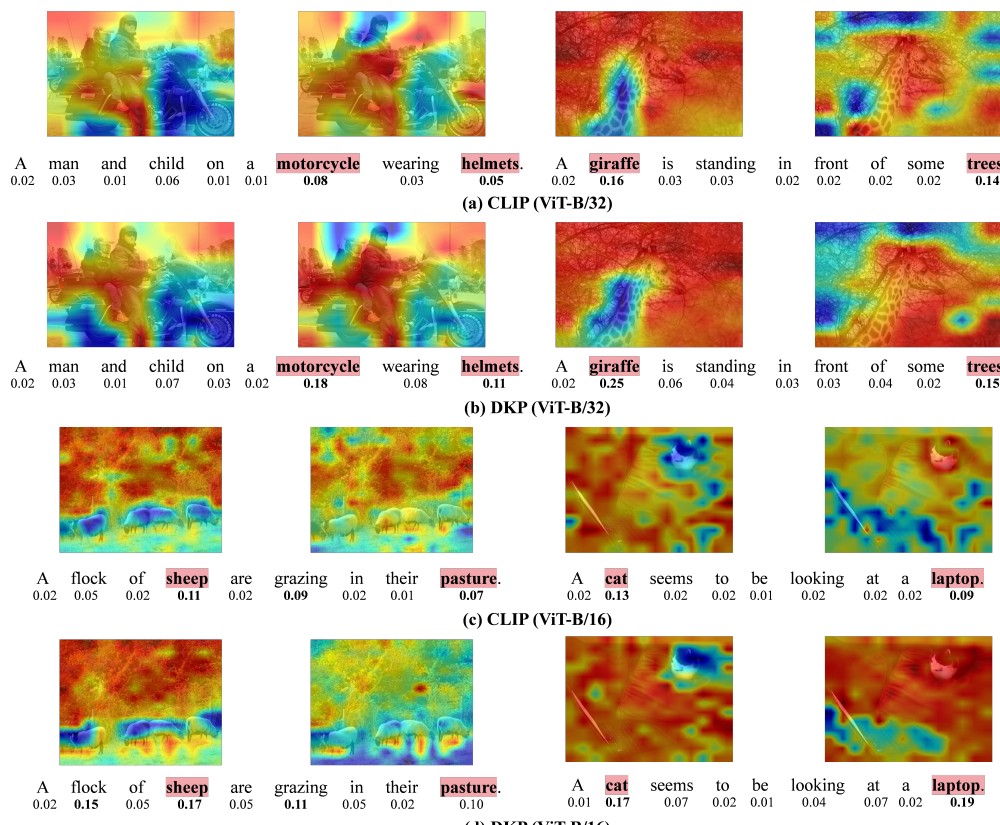

Figure 9: Visualization of token–image attention for CLIP vs. DKP on MS-COCO. DKP increases weights on key tokens (e.g., "motorcycle", "giraffe", "laptop") and produces cleaner spatial focus maps.

independently of encoder depth and dataset domain. These qualitative gains align with the quantitative RSUM boosts reported in the main paper, illustrating that DKP's structure-aware regularization curbs semantic drift and yields embeddings that are simultaneously more clustered within modalities and more discriminative across modalities.

### A.7 PROMPT LEARNING FOR IMAGE-TEXT RETRIEVAL

We have also explored a pure prompt learning approach for image-text retrieval, that is, freezing all parameters of CLIP and only adding learnable visual and textual prompt information to train CLIP. Figure 11 plots retrieval accuracy (RSUM) against the number of learnable parameters on MS-COCO (ViT-B/32). Each method is shown with a distinct marker: a purple cross for zero-shot CLIP, a cyan square for full fine-tuning CLIP, a green star for prompt learning, an orange pentagon for CUSA, and a blue triangle for our DKP. The dashed yellow arrows illustrate the "efficiency path". As we observe, prompt learning already recovers most of the full-tuning score with only 7.12M learnable parameters, yet plateaus around 417.9 RSUM. The prompt method in this approach mimics the CoOp approach by incorporating prompt information into the model's architecture, which consists of 7.12M parameters. As shown in the Figure 11, the inference FLOPs is 8.56G, and the evaluation time is 29 seconds. Pushing to CUSA's or full-tuning CLIP's 150M+ updates yields diminishing gains ($\leq$ 429.7 RSUM). By contrast, **DKP** achieves the best performance (430.81 RSUM) while touching just 21.01M weights. In addition, the inference FLOPs is 4.98G, and the evaluation time is 20 seconds, clearly bending the curve upward and establishing a new Pareto frontier for parameter-efficient image–text retrieval. This also reflects a phenomenon: prompt learning may be effective in cross-modal tasks, but it is insufficient to capture complex downstream data distributions (such as the intricate image-text semantic scenarios in the MS-COCO dataset). While full-tuning

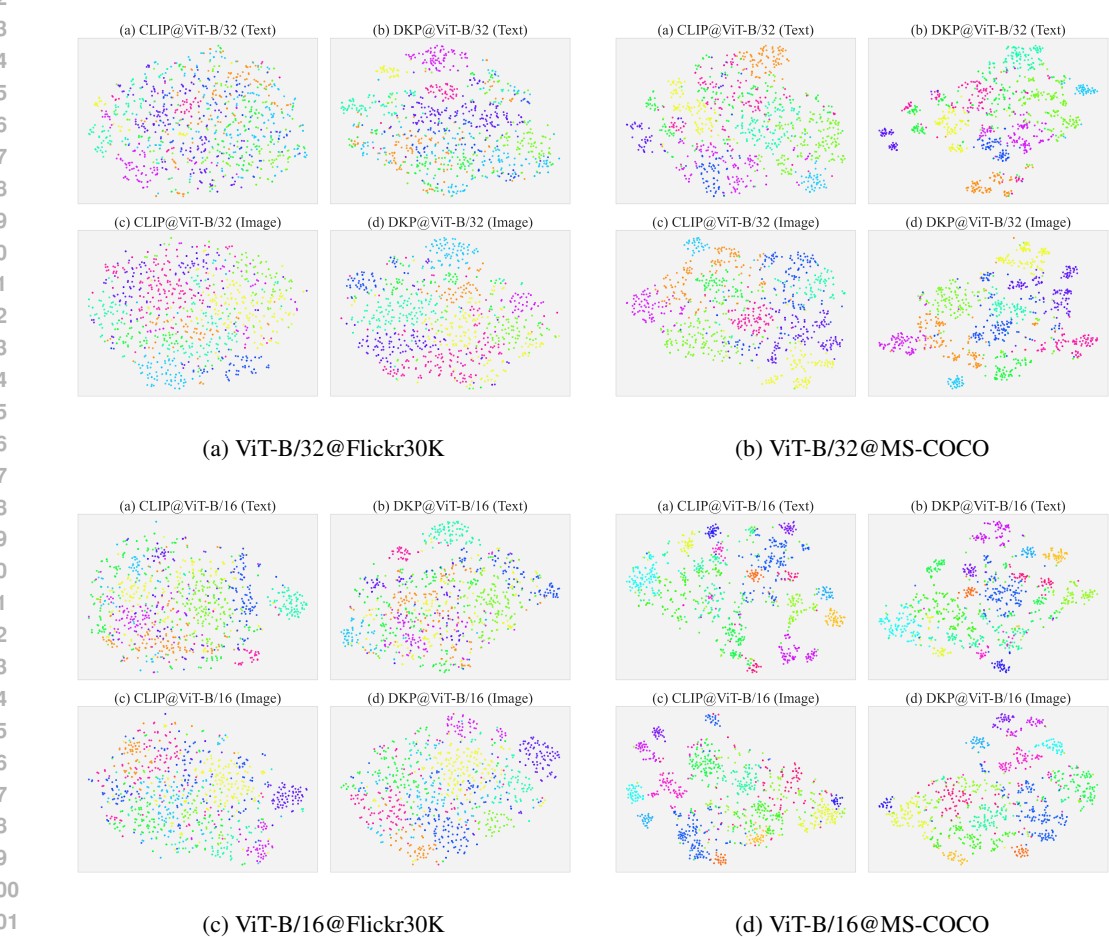

Figure 10: Visualization of intra-modal cohesion using t-SNE embeddings for CLIP and DKP across datasets and backbones. DKP embeddings form tighter, more isolated clusters in both image and text spaces, confirming that SCD reduces intra-modal drift.

Table 7: Architecture comparison of **DKP** with representative parameter-efficient fine-tuning (PEFT) baselines with ViT-B/16. "CLS" = classification; "Mod." = moderate overhead; "∘" = partial coupling (only if both branches adapted).

| Method | What is learned? | Adapted branches | Added params (ViT-B/16) | Extra FLOPs | Needs prompts? | Cross-modal coupling | Intra-modal regulariser | Typical tasks |
|---|---|---|---|---|---|---|---|---|
| CoOp | Static text context tokens | Text | ∼0.03 M | None | ✓ | ✗ | ✗ | Few-shot CLS |
| CoCoOp | Dynamic context (MLP) | Text | 1–2 M | Low | ✓ | ✗ | ✗ | Domain-shift CLS |
| MaPLe | Layer-wise prompts + coupling fn. | Both | 2–3 M | Mod. | ✓ | ✓ | ✗ | Few/zero-shot CLS |
| CLIP-Adapter | Residual bottleneck adapters | Either | 3–5 M | Low | ✗ | ∘ | ✗ | Few-shot CLS |
| VPT | Visual prompts (shallow/deep) | Vision | 0.4–0.6 M | Mod. | ✗ | ✗ | ✗ | Vision tasks |
| LoRA | Low-rank $\Delta W$ on Q/K/V | Any | 3–4 M | None | ✗ | Config. | ✗ | Generic |
| **DKP (Ours)** | Two trainable blocks + SCD head | Both | 21 M | None | ✗ | ✓ | ✓ | ITR |

might be sufficiently effective, it demands an enormous amount of computational resources. A potential future solution lies in combining prompt learning with fine-tuning of key parameters, thereby achieving a more optimal trade-off for image-text retrieval.

## A.8 DISCUSSION WITH PEFT METHODS

We summarize and discuss the architecture design of DKP and representative PEFT baselines in Table 7. Main-stream PEFT strategies can be divided into surface tuning—where prompt vectors (CoOp Zhou et al. (2022b), CoCoOp Zhou et al. (2022a), VPT Jia et al. (2022)) are injected at the

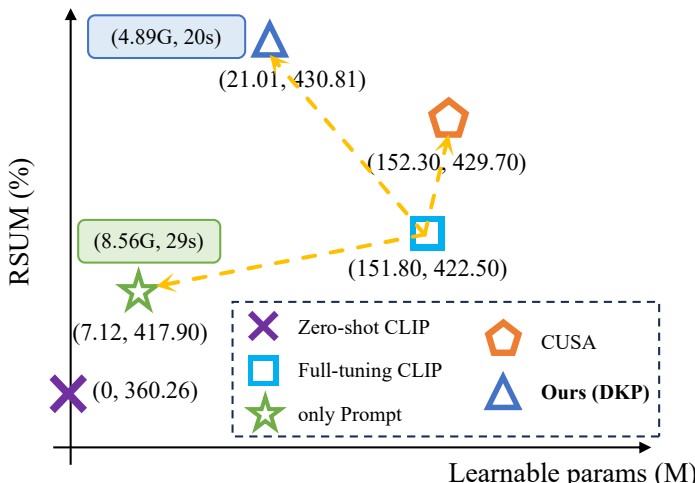

Figure 11: Comparison with prompt learning. The information within the box represents (FLOPs, evaluation time).

input while the backbone remains entirely frozen—and lightweight projection techniques such as Adapters Gao et al. (2024a) or LoRA Hu et al. (2022), which attach small bottleneck layers after each transformer block. By contrast, **DKP** unlocks the model exactly where semantics first mature: the "sweet-spot" intermediate layer (Layer 8) and the final block. As a result, it manipulates less than 14% of the backbone parameters while still accessing the model's internal reasoning pathway.

Unlike prompt-only methods that update solely the text branch, only MaPLe Khattak et al. (2023) and DKP explicitly couple vision & language updates. DKP's *Key-layer Pre-Alignment* loss directly contrasts paired visual and linguistic embeddings at Layer 8, producing a stronger binding between modalities than text-side prompting. Furthermore, existing PEFT baselines leave the internal geometry of each modality unconstrained; **DKP** remedies this through Semantic Consistency Distillation, which transfers image–image and text–text topology into the cross-modal space and effectively suppresses semantic drift—an ability crucial for fine-grained retrieval yet absent from classification-oriented PEFT designs.

Although DKP tunes more weights ($\approx$21M) than prompt or adapter schemes, it still removes at least 86% of the parameters required for full fine-tuning and introduces zero additional forward FLOPs, because all layers are already executed in the vanilla CLIP inference pass. Empirically, this translates into 20–30% faster training and roughly one-third fewer epochs to converge (see Table 3 of the main paper). Finally, whereas most PEFT methods are tailored to closed-set recognition, DKP is purpose-built for image–text retrieval; it matches and even surpasses full fine-tuning on ranking metrics while remaining markedly more portable.

To further contextualize the effectiveness of DKP among existing representative PEFT methods like Partial Jia et al. (2022), Projection Jia et al. (2022), CoOp Zhou et al. (2022b), LoRA Zanella & Ayed (2024), VPT Jia et al. (2022), VLPrompt Zhou et al. (2023), MaPLe Khattak et al. (2023), and CLIP-Adapter Gao et al. (2024a), we provide a quantitative performance comparison on MS-COCO using the ViT-B/32 backbone, as summarized in Table 8. While prompt-based

Table 8: Performance (%) comparison of DKP with representative PEFT baselines on MS-COCO using ViT-B/32.

| Methods | Image-to-Text | | | Text-to-Image | | | RSUM |
|---|---|---|---|---|---|---|---|
| | R@1 | R@5 | R@10 | R@1 | R@5 | R@10 | |
| Partial | 57.2 | 83.3 | 90.5 | 42.7 | 71.4 | 81.5 | 426.6 |
| Projection | 55.1 | 81.5 | 89.2 | 40.0 | 68.3 | 78.7 | 412.8 |
| LoRA | 58.22 | 83.30 | 89.62 | 42.60 | 70.85 | 80.73 | 425.32 |
| CoOp | 53.7 | 77.9 | 86.1 | 36.1 | 62.2 | 72.9 | 388.9 |
| VPT | 55.5 | 79.5 | 87.9 | 40.1 | 67.0 | 77.4 | 407.4 |
| VLPrompt | 59.8 | 83.2 | 90.5 | 43.6 | 70.8 | 80.4 | 428.3 |
| MaPLe | 59.8 | 82.8 | 89.9 | 43.4 | 70.8 | 80.3 | 427.0 |
| CLIP-Adapter | 53.18 | 78.82 | 87.38 | 38.06 | 66.21 | 77.34 | 400.99 |
| **DKP (ViT-B/32)** | 58.28 | 83.54 | 90.24 | 44.12 | 72.17 | 82.46 | 430.81 |

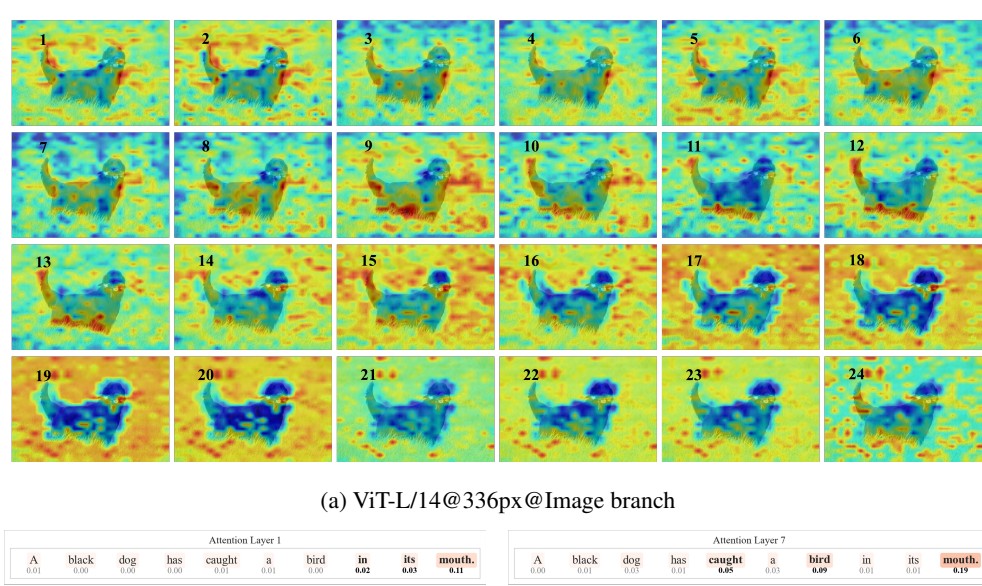

(a) ViT-L/14@336px@Image branch

(b) ViT-L/14@336px@Text branch

Figure 12: Layer-wise analysis of CLIP with ViT-L/14@336px. Attention in the image branch crystallizes around Layer 17, while the text branch matures earlier at Layer 8, revealing asymmetric semantic depths.

methods such as CoOp, VPT, and MaPLe are highly parameter-efficient (ranging from 0.03M to 3M parameters), they consistently underperform in the retrieval task, with RSUM scores between 388.9 and 428.3. This gap highlights a key limitation of prompt-only strategies: although they reduce trainable parameters significantly, they often fail to capture the nuanced semantic structures required for fine-grained cross-modal alignment.

In contrast, DKP achieves a superior RSUM of 430.81 with only 21M parameters—significantly fewer than full fine-tuning (≈151M) and even CUSA (≈152M). More importantly, DKP outperforms all prompt-based and adapter-based PEFT methods, confirming that its design, which combines key-layer pre-alignment and semantic consistency distillation, is better suited for retrieval tasks than generic classification-oriented PEFT approaches.

This result aligns with our earlier discussion: while methods like MaPLe introduce cross-modal coupling through prompts, they lack explicit mechanisms to preserve intra-modal semantic structure. DKP's integration of both cross-modal and intra-modal regularization enables it to achieve higher performance without external knowledge or cumbersome full tuning.

### A.9 LAYER-WISE ANALYSIS AND PERFORMANCE COMPARISONS ON ASYMMETRIC BACKBONE (CLIP WITH VIT-L/14@336PX)

The CLIP family is no longer architecturally symmetric once the image encoder scales to ViT-L/14@336px (24 transformer blocks) while the text branch remains at 12 layers. We conduct layer-wise analysis on this architecture. Figure 12(a) plots class-activation maps for every visual block when the caption is "A black dog has caught a bird in its mouth." Salient regions remain diffuse until

Table 9: Performance comparisons on CLIP (ViT-L/14@336px). Note that † denotes CLIP without fine-tuning and ‡ denotes CLIP with full-parameter fine-tuning.

| Methods | Image-to-Text | | | Text-to-Image | | | RSUM | Learnable parameters (M) |
|---|---|---|---|---|---|---|---|---|
| | R@1 | R@5 | R@10 | R@1 | R@5 | R@10 | | |
| *Flickr30K (1K test set)* | | | | | | | | |
| CLIP-ViT† (ViT-L/14) | 86.70 | 98.10 | 99.00 | 67.22 | 88.90 | 93.28 | 533.20 | 0 |
| CLIP-ViT‡ (ViT-L/14) | 87.30 | 99.00 | 99.50 | 76.40 | 94.80 | 97.40 | 554.40 | 429.13 |
| CUSA (ViT-L/14) | 90.80 | 99.10 | 99.70 | 77.40 | 95.50 | 97.70 | 560.20 | 430.13 |
| **DKP (ViT-L/14)** | 93.30 | 99.50 | 99.90 | 82.76 | 96.88 | 98.40 | **570.74** | **40.55** |
| *MS-COCO (5K test set)* | | | | | | | | |
| CLIP-ViT† (ViT-L/14) | 57.08 | 80.52 | 87.80 | 36.44 | 60.88 | 71.02 | 393.74 | 0 |
| CLIP-ViT‡ (ViT-L/14) | 67.10 | 89.40 | 94.70 | 51.60 | 79.10 | 87.70 | 469.60 | 429.13 |
| CUSA (ViT-L/14) | 67.90 | 90.30 | 94.70 | 52.40 | 79.80 | 88.10 | 473.10 | 430.13 |
| **DKP (ViT-L/14)** | 70.80 | 90.46 | 95.38 | 54.59 | 80.62 | 88.39 | **480.24** | **40.55** |

roughly Layer 15–18, after which attention converges on the dog's head and the captured bird. The parallel text heat-maps Figure 12(b) reach comparable focus much earlier—around Layer 7–8—and then saturate. This confirms that the two modalities crystallize at different depths, a property hidden in the symmetric ViT-B experiments. Nevertheless, there still exist certain key layers across different architectures, presenting potential for pre-alignment. Guided by this observation, we implement exploratory experiments by aligning Layer 17 in the image branch with Layer 8 in the text branch on this architecture. We record the performance and parameter scale in Table 9 and find the following three discoveries.

**Performance advantage.** With less than 10% trainable parameters of full fine-tuning, DKP establishes a new SOTA on both datasets, adding +10.54 RSUM on Flickr30K and +7.14 RSUM on MS-COCO relative to full-parameter fine-tuning CUSA and CLIP‡. The gains are most pronounced on the caption-rich Flickr30K set, suggesting that cross-layer pre-alignment effectively captures fine-grained semantics once the deeper visual features are exposed. Note that CLIP† containing only pre-trained prior knowledge yields the poorest performance because it did not undergo any parameter training and thus could not adapt to the complex cross-modal semantic correspondences for downstream tasks.

**Parameter efficiency.** Updating only two internal blocks, two last blocks, and the output head (about 40.55M parameters, one order of magnitude lighter than any competing method for ViT-L), DKP recovers and even surpasses the retrieval performance of full network optimization, demonstrating the performance-efficiency balance of our method on larger architectures.

**Generalisability of the "key-layer" concept.** Although the absolute depth differs (Layer 8 in ViT-B vs. Layer 17 in ViT-L), a "semantic knee" still exists in both backbones. **We leave these as open research questions for the community, allowing researchers to explore the pre-alignment of key layers in various VLMs variants for their downstream cross-modal tasks.**

**Early alignment and key-layer alignment of ViT-L/14.** To investigate whether early cross-modal alignment could guide visual representation formation, we conducted experiments on the asymmetric CLIP ViT-L/14 model, comparing two alignment strategies: (1) symmetric alignment at the same layer index (Image Layer 8 + Text Layer 8) and (2) asymmetric alignment at semantically crystallized layers (Image Layer 17 + Text Layer 8). The results, summarized in Table 10, clearly demonstrate that symmetric alignment at Layer 8 leads to a noticeable performance drop on both datasets: RSUM decreases from 570.74 to 567.28 on Flickr30K and from 480.24 to 461.88 on MS-COCO. This performance degradation indicates that forcing alignment before the image encoder reaches semantic maturity introduces noise and misalignment, rather than providing beneficial guidance. Our layer-wise attention analysis (Appendix A.9 and Figure 12) confirms that the image encoder of ViT-L/14 exhibits diffuse and unstable attention patterns at Layer 8, lacking focused semantic saliency. In contrast, Layer 17 represents a point where visual features have crystallized, enabling robust cross-modal matching with the text encoder's mature representations at Layer 8.

Table 10: Comparison between early alignment in the ViT-L/14 architecture (i.e., alignment at layer 8 of the image and text encoders) and key-layer alignment at the semantic crystallization layer (layer 17 for image and layer 8 for text).

| Image Layer | Text Layer | Image-to-Text | | | Text-to-Image | | | RSUM |
|:---:|:---:|:---:|:---:|:---:|:---:|:---:|:---:|:---:|
| | | R@1 | R@5 | R@10 | R@1 | R@5 | R@10 | |
| *Flickr30K (1K test set)* | | | | | | | | |
| 8 | 8 | 92.20 | 99.60 | 99.80 | 80.90 | 96.48 | 98.30 | 567.28 |
| 17 | 8 | 93.30 | 99.50 | 99.90 | 82.76 | 96.88 | 98.40 | **570.74** |
| *MS-COCO (5K test set)* | | | | | | | | |
| 8 | 8 | 65.42 | 87.94 | 93.48 | 51.42 | 77.74 | 85.88 | 461.88 |
| 17 | 8 | 70.8 | 90.46 | 95.38 | 54.59 | 80.62 | 88.39 | **480.24** |

Table 11: Results of different methods counted on RSITMD. The best results among all the methods are highlighted in bold, while the second-best results are marked with an underline.

| *RSITMD* | | | | | | | | |
|:---:|:---:|:---:|:---:|:---:|:---:|:---:|:---:|:---:|
| Methods | Reference | Image-to-Text | | | Text-to-Image | | | mR |
| | | R@1 | R@5 | R@10 | R@1 | R@5 | R@10 | |
| MSA | arXiv'24 | 13.63 | 40.80 | 57.92 | 19.25 | 37.61 | 50.44 | 36.61 |
| SMLGN | TGRS'24 | 13.19 | 43.94 | 60.40 | 17.26 | 39.38 | 51.55 | 37.62 |
| SCAT-PRG | TGRS'24 | 12.11 | 42.15 | 61.97 | 16.82 | 32.73 | 48.12 | 35.65 |
| SIRS | TGRS'24 | 11.98 | 41.29 | 60.15 | 15.25 | 35.01 | 48.73 | 35.40 |
| CLIP-Adapter | IJCV'24 | 13.30 | 40.20 | 60.06 | 12.83 | 28.84 | 39.05 | 32.38 |
| CoOp | IJCV'22 | 9.16 | 33.85 | 54.35 | 12.19 | 30.69 | 42.82 | 30.51 |
| CoCoOp | CVPR'22 | 10.21 | 34.20 | 53.36 | 14.53 | 32.89 | 45.94 | 31.85 |
| VPT | ECCV'22 | 15.97 | 41.35 | 60.35 | 14.98 | 32.05 | 40.15 | 34.14 |
| MaPLe | CVPR'23 | 13.59 | 43.25 | 61.92 | 16.96 | 39.97 | 53.02 | 38.12 |
| CUP | TNNLS'25 | 17.70 | **48.40** | **64.82** | **21.57** | 43.70 | 56.75 | 42.16 |
| **DKP (ViT-B/32)** | - | **25.88** | 46.68 | 58.63 | 21.28 | **51.28** | **66.73** | **45.08** |

## A.10 STRONG ROBUSTNESS ON DIVERSE DATASETS

**Datasets and evaluation metrics.** We further verify the strong robustness of DKP on three diverse cross-modal remote sensing datasets including RSITMD Yuan et al. (2022), RSICD Lu et al. (2018), and UCM-Captions Qu et al. (2016). Among them, UCM-Captions is a relatively small-scale dataset that contains 2,100 images belonging to 21 different scene categories. Each image is accompanied by five descriptions, and this dataset is renowned for the high quality of its annotations. RSICD, on the other hand, is a large-scale dataset that includes 10,921 images across 30 scene categories. It has played a significant role in advancing research in this field; however, some of the descriptions in this dataset are repetitive across different images, resulting in a lack of textual diversity. RSITMD is a dataset specifically designed for fine-grained cross-modal retrieval. It contains 4,743 images, and its annotations are carefully crafted to enhance the uniqueness and accuracy of the descriptions by emphasizing object properties and spatial relationships. In addition, keyword annotations are also provided, which effectively alleviates the problems of repetitive sentences and high intra-class similarity observed in the aforementioned datasets. For cross-modal retrieval tasks, we primarily report the R@K value (i.e., K = 1, 5, 10) and the mR value (the average of multiple R@K values) in order to comprehensively evaluate the model's performance under different retrieval scenarios.

**State-of-the-art baselines.** These baselines include both non-CLIP-based approaches, such as SMLGN Chen et al. (2024), SCAT-PRG Wu et al. (2024), MSA Yang et al. (2024) and SIRS Zhu et al. (2024), as well as CLIP-based solutions including the official CLIP-Adapter Gao et al. (2024a), CoOp Zhou et al. (2022b), CoCoOp Zhou et al. (2022a), VPT Jia et al. (2022), MaPLe Khattak et al. (2023), CUP Wang et al. (2025).

Table 12: Results of different methods counted on RSICD. The best results among all the methods are highlighted in bold, while the second-best results are marked with an underline.

| Methods | Reference | Image-to-Text | | | Text-to-Image | | | mR |
|---|---|---|---|---|---|---|---|---|
| | | R@1 | R@5 | R@10 | R@1 | R@5 | R@10 | |
| MSA | arXiv'24 | 5.97 | 22.93 | 39.58 | 7.96 | 23.33 | 35.77 | 22.59 |
| SMLGN | TGRS'24 | 7.85 | 27.14 | 42.58 | 8.87 | 25.53 | 37.24 | 24.87 |
| SCAT-PRG | TGRS'24 | 7.76 | 25.20 | 38.69 | 10.13 | 25.72 | 38.24 | 24.29 |
| SIRS | TGRS'24 | 5.76 | 20.50 | 33.78 | 7.34 | 20.00 | 30.78 | 19.69 |
| CLIP-Adapter | IJCV'24 | 7.67 | 24.87 | 39.73 | 7.11 | 19.48 | 31.01 | 21.65 |
| CoOp | IJCV'22 | 4.31 | 17.89 | 31.73 | 6.32 | 15.89 | 27.60 | 17.29 |
| CoCoOp | CVPR'22 | 6.15 | 21.68 | 34.29 | 7.65 | 24.25 | 36.08 | 21.68 |
| VPT | ECCV'22 | 6.94 | 25.26 | 40.73 | 7.23 | 19.40 | 30.77 | 21.72 |
| MaPLe | CVPR'23 | 6.88 | 24.47 | 40.12 | 7.93 | 26.78 | 42.42 | 24.77 |
| CUP | TNNLS'25 | 9.86 | **31.60** | **47.15** | **12.05** | 32.66 | **48.70** | **30.34** |
| **DKP (ViT-B/32)** | - | **12.26** | 31.11 | 44.28 | 10.56 | **32.72** | 47.32 | 29.71 |

Table 13: Results of different methods counted on UCM-Captions. The best results among all the methods are highlighted in bold, while the second-best results are marked with an underline.

| Methods | Reference | Image-to-Text | | | Text-to-Image | | | mR |
|---|---|---|---|---|---|---|---|---|
| | | R@1 | R@5 | R@10 | R@1 | R@5 | R@10 | |
| MSA | arXiv'24 | 11.52 | 53.24 | 89.62 | 15.24 | 48.57 | 70.48 | 48.11 |
| SMLGN | TGRS'24 | 14.29 | 52.76 | 84.67 | 12.86 | 49.52 | 75.71 | 48.30 |
| SIRS | TGRS'24 | 14.28 | 56.05 | 82.58 | 14.50 | 48.40 | 72.51 | 48.05 |
| CLIP-Adapter | IJCV'24 | 14.61 | 51.03 | 84.14 | 9.83 | 42.52 | 66.79 | 44.82 |
| CoOp | IJCV'22 | 9.78 | 48.34 | 87.41 | 7.19 | 42.76 | 74.19 | 44.95 |
| CoCoOp | CVPR'22 | 11.84 | 51.21 | 84.81 | 14.76 | 47.94 | 77.94 | 48.08 |
| VPT | ECCV'22 | 14.73 | **58.01** | **92.44** | 13.81 | 49.05 | 75.24 | 50.54 |
| MaPLe | CVPR'23 | 12.21 | 52.80 | 84.81 | 13.97 | 49.68 | 75.24 | 48.12 |
| CUP | TNNLS'25 | 15.06 | 56.98 | 91.14 | 15.24 | 54.76 | 79.37 | 52.09 |
| **DKP (ViT-B/32)** | - | **20.95** | 57.14 | 80.48 | **16.76** | **61.14** | **95.71** | **55.36** |

**Results of RSITMD.** As shown in Table 11, DKP achieves state-of-the-art performance on the fine-grained RSITMD dataset, obtaining the highest overall mR of 45.08. Notably, DKP demonstrates particularly strong performance in text-to-image retrieval, where it achieves the best results in R@5 (51.28) and R@10 (66.73), and competitive results in R@1 (21.28). This indicates that our method effectively leverages semantic consistency to align textual queries with relevant images, even in fine-grained scenarios. While CUP achieves the best performance in some image-to-text metrics (R@5 and R@10), DKP significantly outperforms all compared methods in R@1 for image-to-text retrieval (25.88), highlighting its capability for precise matching. These results validate that DKP's key-layer pre-alignment and semantic consistency distillation mechanisms enhance cross-modal alignment even in specialized domains.

**Results of RSICD.** On RSICD dataset (see Table 12), DKP demonstrates highly competitive performance, achieving an mR of 29.71, which is comparable to the top-performing method CUP (30.34). DKP obtains the best results in R@1 for image-to-text retrieval (12.26) and R@5 for text-to-image retrieval (32.72), while securing second-best performance in multiple other metrics including R@5 (31.11) and R@10 (44.28) for image-to-text retrieval, and R@10 (47.32) for text-to-image retrieval. Although CUP achieves slightly better overall mR, DKP shows more balanced performance across both retrieval directions, particularly excelling in the critical R@1 metric for image-to-text retrieval.

Table 14: Cross-domain retrieval performance from general datasets to remote sensing datasets. Models are trained on Flickr30K or MS-COCO and tested directly on remote sensing datasets without fine-tuning.

| Training Dataset | Test Dataset | Method | Image-to-Text | | | Text-to-Image | | | RSUM |
|---|---|---|---|---|---|---|---|---|---|
| | | | R@1 | R@5 | R@10 | R@1 | R@5 | R@10 | |
| Flickr30K | RSICD | CLIP | 3.29 | 8.60 | 14.55 | 2.85 | 10.14 | 16.76 | 56.19 |
| Flickr30K | RSICD | CUSA | 3.39 | 9.79 | 15.46 | 3.37 | 13.52 | 21.94 | 67.44 |
| Flickr30K | RSICD | Ours | 4.03 | 10.80 | 17.84 | 4.48 | 15.57 | 25.65 | 78.37 |
| Flickr30K | RSITMD | CLIP | 5.09 | 13.50 | 22.79 | 5.35 | 16.42 | 25.80 | 88.95 |
| Flickr30K | RSITMD | CUSA | 6.86 | 18.36 | 27.21 | 5.58 | 21.55 | 35.49 | 115.05 |
| Flickr30K | RSITMD | Ours | 5.53 | 17.48 | 26.77 | 7.17 | 22.88 | 38.10 | 117.93 |
| MS-COCO | RSICD | CLIP | 3.57 | 8.78 | 15.65 | 4.03 | 12.61 | 21.50 | 66.14 |
| MS-COCO | RSICD | CUSA | 3.29 | 10.25 | 16.74 | 3.51 | 13.32 | 22.82 | 69.93 |
| MS-COCO | RSICD | Ours | 5.49 | 13.54 | 21.50 | 5.16 | 17.51 | 27.37 | 90.57 |
| MS-COCO | RSITMD | CLIP | 5.53 | 15.27 | 22.35 | 5.58 | 21.59 | 33.76 | 104.08 |
| MS-COCO | RSITMD | CUSA | 6.19 | 17.26 | 25.44 | 7.52 | 23.27 | 34.69 | 114.37 |
| MS-COCO | RSITMD | Ours | 6.64 | 18.58 | 28.76 | 8.36 | 26.77 | 42.39 | 131.50 |

This demonstrates DKP's effectiveness in handling datasets with textual repetition and high intra-class similarity.

**Results of UCM-Captions.** For UCM-Captions dataset (see Table 13), DKP achieves the best overall performance with a mR of 55.36, significantly outperforming all baseline methods. It obtains the highest scores in R@1 for image-to-text retrieval (20.95) and in R@1 (16.76), R@5 (61.14), and R@10 (95.71) for text-to-image retrieval. The remarkable performance in text-to-image retrieval, particularly the near-perfect R@10 score of 95.71, demonstrates DKP's exceptional capability in this domain. While VPT achieves the best results in some image-to-text metrics (R@5 and R@10), DKP's balanced superiority across both retrieval directions confirms that our method effectively leverages semantic alignment in scenarios with well-defined semantic categories.

## A.11 CROSS-DOMAIN GENERALIZATION ABILITY

In order to evaluate the generalization ability and external validity of DKP, we conducted extensive cross-domain retrieval experiments. These experiments included cross-domain migrations from general-purpose datasets (Flickr30K and MS-COCO) to remote sensing datasets (RSICD and RSITMD). The remote sensing datasets represent various visual domains and possess unique feature distributions, which allows us to test the model's robustness to domain-specific variations. We tested domain generalization by evaluating models trained on Flickr30K or MS-COCO directly on remote sensing datasets without fine-tuning. Results in Table 14 show that DKP maintains competitive performance despite the domain gap. For example, when trained on MS-COCO and tested on RSITMD, DKP achieves an RSUM of 131.50, substantially higher than CUSA (114.37) and CLIP (104.08). Similar trends are observed on RSICD, where DKP improves over baselines. These results demonstrate the stronger zero-shot capability of our method, which is trained on the source domain and directly applied to perform zero-shot retrieval tasks in the target domain.

## A.12 PERFORMANCE CHANGE WITH DATA PATTERN

**Dataset size.** To further analyze the correlation between model performance and dataset size, we conducted experiments on Flickr30K and MS-COCO datasets using progressively larger subsets of the training data (30%, 50%, 70%, 90%, and 100%). The models were evaluated on the full test sets, and the results are summarized in Table 15 and Table 16. The results show a consistent positive correlation between dataset size and performance (measured by RSUM) for both datasets. However, the rate of improvement decelerates as more data is added, indicating diminishing returns. For instance, on Flickr30K, the RSUM increases from 502.66 (30% data) to 530.20 (100% data), but the

Table 15: Performance on Flickr30K with varying training data proportions.

| Data Ratio | Image-to-Text | | | Text-to-Image | | | RSUM |
|---|---|---|---|---|---|---|---|
| | R@1 | R@5 | R@10 | R@1 | R@5 | R@10 | |
| 30% | 74.5 | 92.8 | 95.9 | 61.24 | 85.88 | 92.34 | 502.66 |
| 50% | 77.2 | 93.7 | 97.1 | 64.02 | 87.68 | 93.40 | 513.10 |
| 70% | 79.2 | 93.5 | 97.4 | 65.24 | 88.86 | 94.02 | 518.22 |
| 90% | 80.3 | 95.2 | 97.7 | 66.52 | 89.42 | 94.16 | 523.30 |
| 100% | 81.2 | 96.3 | 98.9 | 68.18 | 90.76 | 94.86 | 530.20 |

Table 16: Performance on MS-COCO with varying training data proportions.

| Data Ratio | Image-to-Text | | | Text-to-Image | | | RSUM |
|---|---|---|---|---|---|---|---|
| | R@1 | R@5 | R@10 | R@1 | R@5 | R@10 | |
| 30% | 52.26 | 78.26 | 86.64 | 39.36 | 67.64 | 78.42 | 402.58 |
| 50% | 55.48 | 81.08 | 88.38 | 41.88 | 70.11 | 80.30 | 417.23 |
| 70% | 56.40 | 82.60 | 89.44 | 43.26 | 71.14 | 81.34 | 424.18 |
| 90% | 58.16 | 83.38 | 89.84 | 43.99 | 72.01 | 82.17 | 429.55 |
| 100% | 58.28 | 83.54 | 90.24 | 44.12 | 72.17 | 82.46 | 430.81 |

marginal gains reduce significantly after 70% data. A similar trend is observed on MS-COCO, where the RSUM plateaus after 70% data, rising only slightly from 429.55 (90% data) to 430.81 (100% data). This deceleration suggests that while additional data continues to improve performance, the benefits become less pronounced after a certain point. These findings highlight the data efficiency of DKP, as it achieves strong performance even with limited training data.

**Batchsize.** To investigate the impact of batch size on model performance and training stability, we conducted comprehensive experiments on Flickr30K using ViT-B/32 across varying batch sizes (32, 64, 128, and 256). As shown in Table 17, performance remains relatively stable across smaller batch sizes (32-128), with RSUM scores ranging from 529.4 to 531.56. However, when the batch size increases to 256, we observe a noticeable performance degradation (RSUM drops to 523.62), indicating potential instability in the Semantic Consistency Distillation (SCD) objective due to noisier intra-modal similarity estimates in larger batches. Motivated by the Momentum Contrast (MoCo) He et al. (2020) mechanism, we incorporated a momentum-updated queue to stabilize the intra-modal similarity targets in SCD. This modification successfully recovers performance to 530.08 RSUM at batch size 256, effectively closing the performance gap observed with standard SCD. However, this improvement comes with a computational trade-off: the MoCo-enhanced variant increases GPU memory consumption from 12,269.89 MB to 13,023.27 MB (6% increase), highlighting the inherent memory overhead of maintaining momentum-based queues. This memory-performance trade-off presents an important optimization challenge for future work, particularly when scaling to larger models or resource-constrained environments.

A.13 SEMANTIC CONVERGENCE PATTERNS OF IMAGE COMPLEXITY

To further investigate whether the semantic crystallization layer varies with input complexity, we conducted additional experiments analyzing attention patterns across both simple and complex scenarios. Note that all simple and complex images exhibit the same trend, and we illustrate this by presenting two simple images and two complex images as examples. Simple scenarios typically contain single dominant objects with clear contexts (e.g.,"a white dog is sitting on a couch"), while complex scenarios involve multiple objects, intricate interactions, or cluttered backgrounds (e.g., "a group of people playing music in public on unique instruments"). As shown in Figure 13, we observe that the semantic crystallization point remains consistently around Layer 8 across both simple and complex inputs in ViT-B architectures. While early layers (1–7) exhibit diffuse and unstable

Table 17: Performance and memory analysis under varying batch sizes on Flickr30K (ViT-B/32).

| Batch Size | GPU Memory (MB) | Image-to-Text | | | Text-to-Image | | | RSUM |
|---|---|---|---|---|---|---|---|---|
| | | R@1 | R@5 | R@10 | R@1 | R@5 | R@10 | |
| 32 | 2,366.44 | 82.9 | 96.1 | 98.8 | 68.92 | 90.22 | 94.62 | 531.56 |
| 64 | 3,770.57 | 81.8 | 96.2 | 98.4 | 67.88 | 90.36 | 94.76 | 529.40 |
| 128 | 6,640.93 | 81.2 | 96.3 | 98.9 | 68.18 | 90.76 | 94.86 | 530.20 |
| 256 | 12,269.89 | 79.5 | 95.0 | 98.0 | 67.00 | 89.70 | 94.42 | 523.62 |
| 256† | 13,023.27 | 82.9 | 97.1 | 98.6 | 67.36 | 89.72 | 94.40 | 530.08 |

† With MoCo-like momentum update for stable similarity targets

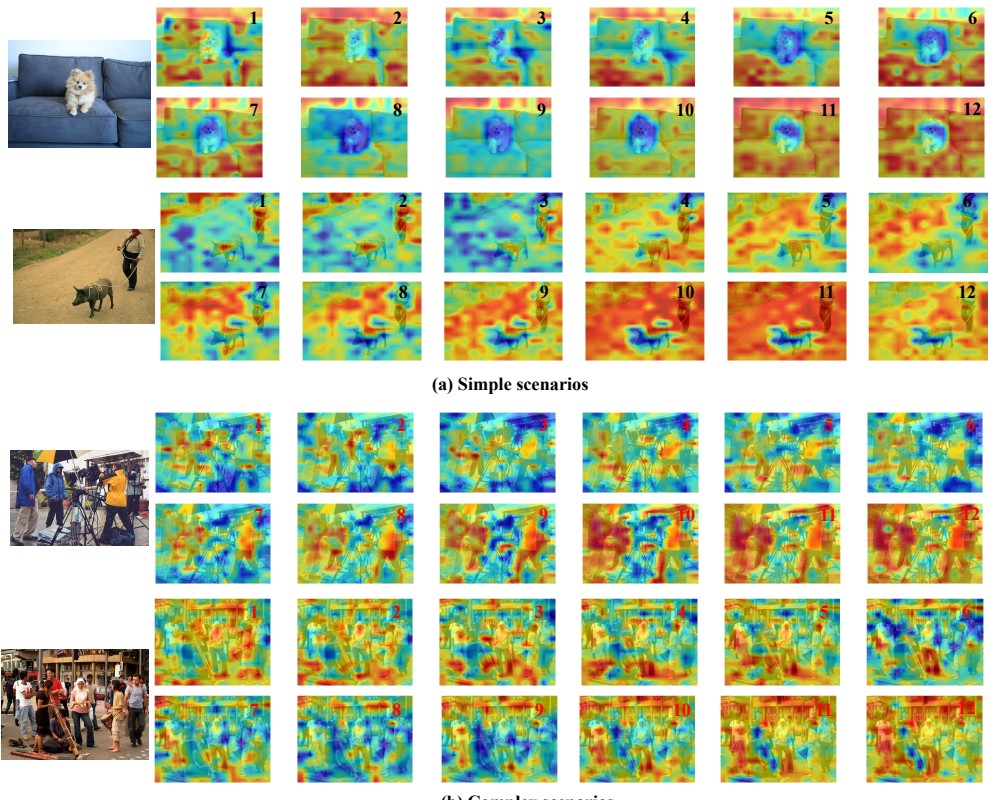

(a) Simple scenarios

(b) Complex scenarios

Figure 13: Semantic convergence trends of simple and complex scenarios in CLIP (ViT-B) architecture.

attention, Layer 8 marks a clear transition where attention consistently converges on semantically salient regions—regardless of input complexity.

### A.14 "SEMANTIC CRYSTALLIZATION" PHENOMENON IN VARIOUS ARCHITECTURES

To further validate the generality of the "semantic crystallization" phenomenon beyond CLIP-based models, we extend our layer-wise attention analysis to two additional vision-language architectures: BLIP Li et al. (2022) and ALBEF Li et al. (2021). As illustrated in Figure 14, we observe a consistent pattern of progressive semantic focusing across both architectures. Taking the caption "A donkey suspended in air after a cart it is attached to over turns" as an example, we analyze the cross-modal attention evolution in BLIP and ALBEF. In early layers (Layers 1–6), attention maps are diffuse and unstable, distributing weights across background and less relevant regions. From Layer 7 onward, both models exhibit a clear convergence toward semantically salient regions (i.e., the donkey and the overturned cart). This transition marks the emergence of structured cross-modal alignment. Notably,

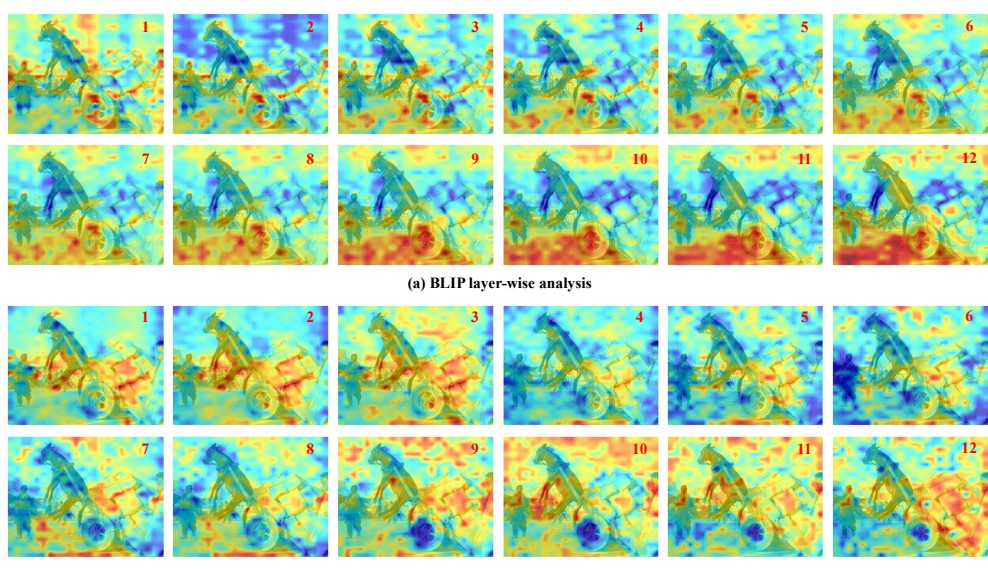

(a) BLIP layer-wise analysis

(b) ALBEF layer-wise analysis

Figure 14: Consistent "semantic crystallization" phenomenon in BLIP and ALBEF.

Table 18: Sensitivity analysis of temperature parameter in SCD.

| $\tau$ | Image-to-Text | | | Text-to-Image | | | RSUM |
|---|---|---|---|---|---|---|---|
| | R@1 | R@5 | R@10 | R@1 | R@5 | R@10 | |
| 0.1 | 75.6 | 92.8 | 97.0 | 62.78 | 86.78 | 92.66 | 507.62 |
| 0.2 | 76.6 | 94.5 | 97.2 | 64.52 | 88.68 | 93.54 | 515.04 |
| 0.3 | 79.2 | 94.9 | 97.1 | 65.54 | 89.30 | 93.72 | 519.76 |
| 0.4 | 80.0 | 95.3 | 97.2 | 66.30 | 89.64 | 94.04 | 522.48 |
| 0.5 | 80.8 | 95.8 | 97.7 | 67.20 | 89.78 | 94.50 | 525.78 |
| 0.6 | 81.8 | 95.6 | 98.1 | 67.90 | 90.22 | 94.54 | 528.16 |
| 0.7 | 81.5 | 95.7 | 98.5 | 68.20 | 90.50 | 94.8 | 529.20 |
| 0.8 | 80.7 | 96.0 | 98.3 | 68.58 | 90.46 | 94.74 | 528.78 |
| 0.9 | 80.9 | 96.0 | 98.3 | 68.44 | 90.46 | 94.72 | 528.82 |
| 1.0 | 81.2 | 96.3 | 98.9 | 68.18 | 90.76 | 94.86 | 530.20 |

while the absolute layer index of semantic crystallization varies slightly across architectures (Layer 8 in CLIP, Layers 7–9 in BLIP and ALBEF), the consistent trend confirms that intermediate layers universally serve as hubs for semantic grounding. These findings strengthen the rationale behind our key-layer pre-alignment strategy and suggest its potential applicability across a broad family of vision-language models.

## A.15 SENSITIVITY ANALYSIS OF TEMPERATURE PARAMETER IN SCD

In our Semantic Consistency Distillation (SCD) mechanism, the temperature parameter $\tau$ plays a critical role in modulating the sharpness of the probability distributions derived from intra-modal and cross-modal similarity matrices. As defined in Equation (3), SCD minimizes the KL-divergence between intra-modal semantic distributions ($F^{\text{i2t}}$, $F^{\text{t2t}}$) and cross-modal predictions ($P^{\text{i2t}}$, $P^{\text{t2i}}$), where these distributions are obtained via softmax normalization scaled by $\tau$:

$$F_j^{\text{i2i}} = \text{softmax}(S_j^{\text{i2i}}/\tau), \quad P_j^{\text{i2t}} = \text{softmax}(S_j^{\text{i2t}}/\tau). \tag{6}$$

A lower $\tau$ value sharpens the distributions, emphasizing high-similarity pairs, while a higher $\tau$ smooths the distributions, allowing broader semantic relationships to be captured. This balancing

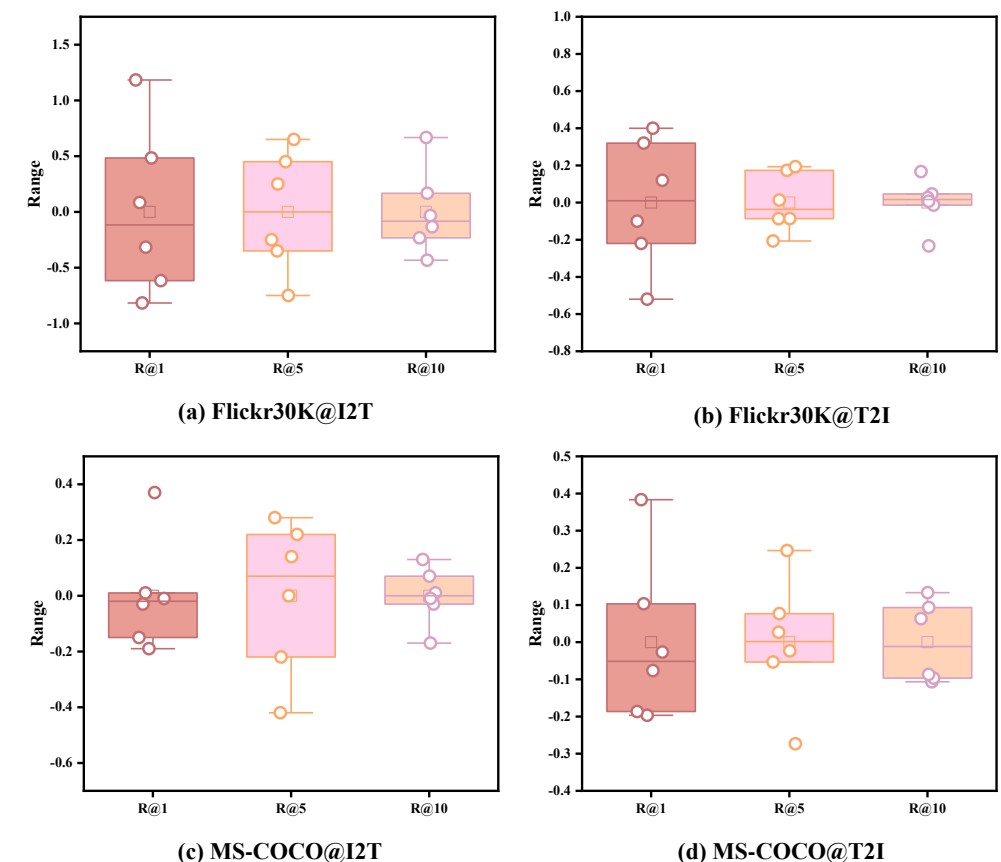

Figure 15: Performance deviation distributions across six random seeds on Flickr30K and MS-COCO datasets. The box plots show the spread of deviations from the mean for Recall@1, 5, and 10 in both image-to-text (I2T) and text-to-image (T2I) retrieval tasks.

act directly impacts the robustness of knowledge distillation and the mitigation of semantic drift. To systematically evaluate the sensitivity of $\tau$, we conducted experiments on Flickr30K with ViT-B/32, varying $\tau$ from 0.1 to 1.0. These results, summarized in Table 18, reveal a clear trend: performance improves consistently as $\tau$ increases, peaking at $\tau = 1.0$ with an RSUM of 530.20. Specifically, Recall@1 for image-to-text retrieval rises from 75.6% ($\tau = 0.1$) to 81.2% ($\tau = 1.0$), while text-to-image Recall@1 increases from 62.78% to 68.18%. This indicates that overly sharp distributions ($\tau < 0.7$) restrict the model's ability to leverage nuanced intra-modal structures, whereas smoother distributions ($\tau \geq 0.7$) facilitate more effective cross-modal alignment by preserving relational semantics across a broader set of samples. Notably, the optimal $\tau = 1.0$ aligns with our default setting, where the temperature is learnable but clamped between 0.1 and 0.99 during training. The superior performance at $\tau = 1.0$ suggests that a balanced, moderately smooth distribution optimally regularizes the model without over-emphasizing specific instances. This finding reinforces the design choice in SCD to avoid extreme temperature values and highlights the importance of calibrating distribution sharpness for semantic consistency.

## A.16 STATISTICAL SIGNIFICANCE AND VARIANCE ANALYSIS

We conducted comprehensive experiments using 6 different random seeds under identical training conditions. For each seed, we evaluated performance on both Flickr30K and MS-COCO datasets, measuring Recall@1, Recall@5, and Recall@10 for image-to-text (I2T) and text-to-image (T2I) retrieval tasks. The deviations from the mean performance for each metric are visualized in Figure 15, which presents box plots summarizing the distribution of these deviations across seeds. The results demonstrate consistently low variance across all metrics and datasets. Specifically, for Flickr30K,

the deviations in R@1 for I2T range from -0.82% to +1.18%, with a median close to zero, indicating stable performance. Similarly, for MS-COCO, the deviations in R@1 for I2T remain within ±0.37%, reinforcing the robustness of our method.

## A.17 LIMITATIONS AND FUTURE RESEARCH DIRECTIONS

While DKP demonstrates that a carefully chosen intermediate layer and a lightweight distillation objective can deliver state-of-the-art retrieval accuracy with a fraction of CLIP's trainable parameters, several caveats remain.

**Manual key-layer selection.** Our empirical study identifies Layer 8 as the "semantic knee" for ViT-B backbones. Extending DKP to very deep or non-ViT encoders (e.g., Swin, ConvNeXt) may require an additional sweep.

**Symmetric-backbone assumption.** All experiments use dual-stream encoders of equal depth (ViT-B/32 and ViT-B/16, 12 layers each). In larger, asymmetric settings (e.g., CLIP ViT-L/14 (24 image layers) paired with a 12-layer text transformer), the two modalities crystallise at different depths. How to choose one or multiple key layers, or whether distinct key layers should be aligned independently, remains unexplored. This becomes an open topic that warrants further in-depth research.

**Compute footprint.** Freezing 86% of the weights reduces gradient memory and optimizer state, but the full backbone is still executed at every step. FLOP savings are therefore modest; extreme resource constraints (e.g., edge devices) may still prefer prompt learning.

**Cluster-level consistency, not instance-level.** Semantic Consistency Distillation enforces distributional similarity, but does not guarantee perfect pair-wise ordering. Failure cases show that visually similar yet semantically distinct images (e.g., "kite" vs. "paraglide") can still co-cluster, suggesting room for finer-grained regularizers.

Despite these limitations, DKP offers a favourable accuracy–speed–size trade-off and provides a principled view of where cross-modal semantics emerge inside large VLMs. Future work could investigate adaptive key-layer selection for asymmetric backbones, integrate DKP with prompt learning, and extend evaluation to more diverse tasks and datasets.

