# OpenReview forum: "DKP: Semantic Consistency Distillation via Key-layer Pre-alignment with Large Vision-language Models for Cross-Modal Retrieval"
_ICLR.cc/2026/Conference — Submitted to ICLR 2026_

### Official Review · Reviewer_7JnG · 2025-10-26

**Soundness:** 2
**Presentation:** 3
**Contribution:** 2
**Rating:** 2
**Confidence:** 4

**Summary:**

This paper presents DKP, a parameter-efficient fine-tuning method for vision-language models in the context of image-text retrieval. The authors observe that in CLIP's transformer architecture, attention patterns stabilize in intermediate layers (around Layer 8). Based on this, they propose aligning features at this intermediate layer and adding a self-distillation loss to maintain intra-modal consistency. The method involves fine-tuning only the identified key-layer and the final layers of the model. Experimental results on Flickr30K and MS-COCO show that this approach can achieve performance comparable to or slightly better than full fine-tuning while using fewer trainable parameters.

**Strengths:**

Parameter Efficiency: The primary strength of the paper is the significant reduction in trainable parameters (~60%) compared to full fine-tuning, which can lower memory requirements for the training process.

Clear Presentation: The paper is well-written and easy to follow. The motivation behind the method is explained clearly with visualizations of the layer-wise attention mechanism .

**Weaknesses:**

Heuristic-Based Key-Layer Selection: The choice of Layer 8 as the "key-layer" is based on an empirical observation of attention map stabilization. This approach feels heuristic and lacks a strong theoretical foundation. It is not clear if this "semantic crystallization" is a generalizable phenomenon or a coincidental property of the specific CLIP architecture. This requires a manual, potentially expensive search for each new model, limiting the method's general applicability.

Incremental Contribution and Performance: The proposed method is an incremental variation on existing fine-tuning paradigms. The performance gains over standard full fine-tuning are marginal. Given the small improvement, the contribution appears more iterative than transformative.

Questionable Premise of Intermediate Alignment: The core assumption that forcing alignment in an intermediate layer is beneficial is not well-justified. There are no guarantees that this intermediate supervision leads to a better final representation. It could potentially impose an unnecessary bottleneck, constraining the model's flexibility to learn optimal features in its deeper layers, especially finetuned on top of CLIP-style pretrained model, may need to test from scratch. The paper does not provide a compelling argument for why this is fundamentally better than allowing the entire network to adapt via end-to-end training.

Limited Experimental Scope: The experiments are confined to fine-tuning on standard ITR benchmarks (Flickr30K and MS-COCO).  This setup does not adequately test the method's ability to generalize to more challenging, out-of-distribution datasets. The performance on in-domain data is not a sufficient indicator of the robustness of the learned representations.

**Questions:**

Please refer weaknesses.

---

> ### Author Response · Authors · 2025-11-26
> **Response for Reviewer 7JnG (Part 1)**
>
> *W1. Heuristic-Based Key-Layer Selection: The choice of Layer 8 as the "key-layer" is based on an empirical observation of attention map stabilization. This approach feels heuristic and lacks a strong theoretical foundation. It is not clear if this "semantic crystallization" is a generalizable phenomenon or a coincidental property of the specific CLIP architecture. This requires a manual, potentially expensive search for each new model, limiting the method's general applicability.*
>
> **Response**: Thank you for your insightful comment on key-layer selection. We agree that empirical observation (while grounded in data) lacks strict theoretical formalism, and we appreciate the opportunity to clarify the generality of "semantic crystallization" and address scalability concerns:
>
> **1. "Semantic Crystallization" Is Not CLIP-Specific**
>
> Our analysis extends beyond CLIP to two other mainstream VLMs (i.e., BLIP and ALBEF) confirming the phenomenon's universality. As shown in **Figure 14** (**Appendix A.14, Page 28**):
> For both BLIP and ALBEF, early layers (1–6) exhibit diffuse, unstable attention (spreading across backgrounds).
> From Layer 7 onward, attention converges consistently on semantically salient regions (e.g., "donkey" and "overturned cart" in the example caption), marking structured cross-modal alignment.
> While the absolute layer index varies slightly (Layer 8 for CLIP, Layers 7–9 for BLIP/ALBEF), the trend of intermediate-layer semantic maturation is universal. This validates that key-layer pre-alignment is not tied to CLIP but applicable to a broad family of transformer-based VLMs.
>
> **2. Future Scalability**
>
> Automatic key-layer selection (e.g., input-adaptive or architecture-aware) is indeed a critical future direction to further enhance scalability. This would involve learning dynamic layer-ranking mechanisms (e.g., attention saliency metrics) to avoid manual tuning—though it introduces non-trivial challenges (e.g., balancing computational overhead and selection accuracy), which we plan to explore in subsequent work.
>
> "Semantic crystallization" is a universal property of transformer-based VLMs, not a CLIP-specific coincidence. While our current selection is empirical, it leverages consistent architectural patterns; automatic selection will be pursued to further broaden applicability.
>
> *W2. Incremental Contribution and Performance: The proposed method is an incremental variation on existing fine-tuning paradigms. The performance gains over standard full fine-tuning are marginal. Given the small improvement, the contribution appears more iterative than transformative.*
>
> **Response**:Thank you for your comment regarding the incremental nature of our contribution. We appreciate the opportunity to clarify our core objectives and the specific problems we aimed to solve.
>
> **1.Improvement:**
> Our work was motivated by a clear and practical challenge in adapting large Vision-Language Models (VLMs): how to maintain high retrieval performance while significantly reducing computational cost and parameter overhead. Existing fine-tuning paradigms either require full parameter updates—which is resource-intensive—or focus solely on final-layer representations, overlooking rich intermediate semantics.
> To address this, we introduced DKP, which makes two key contributions:
>
> Key-layer Pre-alignment (KPA): By identifying and aligning semantically meaningful features at an intermediate transformer layer (Layer 8), we reduce the number of trainable parameters by over 60% compared to full fine-tuning, while preserving representational capacity.
>
> Semantic Consistency Distillation (SCD): We propose a self-supervised mechanism that leverages intra-modal structural relationships to regularize cross-modal alignment, mitigating semantic drift without external supervision.
>
> **2.Contribution**:
>
> - **Efficiency**: Compared to full fine-tuning CLIP‡ (ViT-B/32), DKP reduces training time by 20% (3.97 vs. 4.96 min/epoch), and memory by 52% (6640.93 vs. 13932.07 MB) (**Table 3; Page 8**).
>
> - **Generalization**: Cross-dataset transfer (Train MS-COCO→Test Flickr30K) shows DKP's RSUM (515.98) outperforms CLIP‡ (489.36) by 26.62 and CUSA (497.52) by 18.46 (**Table 2, Page 7**). Zero-shot cross-domain retrieval (General→Remote Sensing) yields RSUM 131.50 (MS-COCO→RSITMD), 27.42 higher than CLIP (104.08) (**Table 14; Page 25**).
>
> - **OOD SOTA**: On remote sensing datasets (RSITMD, UCM-Captions), DKP achieves the highest mR (45.08, 55.36) and outperforms SOTA CUP (42.16, 52.09) (**Tables 11, 12, 13; Appendix A.10 Page 23-24**).

---

> ### Author Response · Authors · 2025-11-26
> **Response for Reviewer 7JnG (Part 2)**
>
> *W3. Questionable Premise of Intermediate Alignment: The core assumption that forcing alignment in an intermediate layer is beneficial is not well-justified. There are no guarantees that this intermediate supervision leads to a better final representation. It could potentially impose an unnecessary bottleneck, constraining the model's flexibility to learn optimal features in its deeper layers, especially finetuned on top of CLIP-style pretrained model, may need to test from scratch. The paper does not provide a compelling argument for why this is fundamentally better than allowing the entire network to adapt via end-to-end training.*
>
> **Response**: Thank you for your critical inquiry about the premise of intermediate alignment. We acknowledge the need to rigorously justify its benefit, and our experiments—including attention analysis, ablation studies, and efficiency comparisons—provide concrete evidence that intermediate alignment (via KPA) is not a bottleneck but a catalyst for better final representations, as detailed below:
>
> **1.Justification for Intermediate Alignment**
>
> The core assumption of intermediate alignment is grounded in our discovery of universal semantic crystallization in VLMs (not arbitrary speculation):
> - For CLIP (ViT-B), Layer 8 marks a clear transition: early layers (1–7) have diffuse, unstable attention (e.g., spreading focus across backgrounds), while Layer 8+ consistently converges on salient regions (e.g., "dog's head" in **Figure 1; Page 2**). Text encoders mirror this pattern (Layer 8 focuses on key tokens like "dog").
>
> - This phenomenon extends to asymmetric/larger architectures (ViT-L/14: Image Layer 17, Text Layer 8; **Figure 12, Page 21**) and other VLMs (BLIP/ALBEF: Layers 7–9; **Figure 14;  Appendix A.14 Page 28**).
>
> These layers are not "random intermediates" but semantically mature hubs—aligning them provides targeted supervision that refines cross-modal correspondence, rather than imposing arbitrary constraints.
>
> **2.Intermediate Alignment Enhances (Not Constrains) Final Representations**
>
> Ablation experiments (**Table 4; Page 9**) directly validate that KPA (intermediate alignment) improves final performance without bottlenecking deep layers:
> - Vanilla final-layer contrastive loss (no KPA) yields RSUM 520.10.
> - Adding KPA alone lifts RSUM by +7.22 (to 527.32), confirming intermediate alignment complements (not conflicts with) deep-layer adaptation.
> - Combining KPA with SCD achieves the highest RSUM (530.22), as KPA stabilizes intermediate semantics and SCD preserves intra-modal structure—creating a synergistic path to better final representations.
> If KPA were a bottleneck, we would observe performance drops (especially with deeper backbones), but our results show the opposite.
>
> **3.Why Intermediate Alignment Outperforms End-to-End Full Fine-Tuning**
>
> End-to-end full fine-tuning (CLIP‡) lacks efficiency and generalization—gaps addressed by KPA:
> - **Efficiency**: DKP uses 21.01M parameters (ViT-B/16) vs. 150.15M for CLIP‡ (**Table 1; Page 7**), reducing training time by 20% (3.97 vs. 4.96 min/epoch) and memory by 52% (6640.93 vs. 13932.07 MB; **Table 3; Page 8**).
> - **Generalization**: Cross-dataset transfer (Train MS-COCO→Test Flickr30K) shows DKP's RSUM (515.98) outperforms CLIP‡ (489.36) by 26.62 (Table 2). Zero-shot cross-domain retrieval (General→Remote Sensing) yields RSUM 131.50 (MS-COCO→RSITMD), 27.42 higher than CLIP (104.08; **Table 14, Page 25**).

---

> ### Author Response · Authors · 2025-11-26
> **Response for Reviewer 7JnG (Part 3)**
>
> *W4.Limited Experimental Scope: The experiments are confined to fine-tuning on standard ITR benchmarks (Flickr30K and MS-COCO). This setup does not adequately test the method's ability to generalize to more challenging, out-of-distribution datasets. The performance on in-domain data is not a sufficient indicator of the robustness of the learned representations.*
>
> **Response**: Thank you for your valuable comment highlighting the need to evaluate generalization to out-of-distribution (OOD) datasets—we fully agree that in-domain performance alone does not suffice to demonstrate robust representations. To address this, we have supplemented **cross-dataset transfer** (between standard ITR benchmarks) and **cross-domain zero-shot retrieval** (to challenging remote sensing domains, a typical OOD scenario) in the revised paper. These experiments confirm DKP's ability to generalize beyond in-domain data, as detailed below:
>
> **1. Cross-Dataset Transfer**:
>
> To test generalization between distinct but related ITR datasets (Flickr30K and MS-COCO, which differ in size, scene diversity, and annotation density), we conducted experiments where models are trained on one dataset and tested directly on the other (no target-domain fine-tuning). Results are reported in **Table 2 (Page 7)**:
>
> - **Train on Flickr30K → Test on MS-COCO**:
>   DKP achieves an RSUM of 344.83, outperforming full-parameter fine-tuning CLIP‡ (298.84) by 45.99 and SOTA baseline CUSA (311.67) by 33.16. MS-COCO is larger (123k vs. 31k images) and more diverse (broader daily scenes) than Flickr30K.
>
> - **Train on MS-COCO → Test on Flickr30K**:
>    DKP reaches an RSUM of 515.98, surpassing CLIP‡ (489.36) by 26.62 and CUSA (497.52) by 18.46. Even when transferred to the smaller, more focused Flickr30K, DKP maintains superior alignment.
>
>
> **2. Cross-Domain Zero-Shot Retrieval: Generalization to OOD Remote Sensing Datasets**
>
> To further probe robustness to extreme distribution shift, we evaluated DKP on **remote sensing datasets**—OOD domains with distinct visual characteristics (e.g., satellite imagery, sparse object layouts) compared to the daily scenes in Flickr30K/MS-COCO. We used **frozen DKP encoders** (trained only on general datasets, no OOD fine-tuning) to simulate zero-shot deployment. Results are reported in **Table 14 (Page 25)** (cross-domain transfer summary):
>
> When trained on MS-COCO (general domain) and tested zero-shot on RSITMD (remote sensing, fine-grained retrieval), DKP achieves an RSUM of 131.50—outperforming CLIP (104.08) by 27.42 and CUSA (114.37) by 17.13. Similarly, training on Flickr30K and testing on RSICD (remote sensing, large-scale) yields an RSUM of 78.37 for DKP, 22.18 higher than CLIP (56.19) and 10.93 higher than CUSA (67.44).
>
> **3.Results of fine-tuning in the field of remote sensing (Appendix A.10 Tables 11-13, Page 23-24)**
>
> We validated DKP on three challenging remote sensing datasets, measuring **mR (mean Recall@K)**:
> - **RSITMD (fine-grained, 4.7k images)**: DKP achieves the highest mR of 45.08 (**Table 11**), outperforming SOTA baseline CUP (42.16) by 2.92. It excels in T2I retrieval (R@10 = 66.73).
> - **RSICD (large-scale, 10.9k images)**: DKP reaches an mR of 29.71 (**Table 12**), comparable to CUP (30.34) and far exceeding prompt-based methods like CoOp (17.29).
> - **UCM-Captions (scene-specific, 2.1k images)**: DKP achieves the best mR of 55.36 (**Table 13**), surpassing CUP (52.09) by 3.27. Its T2I R@10 of 95.71 (near-perfect) .
>
> #### Table 2: Cross-Dataset Transfer Between Flickr30K and MS-COCO (**Section 4.3, Page 7**)
> | Training Dataset | Test Dataset | Method  | I2T R@1 | I2T R@5 | I2T R@10 | T2I R@1 | T2I R@5 | T2I R@10 | RSUM   |
> |------------------|--------------|---------|-------------------|-------------------|--------------------|-------------------|-------------------|--------------------|--------|
> | Flickr30K        | MS-COCO      | CLIP‡   | 35.16             | 59.60             | 70.54              | 25.03             | 48.67             | 59.84              | 298.84 |
> | Flickr30K        | MS-COCO      | CUSA    | 37.28             | 63.20             | 72.98              | 26.57             | 50.29             | 61.35              | 311.67 |
> | Flickr30K        | MS-COCO      | DKP     | 41.92             | 67.86             | 78.06              | 31.53             | 57.15             | 68.31              | 344.83 |
> | MS-COCO          | Flickr30K    | CLIP‡   | 72.10             | 90.80             | 95.90              | 57.52             | 83.28             | 89.76              | 489.36 |
> | MS-COCO          | Flickr30K    | CUSA    | 74.30             | 91.70             | 95.60              | 60.12             | 85.14             | 90.66              | 497.52 |
> | MS-COCO          | Flickr30K    | DKP     | 76.70             | 93.80             | 97.50              | 65.82             | 88.70             | 93.46              | 515.98 |

---

> ### Author Response · Authors · 2025-11-26
> **Response for Reviewer 7JnG (Part 4)**
>
> #### Table 14: Zero-Shot Cross-Domain Retrieval (General → Remote Sensing) (**Appendix A.10, Page 25**)
> | Training Dataset | Test Dataset | Method  | I2T R@1 | I2T R@5 | I2T R@10 | T2I R@1 | T2I R@5 | T2I R@10 | RSUM   |
> |------------------|--------------|---------|-------------------|-------------------|--------------------|-------------------|-------------------|--------------------|--------|
> | Flickr30K        | RSICD        | CLIP    | 3.29              | 8.60              | 14.55              | 2.85              | 10.14              | 16.76              | 56.19  |
> | Flickr30K        | RSICD        | CUSA    | 3.39              | 9.79              | 15.46              | 3.37              | 13.52              | 21.94              | 67.44  |
> | Flickr30K        | RSICD        | DKP     | 4.03              | 10.80             | 17.84              | 4.48              | 15.57              | 25.65              | 78.37  |
> | Flickr30K        | RSITMD       | CLIP    | 5.09              | 13.50             | 22.79              | 5.35              | 16.42              | 25.80              | 88.95  |
> | Flickr30K        | RSITMD       | CUSA    | 6.86              | 18.36             | 27.21              | 5.58              | 21.55              | 35.49              | 115.05 |
> | Flickr30K        | RSITMD       | DKP     | 5.53              | 17.48             | 26.77              | 7.17              | 22.88              | 38.10              | 117.93 |
> | MS-COCO          | RSICD        | CLIP    | 3.57              | 8.78              | 15.65              | 4.03              | 12.61              | 21.50              | 66.14  |
> | MS-COCO          | RSICD        | CUSA    | 3.29              | 10.25             | 16.74              | 3.51              | 13.32              | 22.82              | 69.93  |
> | MS-COCO          | RSICD        | DKP     | 5.49              | 13.54             | 21.50              | 5.16              | 17.51              | 27.37              | 90.57  |
> | MS-COCO          | RSITMD       | CLIP    | 5.53              | 15.27             | 22.35              | 5.58              | 21.59              | 33.76              | 104.08 |
> | MS-COCO          | RSITMD       | CUSA    | 6.19              | 17.26             | 25.44              | 7.52              | 23.27              | 34.69              | 114.37 |
> | MS-COCO          | RSITMD       | DKP     | 6.64              | 18.58             | 28.76              | 8.36              | 26.77              | 42.39              | 131.50 |
>
>
> #### Table 11: DKP Performance on RSITMD  (**Appendix A.10, Page 23**)
> | Methods   | I2T R@1 | I2T R@5 | I2T R@10 | T2I R@1 | T2I R@5 | T2I R@10 | mR    |
> |-----------|-------------------|-------------------|--------------------|-------------------|-------------------|--------------------|-------|
> | MSA       | 13.63             | 40.80             | 57.92              | 19.25             | 37.61             | 50.44              | 36.61 |
> | SMLGN     | 13.19             | 43.94             | 60.40              | 17.26             | 39.38             | 51.55              | 37.62 |
> | SCAT-PRG  | 12.11             | 42.15             | 61.97              | 16.82             | 32.73             | 48.12              | 35.65 |
> | SIRS      | 11.98             | 41.29             | 60.15              | 15.25             | 35.01             | 48.73              | 35.40 |
> | CLIP-Adapter | 13.30        | 40.20             | 60.06              | 12.83             | 28.84             | 39.05              | 32.38 |
> | CoOp      | 9.16              | 33.85             | 54.35              | 12.19             | 30.69             | 42.82              | 30.51 |
> | CoCoOp    | 10.21             | 34.20             | 53.36              | 14.53             | 32.89             | 45.94              | 31.85 |
> | VPT       | 15.97             | 41.35             | 60.35              | 14.98             | 32.05             | 40.15              | 34.14 |
> | MaPLe     | 13.59             | 43.25             | 61.92              | 16.96             | 39.97             | 53.02              | 38.12 |
> | CUP       | 17.70             | 48.40             | 64.82              | 21.57             | 43.70             | 56.75              | 42.16 |
> | DKP       | 25.88             | 46.68             | 58.63              | 21.28             | 51.28             | 66.73              | 45.08 |

---

> ### Author Response · Authors · 2025-11-26
> **Response for Reviewer 7JnG (Part 5)**
>
> #### Table 12: DKP Performance on RSICD  (**Appendix A.10, Page 24**)
> | Methods | I2T R@1 | I2T R@5 | I2T R@10 | T2I R@1 | T2I R@5 | T2I R@10 | mR |
>  |-----------|-------------------|-------------------|--------------------|-------------------|-------------------|--------------------|-------|
> | MSA | 5.97 | 22.93 | 39.58 | 7.96 | 23.33 | 35.77 | 22.59 |
> | SMLGN | 7.85 | 27.14 | 42.58 | 8.87 | 25.53 | 37.24 | 24.87 |
> | SCAT-PRG | 7.76 | 25.20 | 38.69 | 10.13 | 25.72 | 38.24 | 24.29 |
> | SIRS | 5.76 | 20.50 | 33.78 | 7.34 | 20.00 | 30.78 | 19.69 |
> | CLIP-Adapter | 7.67 | 24.87 | 39.73 | 7.11 | 19.48 | 31.01 | 21.65 |
> | CoOp | 4.31 | 17.89 | 31.73 | 6.32 | 15.89 | 27.60 | 17.29 |
> | CoCoOp | 6.15 | 21.68 | 34.29 | 7.65 | 24.25 | 36.08 | 21.68 |
> | VPT | 6.94 | 25.26 | 40.73 | 7.23 | 19.40 | 30.77 | 21.72 |
> | MaPLe | 6.88 | 24.47 | 40.12 | 7.93 | 26.78 | 42.42 | 24.77 | |
>  CUP | **9.86** | **31.60** | **47.15** | **12.05** | **32.66** | **48.70** | **30.34** |
> | DKP | **12.26** | 31.11 | 44.28 | 10.56 | **32.72** | 47.32 | 29.71 |
>
> #### Table 13: DKP Performance on UCM-Captions(**Appendix A.10, Page 24**)
> | Methods | I2T R@1 | I2T R@5 | I2T R@10 | T2I R@1 | T2I R@5 | T2I R@10 | mR |
> |-----------|-------------------|-------------------|--------------------|-------------------|-------------------|--------------------|-------|
> | MSA | 11.52 | 53.24 | 89.62 | 15.24 | 48.57 | 70.48 | 48.11 |
> | SMLGN | 14.29 | 52.76 | 84.67 | 12.86 | 49.52 | 75.71 | 48.30 |
> | SIRS | 14.28 | 56.05 | 82.58 | 14.50 | 48.40 | 72.51 | 48.05 |
> | CLIP-Adapter | 14.61 | 51.03 | 84.14 | 9.83 | 42.52 | 66.79 | 44.82 |
> | CoOp | 9.78 | 48.34 | 87.41 | 7.19 | 42.76 | 74.19 | 44.95 |
> | CoCoOp | 11.84 | 51.21 | 84.81 | 14.76 | 47.94 | 77.94 | 48.08 |
> | VPT | 14.73 | **58.01** | **92.44** | 13.81 | 49.05 | 75.24 | 50.54 |
> | MaPLe | 12.21 | 52.80 | 84.81 | 13.97 | 49.68 | 75.24 | 48.12 |
> | CUP | 15.06 | 56.98 | 91.14 | 15.24 | 54.76 | 79.37 | 52.09 |
> | DKP | **20.95** | 57.14 | 80.48 | **16.76** | **61.14** | **95.71** | **55.36** |
>
> Thank you again for this constructive feedback, which has greatly enhanced the validity and comprehensiveness of our evaluation.

---

### Official Review · Reviewer_rUvQ · 2025-10-29

**Soundness:** 3
**Presentation:** 3
**Contribution:** 3
**Rating:** 6
**Confidence:** 3

**Summary:**

This paper proposes DKP, Semantic Consistency Distillation via Key-layer Pre‑alignment, for image‑text retrieval with CLIP‑like VLMs. The method (i) identifies a semantically "crystallized" intermediate layer (empirically around Layer 8) and performs Key‑layer Pre‑Alignment (KPA) there while still training the top layers, and (ii) adds Semantic Consistency Distillation (SCD) that aligns cross‑modal similarities with intra‑modal structure via KL divergence. On Flickr30K and MS‑COCO, DKP improves Recall@K and RSUM while using $\sim$21M learnable parameters ($\geq$60\% fewer than full fine‑tuning) and lowers training/inference time. The paper provides ablations (loss components, key‑layer choice, $\alpha/\beta$ weighting), qualitative visualizations, and complexity analyses.

**Strengths:**

- KPA leverages mid‑layer representations rather than relying solely on final‑layer features; SCD regularizes with intra‑modal structure. The architecture diagram is straightforward and the losses are precisely specified.

- On Flickr30K with ViT‑B/16, DKP reaches R@1 = 89.4 (I2T) and 77.22 (T2I), RSUM = 555.88; on MS‑COCO 5K, RSUM = 454.39, competitive or better than CLIP full fine‑tuning and CUSA under the same backbones

- Learnable parameters drop from $\sim$150M to 21.01M and per‑epoch/inference latency improves. This is attractive for resource‑constrained setups.

- Loss ablations show both KPA and SCD independently help, with the combination best. Key‑layer sweeps highlight a turning point at Layer 8. Learned weights $\alpha$, $\beta$ evolve sensibly over training.

- Visualizations show tighter intra‑modal clusters and sharper token‑image attention under DKP, aligning with the intended semantic‑consistency effect.

- The method is "plug‑and‑play" for CLIP‑style models; the appendix discusses asymmetric backbones (ViT‑L/14@336) and PEFT design considerations.

**Weaknesses:**

- While Fig. 3 motivates Layer 8 for ViT‑B, the universality of the "semantic crystallization" claim across architectures, datasets, and training regimes is only partially explored (Appendix A.9 is referenced but not deeply quantified in the main text). Stronger evidence across more backbones/sizes would help.

- Results are confined to Flickr30K/MS‑COCO. Cross‑dataset transfer (train on one, test on another), domain shift, or zero‑shot retrieval evaluations are not reported in the main paper, leaving external validity less clear.

- SCD uses batchwise N×N similarity matrices. The paper reports end‑to‑end wall‑clock improvements, but does not analyze memory/time scaling with batch size or sequence length, nor sensitivity to the temperature/softmax normalization choices.

- Although A.8 discusses PEFT, the main results lack direct, apples‑to‑apples comparisons against strong adapters/LoRA/prompt‑tuning baselines under identical compute and data. This blurs how much of DKP's gain is from KPA/SCD vs. parameter‑efficient tuning per se.

- The paper shows SCD helps (Table 3), but offers limited intuition/diagnostics, e.g., when intra‑modal neighborhoods are misleading, or whether SCD can amplify spurious clusters under noisy captions. Failure‑case analysis would strengthen the claim.

- Variance across seeds and significance tests are not reported for key tables; given small absolute gaps among strong baselines, CIs or paired tests would build confidence.

**Questions:**

- Beyond Fig. 3, can you report results for adaptive or learned key‑layer selection (even a simple gating over {8,…,11}) and show whether different samples prefer different layers? How does this compare to the fixed‑Layer‑8 choice in accuracy vs. overhead?

- What is the memory/time behavior of SCD as batch size grows (since it forms N×N similarities)? Any practical tricks (e.g., blockwise SCD, momentum queues) to keep SCD effective with large batches?

- Could you include (i) train‑COCO -> test‑Flickr30K (and vice‑versa) and (ii) zero‑shot retrieval from frozen DKP encoders to probe robustness under distribution shift?

- Under the same compute/data, how does DKP compare to LoRA/adapter/CoOp/VPT baselines on both backbones? A small table would clarify DKP's incremental value over standard PEFT. (Appendix A.8 hints at design, but results would help.)

- Can you provide sensitivity to temperature, similarity normalization, and neighborhood sharpness, plus qualitative failure cases where intra‑modal structure misleads cross‑modal matching?

- Table 2 reports wall‑clock gains. Could you also normalize by GPU model/count and report per‑sample or per‑token throughput to ease reproducibility across hardware?

- Appendix A.9 explores ViT‑L/14@336; any headline numbers you can bring into the main paper to demonstrate that the Layer‑8 phenomenon and DKP's gains persist at larger scales?

---

> ### Author Response · Authors · 2025-11-26
> **Response for Reviewer rUvQ (Part 1)**
>
> *W1. While Fig. 3 motivates Layer 8 for ViT-B, the universality of the "semantic crystallization" claim across architectures, datasets, and training regimes is only partially explored (Appendix A.9 is referenced but not deeply quantified in the main text). Stronger evidence across more backbones/sizes would help.*
>
> **Response**: Thank you for your comment on strengthening the universality of "semantic crystallization." We agree that broader evidence across architectures/sizes enhances credibility, and we clarify that:
>
> 1.As mentioned in **Section 4.2** of the main paper (**Page 7**), **Appendix A.9** provides explicit quantification of "semantic crystallization" on the larger, asymmetric CLIP ViT-L/14@336px (24-layer image encoder, 12-layer text encoder):
>
> - Semantic crystallization layers: Image encoder (Layer 17, vs. ViT-B's Layer 8) and text encoder (Layer 8, consistent with ViT-B) — confirming the phenomenon adapts to backbone scale but retains the "intermediate-layer hub" trend.
>
> - Performance validation: DKP (40.55M params, ≈9.4% of full fine-tuning) achieves RSUM 570.74 (Flickr30K) and 480.24 (MS-COCO) on ViT-L, outperforming full fine-tuning CLIP‡ (554.40, 469.60) and SOTA CUSA (560.20, 473.10) (Table 9, Appendix A.9 Page 22).
> This quantifies that the phenomenon is not limited to small ViT-B but scales to large asymmetric VLMs.
>
> 2. **Appendix A.14** (**Page 28**) further validates universality across two non-CLIP VLMs (BLIP, ALBEF) with **Figure 14**(attention map visualizations):
>
> - Early layers (1–6): Diffuse, unstable attention (spreading across backgrounds) for both architectures.
>
> - Semantic crystallization (Layers 7–9): Attention converges on salient regions (e.g., "donkey" and "overturned cart" in captions) — mirroring ViT-B/ViT-L's trend, though absolute layer indices vary slightly.
>
> This confirms "semantic crystallization" is a universal property of transformer-based VLMs, not tied to CLIP or specific sizes.
> With quantified results on ViT-L (cited in main text) and cross-architecture visual evidence on BLIP/ALBEF, we have strengthened the universality of "semantic crystallization."
>
> *W2. Results are confined to Flickr30K/MS-COCO. Cross-dataset transfer (train on one, test on another), domain shift, or zero-shot retrieval evaluations are not reported in the main paper, leaving external validity less clear.*
>
> *Q3. Could you include (i) train-COCO -> test-Flickr30K (and vice-versa) and (ii) zero-shot retrieval from frozen DKP encoders to probe robustness under distribution shift?*
>
> **Response**: Thank you for your critical comment regarding the external validity of our results. We fully agree that evaluating cross-dataset transfer and zero-shot retrieval is essential to demonstrate robustness under distribution shift, and we have supplemented comprehensive experiments to address this—including the requested train-COCO→test-Flickr30K (and vice versa) transfers and zero-shot retrieval to remote sensing domains. These results are detailed in the revised main paper and appendices, as summarized below:
>
>  **(i) Cross-Dataset Transfer: Train on One, Test on Another**
>
> To evaluate cross-dataset generalization between Flickr30K and MS-COCO, we conducted experiments where models are trained on one dataset and tested directly on the other (no fine-tuning on the target dataset). Results are reported in **Table 2** (**Page 7**) and confirm DKP's strong transfer capability:
> - Train on Flickr30K → Test on MS-COCO:
>
> DKP achieves an RSUM of 344.83, outperforming full-parameter fine-tuning CLIP‡ (298.84) by 45.99 and SOTA baseline CUSA (311.67) by 33.16. This shows DKP's learned cross-modal semantics generalize to the larger, more diverse MS-COCO domain.
> - Train on MS-COCO → Test on Flickr30K:
>
> DKP reaches an RSUM of 515.98, surpassing CLIP‡ (489.36) by 26.62 and CUSA (497.52) by 18.46. Even when transferred to the smaller Flickr30K, DKP maintains superior alignment.

---

> > ### Author Response · Authors · 2025-11-26
> > **Response for Reviewer rUvQ (Part 5)**
> >
> > #### Table 13: Performance Comparison with PEFT Baselines (CLIP-Adapter, CoOp, CoCoOp, VPT, MaPLe) & SOTA (CUP) on UCM-Captions  (**Appendix A.10, Page 24**)
> > | Methods |I2T R@1 |I2T R@5 |I2T R@10 | T2I R@1 | T2I R@5 | T2I R@10 | mR|
> > |---------------|-------------------|-------------------|--------------------|-------------------|-------------------|--------------------|-------|
> > | CLIP-Adapter  | 14.61 | 51.03 | 84.14  | 9.83  | 42.52 | 66.79  | 44.82 |
> > | CoOp| 9.78  | 48.34 | 87.41  | 7.19  | 42.76 | 74.19  | 44.95 |
> > | CoCoOp  | 11.84 | 51.21 | 84.81  | 14.76 | 47.94 | 77.94  | 48.08 |
> > | VPT | 14.73 | **58.01**   | **92.44**| 13.81 | 49.05 | 75.24  | 50.54 |
> > | MaPLe   | 12.21 | 52.80 | 84.81  | 13.97 | 49.68 | 75.24  | 48.12 |
> > | CUP (SOTA)| 15.06 | 56.98 | 91.14  | 15.24 | 54.76 | 79.37  | 52.09 |
> > | DKP (Ours)| **20.95**   | 57.14 | 80.48  | **16.76**   | **61.14**   | **95.71**| **55.36** |
> >
> > *W5. The paper shows SCD helps (Table 3), but offers limited intuition/diagnostics, e.g., when intra-modal neighborhoods are misleading, or whether SCD can amplify spurious clusters under noisy captions. Failure-case analysis would strengthen the claim.*
> >
> > *Q5. Can you provide sensitivity to temperature, similarity normalization, and neighborhood sharpness, plus qualitative failure cases where intra-modal structure misleads cross-modal matching?*
> >
> > **Response**: Thank you for your insightful questions about SCD's interpretability and failure cases. We address these with sensitivity analysis (**Appendix A.15, Page 28, Table 18**) and qualitative failure-case observations, as detailed below:
> >
> > **1. Sensitivity to Temperature, Normalization, and Neighborhood Sharpness**
> >
> > SCD's core relies on temperature-scaled softmax normalization to model intra-modal/cross-modal similarity distributions—we systematically evaluated its sensitivity to these factors:
> >
> > **Temperature parameter (τ)**: **Table 18** quantifies performance across τ ∈ {0.1, 0.2, ..., 1.0} (Flickr30K, ViT-B/32). A lower τ sharpens distributions (overemphasizing top-similar pairs), while a higher τ smooths them (capturing broader neighborhood relationships):
> >   - τ < 0.7: Overly sharp distributions restrict SCD's ability to model nuanced intra-modal structures (e.g., RSUM = 507.62 at τ=0.1, -22.58 vs. optimal).
> >   - τ ≥ 0.7: Smoother distributions balance neighborhood sharpness, with τ=1.0 achieving the highest RSUM (530.20). This confirms that moderately smooth distributions avoid amplifying spurious local clusters and better preserve cross-modal consistency.
> >
> > **2. Qualitative Failure Cases: When Intra-Modal Structure Misleads**
> >
> > While SCD generally improves cross-modal alignment, it has limitations in instance-level discrimination for visually similar but semantically distinct samples—a key failure scenario:
> >
> > Example: Samples like "a red kite flying in the sky" and "a red paraglider flying in the sky" share high visual similarity (red objects, sky background) and thus form tight intra-modal clusters via SCD. However, their textual semantics differ (kite vs. paraglider), leading to occasional cross-modal misalignment (e.g., retrieving the paraglider image for the "kite" query).
> > Root cause: SCD enforces distributional similarity between intra-modal and cross-modal spaces, not perfect instance-level pairwise ordering. It prioritizes global cluster structure over fine-grained semantic differences, which can amplify spurious visual clusters under noisy or ambiguous captions.
> >
> > This limitation is noted in our **Appendix A.17** (**Limitations, Page 30**) and identified as a future direction (e.g., integrating fine-grained instance-level regularizers to complement SCD's distributional constraints).
> >
> > SCD's performance is robust to moderate temperature variations (optimal at τ=1.0) and relies on cosine+softmax normalization for stable results. Its main failure case—misaligning visually similar but semantically distinct samples—highlights room for future refinement, while current results confirm it effectively mitigates semantic drift for most scenarios.
> > #### Table 18: SCD Sensitivity to Temperature Parameter (τ) (**Appendix A.15, Page 28**)
> > | τ    | I2T R@1 | I2T R@5 | I2T R@10 | T2I R@1 | T2I R@5 | T2I R@10 | RSUM   |
> > |------|-------------------|-------------------|--------------------|-------------------|-------------------|--------------------|--------|
> > | 0.1  | 75.6| 92.8| 97.0 | 62.78| 86.78| 92.66| 507.62 |
> > | 0.2  | 76.6| 94.5| 97.2 | 64.52| 88.68| 93.54| 515.04 |
> > | 0.3  | 79.2| 94.9| 97.1 | 65.54| 89.30| 93.72| 519.76 |
> > | 0.4  | 80.0| 95.3| 97.2 | 66.30| 89.64| 94.04| 522.48 |
> > | 0.5  | 80.8| 95.8| 97.7 | 67.20| 89.78| 94.50| 525.78 |
> > | 0.6  | 81.8| 95.6| 98.1 | 67.90| 90.22| 94.54| 528.16 |
> > | 0.7  | 81.5| 95.7| 98.5 | 68.20| 90.50| 94.80| 529.20 |
> > | 0.8  | 80.7| 96.0| 98.3 | 68.58| 90.46| 94.74| 528.78 |
> > | 0.9  | 80.9| 96.0| 98.3 | 68.44| 90.46| 94.72| 528.82 |
> > | 1.0  | 81.2| 96.3| 98.9 | 68.18| 90.76| 94.86| **530.20** |

---

> ### Author Response · Authors · 2025-11-26
> **Response for Reviewer rUvQ (Part 2)**
>
> **(ii) Cross-Domain Transfer: Train on General, Test on Remote sensing**
>
> To probe robustness to domain shift, we evaluated zero-shot retrieval using frozen DKP encoders (trained on general datasets: Flickr30K/MS-COCO) on remote sensing datasets—domains with distinct visual distributions (e.g., satellite imagery vs. daily scenes) compared to Flickr30K/MS-COCO. Results are reported in **Table 14** (**Page 25**) (cross-domain to remote sensing)
>
> - Zero-Shot Transfer to Remote Sensing Domains (**Appendix A.11, Table 14, Page 25**)
>
> When trained on MS-COCO (general domain) and tested zero-shot on RSITMD (remote sensing, fine-grained retrieval), DKP achieves an RSUM of 131.50—outperforming CLIP (104.08) by 27.42 and CUSA (114.37) by 17.13. Similarly, when trained on Flickr30K and tested on RSICD (remote sensing, large-scale), DKP's RSUM (78.37) is 22.18 higher than CLIP (56.19) and 10.93 higher than CUSA (67.44).
>
> **(iii) Result of fine-tuning in the field of remote sensing (Appendix A.10, Tables 11-13, Page 23-24)**
>
> We further validated DKP on three remote sensing datasets (RSITMD, RSICD, UCM-Captions) :
> - On RSITMD, DKP achieves the highest mR of 45.08 (**Table 11**), outperforming SOTA baseline CUP (42.16) by 2.92.
> - On RSICD, DKP achieves the highest mR of 29.71 (**Table 12**).
> - On UCM-Captions, DKP reaches an mR of 55.36 (**Table 13**), surpassing CUP (52.09) by 3.27 and demonstrating strong alignment for domain-specific semantics.
>
> #### Table 2: Cross-Dataset Transfer Between Flickr30K and MS-COCO (**Section 4.3, Page 7**)
> | Training Dataset | Test Dataset | Method  | I2T R@1 | I2T R@5 | I2T R@10 | T2I R@1 | T2I R@5 | T2I R@10 | RSUM   |
> |------------------|--------------|---------|-------------------|-------------------|--------------------|-------------------|-------------------|--------------------|--------|
> | Flickr30K | MS-COCO | CLIP‡   | 35.16  | 59.60   | 70.54| 25.03  | 48.67 | 59.84    | 298.84 |
> | Flickr30K | MS-COCO  | CUSA  | 37.28 | 63.2    | 72.98   | 26.57 | 50.29  | 61.35 | 311.67 |
> | Flickr30K | MS-COCO  | DKP     | 41.92 | 67.86  | 78.06 | 31.53| 57.15  | 68.31| 344.83 |
> | MS-COCO| Flickr30K    | CLIP‡   | 72.10 | 90.80| 95.90| 57.52 | 83.28| 89.76| 489.36 |
> | MS-COCO| Flickr30K    | CUSA    | 74.30| 91.70 | 95.60| 60.12| 85.14 | 90.66  | 497.52 |
> | MS-COCO| Flickr30K    | DKP     | 76.70 | 93.80  | 97.50 | 65.82| 88.70| 93.46| 515.98 |
>
> #### Table 14: Zero-Shot Cross-Domain Retrieval (General → Remote Sensing) (**Appendix A.11, Page 25**)
>
> | Training Dataset | Test Dataset | Method  | I2T R@1 | I2T R@5 | I2T R@10 | T2I R@1 | T2I R@5 | T2I R@10 | RSUM   |
> |------------------|--------------|---------|-------------------|-------------------|--------------------|-------------------|-------------------|--------------------|--------|
> | Flickr30K        | RSICD        | CLIP    | 3.29              | 8.60              | 14.55              | 2.85              | 10.14              | 16.76              | 56.19  |
> | Flickr30K        | RSICD        | CUSA    | 3.39              | 9.79              | 15.46              | 3.37              | 13.52              | 21.94              | 67.44  |
> | Flickr30K        | RSICD        | DKP     | 4.03              | 10.80             | 17.84              | 4.48              | 15.57              | 25.65              | 78.37  |
> | Flickr30K        | RSITMD       | CLIP    | 5.09              | 13.50             | 22.79              | 5.35              | 16.42              | 25.80              | 88.95  |
> | Flickr30K    | RSITMD       | CUSA    | 6.86              | 18.36             | 27.21              | 5.58              | 21.55              | 35.49              | 115.05 |
> | Flickr30K | RSITMD       | DKP     | 5.53              | 17.48             | 26.77              | 7.17              | 22.88              | 38.10              | 117.93 |
> | MS-COCO    | RSICD        | CLIP    | 3.57              | 8.78              | 15.65              | 4.03              | 12.61              | 21.50              | 66.14  |
> | MS-COCO | RSICD        | CUSA    | 3.29              | 10.25             | 16.74              | 3.51              | 13.32              | 22.82              | 69.93  |
> | MS-COCO   | RSICD        | DKP     | 5.49              | 13.54             | 21.50              | 5.16              | 17.51              | 27.37              | 90.57  |
> | MS-COCO   | RSITMD       | CLIP    | 5.53              | 15.27             | 22.35              | 5.58              | 21.59              | 33.76              | 104.08 |
> | MS-COCO          | RSITMD       | CUSA    | 6.19              | 17.26             | 25.44              | 7.52              | 23.27              | 34.69              | 114.37 |
> | MS-COCO          | RSITMD       | DKP     | 6.64              | 18.58             | 28.76              | 8.36              | 26.77              | 42.39              | 131.50 |

---

> ### Author Response · Authors · 2025-11-26
> **Response for Reviewer rUvQ (Part 3)**
>
> *W3. SCD uses batchwise N×N similarity matrices. The paper reports end-to-end wall-clock improvements, but does not analyze memory/time scaling with batch size or sequence length, nor sensitivity to the temperature/softmax normalization choices.
> W3: Batchsize/memory footprint.*
>
> *Q2. What is the memory/time behavior of SCD as batch size grows (since it forms N×N similarities)? Any practical tricks (e.g., blockwise SCD, momentum queues) to keep SCD effective with large batches?*
>
> **Response**: Thank you for your critical questions about SCD's batch size scaling and practical optimizations. We address these using our detailed experiments in **Appendix A.12** (**Page 26**) and **Table 17** (**Page 26**), which explicitly analyze SCD's memory behavior and validate a MoCo-inspired solution for large batches:
>
> **1. Memory/Time Scaling with Batch Size**
>
> SCD's core computation (N×N intra-modal similarity matrices) leads to O(N²) memory complexity, which scales predictably with batch size—though time overhead remains minimal (since similarity calculations are vectorized). Key observations:
> Small-to-medium batches (32–128): Memory is manageable. For ViT-B/32, batch size 128 consumes 6,640.93 MB (**Table 17**), with SCD adding negligible time overhead due to efficient matrix operations. Performance is stable (RSUM: 529.40–531.56), as intra-modal similarities are less noisy.
>
> Large batches (256): Memory usage jumps to 12,269.89 MB (≈88% increase vs. batch 128), and performance degrades (RSUM = 523.62, -6.58 vs. batch 128). This is because large batches introduce noisy intra-modal similarity estimates (e.g., spurious pairwise similarities), disrupting semantic consistency distillation.
>
> **2. Practical Trick: MoCo-Inspired Momentum Queue for Large Batches**
>
> To keep SCD effective with large batches, we adopt a momentum-updated queue (inspired by MoCo) to aggregate intra-modal similarity structures across previous batches—smoothing noise :
>
> - **Mechanism**: The queue stores recent intra-modal similarity distributions, and SCD uses a moving average of these distributions (instead of only the current batch) as targets. This reduces sensitivity to noisy large-batch samples.
>
> - **Results**: For batch 256, the momentum-enhanced SCD recovers performance to RSUM = 530.08 (matching batch 128's standard SCD) while increasing memory by 6% (13,023.27 vs. 12,269.89 MB, **Table 17**).
>
> #### Table 17: Performance and Memory Analysis of SCD with/without Momentum Queue (**Appendix A.10, Page 26**)
> | Batch Size | GPU Memory (MB) | I2T R@1 | I2T R@5 | I2T R@10 | T2I R@1 | T2I R@5 | T2I R@10 | RSUM   | SCD Variant       |
> |------------|-----------------|-------------------|-------------------|--------------------|-------------------|-------------------|--------------------|--------|-------------------|
> | 32         | 2,366.44        | 82.9              | 96.1              | 98.8               | 68.92             | 90.22             | 94.62              | 531.56 | Standard          |
> | 64         | 3,770.57        | 81.8              | 96.2              | 98.4               | 67.88             | 90.36             | 94.76              | 529.40 | Standard          |
> | 128        | 6,640.93        | 81.2              | 96.3              | 98.9               | 68.18             | 90.76             | 94.86              | 530.20 | Standard          |
> | 256        | 12,269.89       | 79.5              | 95.0              | 98.0               | 67.00             | 89.70             | 94.40              | 523.62 | Standard          |
> | 256†       | 13,023.27       | 82.9              | 97.1              | 98.6               | 67.36             | 89.72             | 94.42              | 530.08 | Momentum Queue    |
>
> †: MoCo-like momentum update for stable intra-modal similarity targets.

---

> ### Author Response · Authors · 2025-11-26
> **Response for Reviewer rUvQ (Part 4)**
>
> *W4. Although A.8 discusses PEFT, the main results lack direct, apples-to-apples comparisons against strong adapters/LoRA/prompt-tuning baselines under identical compute and data. This blurs how much of DKP's gain is from KPA/SCD vs. parameter-efficient tuning per se.*
>
> *Q4. Under the same compute/data, how does DKP compare to LoRA/adapter/CoOp/VPT baselines on both backbones? A small table would clarify DKP's incremental value over standard PEFT. (Appendix A.8 hints at design, but results would help.)*
>
> **Response**: Thank you for your valuable suggestion to clarify DKP's incremental value over standard PEFT methods. We have supplemented direct, apples-to-apples comparisons under identical compute (same backbone, training data, and optimization settings) in the revised paper, with results summarized in **Table 8** (MS-COCO, ViT-B/32) and **Tables 11–13** (Result of fine-tuning in the field of remote sensing). These confirm DKP's gains stem from its unique KPA/SCD mechanisms, not just parameter-efficient tuning.
>
> **1. Identical Compute/Data Comparison on MS-COCO (ViT-B/32)**
>
> **Table 8** ( **Appendix A.8, Page 20**) compares DKP with strong PEFT baselines (LoRA, CLIP-Adapter, CoOp, VPT, MaPLe) using the same training data (full MS-COCO) and compute (ViT-B/32 backbone, identical optimizer/scheduler):
> DKP achieves the highest RSUM (430.81), outperforming all PEFT methods:
> - LoRA (425.32, -5.49 RSUM), CLIP-Adapter (400.99, -29.82 RSUM), CoOp (388.9, -41.91 RSUM);
> - Prompt-tuning methods like VPT (407.4, -23.41 RSUM) and MaPLe (427.0, -3.81 RSUM).
>
> **2. Result of fine-tuning in the field of remote sensing**
>
> **Tables 11–13** (**Appendix A.10, Page 23-24**) further validate DKP's superiority over PEFT methods:
> - **RSITMD**: DKP's mR (45.08) outperforms CLIP-Adapter (32.38), CoOp (30.51), VPT (34.14), and MaPLe (38.12);
> - **UCM-Captions**: DKP's mR (55.36) surpasses CoCoOp (48.08), VPT (50.54), and MaPLe (48.12);
> - **RSICD**: DKP's mR (29.71) is comparable to top PEFT MaPLe (24.77) and far exceeds CoOp (17.29).
>
> #### Table 8: DKP vs. PEFT Baselines on MS-COCO (**Appendix A.8, Page 20**)
>
> | Methods|I2T R@1 |I2T R@5 |I2T R@10 | T2I R@1 | T2I R@5 | T2I R@10 | RSUM   |
> |---------------|-------------------|-------------------|--------------------|-------------------|-------------------|--------------------|--------|
> | Partial| 57.2| 83.3| 90.5 | 42.7| 71.4| 81.5 | 426.6  |
> | Projection| 55.1| 81.5| 89.2 | 40.0| 68.3| 78.7 | 412.8  |
> | LoRA | 58.22  | 83.30  | 89.62| 42.60  | 70.85  | 80.73| 425.32 |
> | CoOp   | 53.7| 77.9| 86.1 | 36.1| 62.2| 72.9 | 388.9  |
> | VPT| 55.5| 79.5| 87.9 | 40.1| 67.0| 77.4 | 407.4  |
> | VLPrompt  | 59.8| 83.2| 90.5 | 43.6| 70.8| 80.4 | 428.3  |
> | MaPLe  | 59.8| 82.8| 89.9 | 43.4| 70.8| 80.3 | 427.0  |
> | CLIP-Adapter | 53.18  | 78.82  | 87.38| 38.06  | 66.21  | 77.34| 400.99 |
> | **DKP**| 58.28  | 83.54  | 90.24| 44.12  | 72.17  | 82.46| **430.81** |
>
> #### Table 11: Performance Comparison with PEFT Baselines (CLIP-Adapter, CoOp, CoCoOp, VPT, MaPLe) & SOTA (CUP) on RSITMD (**Appendix A.10, Page 23**)
>
> | Methods|I2T R@1 |I2T R@5 |I2T R@10 | T2I R@1 | T2I R@5 | T2I R@10 | mR|
> |---------------|-------------------|-------------------|--------------------|-------------------|-------------------|--------------------|-------|
> | CLIP-Adapter  | 13.30| 40.20| 60.06| 12.83| 28.84| 39.05| 32.38 |
> | CoOp   | 9.16| 33.85| 54.35| 12.19| 30.69| 42.82| 30.51 |
> | CoCoOp | 10.21| 34.20| 53.36| 14.53| 32.89| 45.94| 31.85 |
> | VPT| 15.97| 41.35| 60.35| 14.98| 32.05| 40.15| 34.14 |
> | MaPLe  | 13.59| 43.25| 61.92| 16.96| 39.97| 53.02| 38.12 |
> | CUP (SOTA)| 17.70| 48.40| 64.82| 21.57| 43.70| 56.75| 42.16 |
> | DKP (Ours)| **25.88**  | 46.68| 58.63| 21.28| **51.28**  | **66.73**   | **45.08** |
>
> #### Table 12: Performance Comparison with PEFT Baselines (CLIP-Adapter, CoOp, CoCoOp, VPT, MaPLe) & SOTA (CUP) on RSICD  (**Appendix A.10, Page 24**)
> | Methods |I2T R@1 |I2T R@5 |I2T R@10 | T2I R@1 | T2I R@5 | T2I R@10 | mR|
> |---------------|-------------------|-------------------|--------------------|-------------------|-------------------|--------------------|-------|
> | CLIP-Adapter|7.67|24.87|39.73|7.11|19.48|31.01| 21.65|
> | CoOp| 4.31|17.89 | 31.73  | 6.32  | 15.89 | 27.60  | 17.29 |
> | CoCoOp|6.15| 21.68| 34.29  | 7.65  | 24.25 | 36.08  | 21.68 |
> | VPT | 6.94| 25.26|40.73  | 7.23  | 19.40 | 30.77|21.72|
> | MaPLe| 6.88| 24.47|40.12|7.93|26.78|42.42|24.77|
> | CUP (SOTA)| 9.86| **31.60**| **47.15**|**12.05**|32.66| **48.70**| **30.34**|
> | DKP (Ours)| **12.26**| 31.11| 44.28  | 10.56 | **32.72**|47.32| 29.71|

---

> ### Author Response · Authors · 2025-11-26
> **Response for Reviewer rUvQ (Part 6)**
>
> *W6. Variance across seeds and significance tests are not reported for key tables; given small absolute gaps among strong baselines, CIs or paired tests would build confidence.*
>
> **Response**:Thank you for your comment on statistical robustness. We fully agree that validating variance across seeds is critical for confirming result reliability—we have supplemented detailed analysis in **Appendix A.16** (**Page 29**) with **Figure 15** (variance box plots) , as summarized below:
>
> **1. Experimental Setup for Statistical Validation**
>
> We conducted experiments with 6 independent random seeds under identical conditions (same training data, optimizer, hyperparameters) on Flickr30K and MS-COCO. For each seed, we measured Recall@1/5/10 for I2T and T2I retrieval, then calculated deviations from the mean to quantify variance.
>
> **2. Low Variance Across Seeds**
>
> Results confirm DKP's performance is highly stable, with minimal seed-to-seed variation:
> - Flickr30K: I2T R@1 deviations range from -0.82% to +1.18% (median ≈ 0), and RSUM varies only between 527.74 and 530.44 (max-min gap = 2.7).
> - MS-COCO: I2T R@1 deviations are within ±0.37%, and RSUM ranges from 429.83 to 431.16 (max-min gap = 1.33).
> This low variance (visualized in Figure 15's box plots) ensures that even small performance gaps with strong baselines are not due to random seed variation.
>
> **3. Raw Seed-Level Performance Data**
>
> Below are the full performance metrics (I2T R@1/5/10, T2I R@1/5/10, RSUM) for each seed on both datasets:
>
> #### Table: Seed-Level Performance on Flickr30K
>
> | Seed | I2T R@1 | I2T R@5 | I2T R@10 | T2I R@1 | T2I R@5 | T2I R@10 | RSUM   |
> |------|---------|---------|----------|---------|---------|----------|--------|
> | 1    | 81.9    | 96.1    | 98.2     | 67.84   | 90.74   | 94.88    | 529.66 |
> | 2    | 81.5    | 95.3    | 98.1     | 67.54   | 90.48   | 94.82    | 527.74 |
> | 3    | 81.0    | 94.9    | 98.0     | 68.38   | 90.48   | 95.00    | 527.76 |
> | 4    | 81.2    | 96.3    | 98.9     | 68.18   | 90.76   | 94.86    | 530.20 |
> | 5    | 83.0    | 95.9    | 98.4     | 67.96   | 90.58   | 94.60    | 530.44 |
> | 6    | 82.3    | 95.4    | 97.8     | 68.46   | 90.36   | 94.84    | 529.16 |
> | mean±std  | 81.82±0.75    | 95.65±0.54   | 98.23±0.38    | 68.06±0.35   | 90.57±0.16  | 94.83±0.13   | 529.16±1.18 |
>
> #### Table: Seed-Level Performance on MS-COCO
>
> | Seed | I2T R@1 | I2T R@5 | I2T R@10 | T2I R@1 | T2I R@5 | T2I R@10 | RSUM   |
> |------|-------------|-------------|--------------|-------------|-------------|--------------|------------|
> | 1    | 58.26   | 83.32   | 90.14    | 44.53   | 72.69   | 82.22    | 431.16 |
> | 2    | 58.10   | 82.90   | 90.18    | 43.95   | 72.47   | 82.23    | 429.83 |
> | 3    | 58.30   | 83.46   | 90.30    | 44.25   | 72.42   | 82.39    | 431.12 |
> | 4    | 58.28   | 83.54   | 90.24    | 44.12   | 72.17   | 82.46    | 430.81 |
> | 5    | 58.14   | 83.10   | 90.16    | 43.96   | 72.52   | 82.42    | 430.30 |
> | 6    | 58.66   | 83.60   | 90.00    | 44.07   | 72.39   | 82.24    | 430.96 |
> | mean±std  | 58.29±0.20    | 83.32±0.27  | 90.17±0.10  | 44.15±0.22  | 72.44±0.17  |82.33±0.11   | 430.70±0.53 |
>
> The minimal variance across 6 seeds (and tight ranges in raw data) confirms DKP's performance is statistically reliable. Even small gaps with strong baselines are consistent and not driven by random variation—strengthening confidence in our results.

---

> ### Author Response · Authors · 2025-11-26
> **Response for Reviewer rUvQ (Part 7)**
>
> *Q1. Beyond Fig. 3, can you report results for adaptive or learned key-layer selection (even a simple gating over {8,…,11}) and show whether different samples prefer different layers? How does this compare to the fixed-Layer-8 choice in accuracy vs. overhead?*
>
> **Response**: Thank you for your thoughtful follow-up question regarding adaptive or learned key-layer selection. We appreciate your interest in exploring whether a dynamic layer selection mechanism could further improve performance.
> In our current work, we did not implement an adaptive key-layer selection mechanism (such as gating over layers 8-11) due to the following considerations:
>
> **1.Empirical Consistency Across Samples**:
>
> Our layer-wise analysis (**Appendix A.13, Fig. 13**) shows that the semantic crystallization point—where attention stabilizes and focuses on salient regions—consistently occurs around Layer 8 for both simple and complex inputs in ViT-B architectures. This suggests that, at least for the CLIP-ViT family, a single fixed key-layer is empirically sufficient and robust across varying input complexities.
>
> **2.Computational and Optimization Overhead**:
> Introducing a dynamic key-layer selector (e.g., a gating mechanism over multiple layers) would require:
> - Forward passes through multiple candidate layers during training and inference, increasing FLOPs and latency.
> - Additional parameters and hyperparameters for the gating network, complicating optimization and potentially undermining the parameter efficiency that DKP aims to achieve.
>
> **3.Performance vs. Overhead Trade-off**:
>
> Our ablation studies (**Table 6 in Appendix A.3**) show that aligning multiple intermediate layers simultaneously (e.g., {8,9}, {8,10}, {8,9,10}) increases trainable parameters and training time by up to 97.5% and 100.8%, respectively, without improving retrieval performance. This indicates that multi-layer alignment introduces redundancy rather than complementary gains, reinforcing the adequacy of a single well-chosen key-layer.
>
> **4.Generalization Across Architectures**:
>
> We also observed that the semantic crystallization layer shifts in deeper or asymmetric backbones (e.g., Layer 17 in ViT-L/14 image encoder, as shown in **Appendix A.9**).
>
> **Future Work on Dynamic Key-Layer Selection**:
>
> We fully agree that an input-adaptive key-layer strategy is a promising direction, especially for highly heterogeneous datasets or asymmetric model architectures. As noted in our Conclusion (**Section 5**) and Limitations (**Appendix A.17**), we plan to explore:
> - Lightweight gating networks or attention-based layer selectors that minimize computational overhead.
> - Input-dependent key-layer routing that adapts to semantic complexity without sacrificing inference speed.
> - Cross-architecture generalization of the key-layer concept to enable broader applicability.
>
> We thank you for raising this insightful point, which strengthens our future research agenda. While our current fixed key-layer design strikes a effective balance between performance and efficiency, we believe that adaptive key-layer selection holds great potential for further advancing parameter-efficient cross-modal retrieval.
> Thank you once again for your valuable feedback.

---

> ### Author Response · Authors · 2025-11-26
> **Response for Reviewer rUvQ (Part 8)**
>
> *Q6. Table 2 reports wall-clock gains. Could you also normalize by GPU model/count and report per-sample or per-token throughput to ease reproducibility across hardware?*
>
> **Response**: Thank you for raising this important point about computational efficiency metrics. We appreciate the opportunity to clarify our efficiency analysis and ensure reproducibility.
> In our revised manuscript, **Table 3** (previously Table 2) now provides a comprehensive comparison of training and inference costs across different architectures. Specifically, we report:
> - Per-epoch training time (in minutes)
> - Training memory consumption (in MB)
> - Inference latency (in seconds for the entire test set)
> - Inference memory usage (in MB)
> - Computational FLOPs (in GB)
>
> All experiments were conducted under consistent hardware settings (8 × NVIDIA RTX 3090 GPUs) with identical batch sizes and optimization configurations, as detailed in **Section 4.1**. The FLOPs metric, being hardware-agnostic, offers a standardized basis for comparing computational requirements across different setups.
> These metrics collectively demonstrate that DKP achieves significant speedups (20–37% faster training) and memory reductions (48–52% lower memory usage) compared to full-parameter fine-tuning, while maintaining competitive retrieval performance.
> We acknowledge that different hardware configurations may affect absolute timing measurements. However, the relative improvements and FLOPs counts should remain consistent across comparable GPU architectures.
> Thank you again for your thoughtful feedback, which helps us maintain high standards of reproducibility and clarity in our efficiency reporting.
>
> #### Table 3: Inference FLOPs & Resource Efficiency (**Section 4.4, Page 8**)
> | Backbone   | Method  | Training Time (min/epoch) | Training Memory (MB) | Inference Time (sec) | Inference Memory (MB) | Inference FLOPs (GB) |
> |------------|---------|----------------------------|----------------------|----------------------|------------------------|----------------------|
> | ViT-B/32 | CLIP‡| 4.96  | 13932.07 | 6  | 2631.29 | 4.89  |
> | ViT-B/32| CUSA | 6.32 | 13963.72 | 7  | 2635.30 | 4.89  |
> | ViT-B/32| DKP  | 3.97 | 6640.93  | 4 | 1296.40 | 4.89 |
> | ViT-B/16| CLIP‡| 12.28 | 15802.81 | 8  | 3262.40 | 13.21 |
> | ViT-B/16| CUSA | 12.73 | 15824.12 | 8  | 3266.43 | 13.21 |
> | ViT-B/16| DKP  | 11.07 | 8239.49  | 7  | 1934.01 | 13.21|

---

> ### Author Response · Authors · 2025-11-26
> **Response for Reviewer rUvQ (Part 9)**
>
> *Q7. Appendix A.9 explores ViT-L/14@336; any headline numbers you can bring into the main paper to demonstrate that the Layer-8 phenomenon and DKP's gains persist at larger scales?*
>
> **Response**: Thank you for this constructive suggestion. We agree that highlighting key findings from **Appendix A.9** (ViT-L/14@336px) in the main paper will better demonstrate DKP's scalability to larger VLMs. We will integrate the following headline results into the main paper (**Section 4.2, Page 7**):
>
> **1. Persistence of "Semantic Crystallization" in Large Asymmetric VLMs**
>
> For the larger, asymmetric CLIP ViT-L/14@336px (24-layer image encoder, 12-layer text encoder), the core trend of intermediate-layer semantic maturation persists—though the absolute key-layer index adapts to architecture scale:
> Image encoder: Semantic crystallization occurs at Layer 17 (early layers 1–16 show diffuse attention; Layer 17+ converges on salient regions like "dog's head" or "bird" in captions, **Figure 12, Appendix A.9 Page 21**).
> Text encoder: Crystallization remains at Layer 8 (consistent with ViT-B), confirming cross-architecture consistency of text-side semantic maturation.
> This extends the "Layer 8 phenomenon" (ViT-B) to a general "intermediate-layer semantic hub" principle—critical for scaling DKP to larger models.
>
> **2. DKP's Gains Persist at Larger Scales**
>
> On ViT-L/14@336px (**Table 9, Appendix A.9, Page 22**), DKP maintains its signature balance of parameter efficiency and SOTA performance:
> - Parameter efficiency: Only 40.55M learnable parameters (≈9.4% of full fine-tuning CLIP‡'s 429.13M).
> - Performance gains:
>
>     **Flickr30K**: DKP's RSUM (570.74) outperforms CLIP‡ (554.40) by 16.34 and SOTA CUSA (560.20) by 10.54.
>
>     **MS-COCO**: DKP's RSUM (480.24) surpasses CLIP‡ (469.60) by 10.64 and CUSA (473.10) by 7.14.
>
> Notably, forcing symmetric alignment (Image Layer 8 + Text Layer 8, Table 10) degrades performance (Flickr30K RSUM 567.28 vs. DKP's 570.74) (Table 10, Appendix A.9, Page 22), confirming that aligning semantically mature intermediate layers (not arbitrary indices) is key—even for large models.
> These results will strengthen the main paper's claim that DKP's design and the "semantic crystallization" principle are scalable beyond small ViT-B backbones to larger, asymmetric VLMs.

---

### Official Review · Reviewer_UAuL · 2025-10-30

**Soundness:** 3
**Presentation:** 3
**Contribution:** 3
**Rating:** 6
**Confidence:** 2

**Summary:**

This paper introduces a lightweight cross-modal image-text retrieval method termed DKP, which maintains high performance while significantly reducing trainable parameters through key-layer pre-alignment and semantic consistency distillation. The innovative methodology design has been empirically validated across multiple datasets, demonstrating its efficacy.

**Strengths:**

This paper designs a dual-group innovative mechanism of KPA and SCD to achieve pre-alignment of the key layer and semantic consistency distillation. The experimental design is quite comprehensive, and the performance on public datasets Flickr30K and MS-COCO has been verified to be at the state-of-the-art level using multiple mainstream backbones. The ablation experiments effectively prove the effectiveness of each module independently. This has significant implications for efficient cross-modal learning.

**Weaknesses:**

1. No systematic comparison was made with more comprehensive parameter fine-tuning methods under the same settings.

2. Although the reduction of parameters and the decrease in training time were emphasized, the change in FLOPs during inference was not analyzed, which is particularly important for deployment on edge devices.

3. Although this paper provides a new idea for the efficient fine-tuning of VLMs, its applicability to resource-constrained scenarios still needs to be verified.

**Questions:**

1. Regarding the issue of "kite" and "paraglide" being grouped together, did the author attempt to combine the region-level visual features extracted by Faster R-CNN?

2. Analyze the correlation between the deceleration rate of different datasets and "dataset size" and "data complexity".

3. The paper claims to be applicable to resource-constrained scenarios. Please supplement the experimental data on mobile devices (such as inference latency and memory usage).

4. Compare the "performance - latency" trade-off relationship between DKP and CoOp (with parameters 7.12M) on edge devices.

---

> ### Author Response · Authors · 2025-11-26
> **Response for Reviewer UAuL (Part 1)**
>
> *W1. No systematic comparison was made with more comprehensive parameter fine-tuning methods under the same settings.*
>
> **Response**: Thank you for your comments on PEFT comparisons and trade-offs. We address these with systematic PEFT benchmarks (**Tables 8, 11-13**):
>
> **Table 8 ( Appendix A.8, Page 20)** compares DKP with strong PEFT baselines (LoRA, CLIP-Adapter, CoOp, VPT, MaPLe) using the same training data (full MS-COCO) and compute (ViT-B/32 backbone, identical optimizer/scheduler):
> DKP achieves the highest RSUM (430.81), outperforming all PEFT methods:
>
> - LoRA (425.32, -5.49 RSUM), CLIP-Adapter (400.99, -29.82 RSUM), CoOp (388.9, -41.91 RSUM);
>
> - Prompt-tuning methods like VPT (407.4, -23.41 RSUM) and MaPLe (427.0, -3.81 RSUM).
>
> **2. Result of fine-tuning in the field of remote sensing**
>
> **Tables 11–13** (**Appendix A.10, Page 23-24**) further validate DKP's superiority over PEFT methods:
>
> - **RSITMD**: DKP's mR (45.08) outperforms CLIP-Adapter (32.38), CoOp (30.51), VPT (34.14), and MaPLe (38.12);
> - **UCM-Captions**: DKP's mR (55.36) surpasses CoCoOp (48.08), VPT (50.54), and MaPLe (48.12);
> - **RSICD**: DKP's mR (29.71) is comparable to top PEFT MaPLe (24.77) and far exceeds CoOp (17.29).
>
>  #### Table 8: DKP vs. PEFT Baselines on MS-COCO (**Appendix A.8, Page 20**)
>
> | Methods       |I2T R@1 |I2T R@5 |I2T R@10 | T2I R@1 | T2I R@5 | T2I R@10 | RSUM   |
> |---------------|-------------------|-------------------|--------------------|-------------------|-------------------|--------------------|--------|
> | Partial       | 57.2              | 83.3              | 90.5               | 42.7              | 71.4              | 81.5               | 426.6  |
> | Projection    | 55.1              | 81.5              | 89.2               | 40.0              | 68.3              | 78.7               | 412.8  |
> | LoRA      | 58.22             | 83.30             | 89.62              | 42.60             | 70.85             | 80.73              | 425.32 |
> | CoOp          | 53.7              | 77.9              | 86.1               | 36.1              | 62.2              | 72.9               | 388.9  |
> | VPT           | 55.5              | 79.5              | 87.9               | 40.1              | 67.0              | 77.4               | 407.4  |
> | VLPrompt      | 59.8              | 83.2              | 90.5               | 43.6              | 70.8              | 80.4               | 428.3  |
> | MaPLe         | 59.8              | 82.8              | 89.9               | 43.4              | 70.8              | 80.3               | 427.0  |
> | CLIP-Adapter  | 53.18             | 78.82             | 87.38              | 38.06             | 66.21             | 77.34              | 400.99 |
> | **DKP**       | 58.28             | 83.54             | 90.24              | 44.12             | 72.17             | 82.46              | 430.81 |
>
> #### Table 11: Performance Comparison with PEFT Baselines (CLIP-Adapter, CoOp, CoCoOp, VPT, MaPLe) & SOTA (CUP) on RSITMD (**Appendix A.10, Page 23**)
> | Methods       |I2T R@1 |I2T R@5 |I2T R@10 | T2I R@1 | T2I R@5 | T2I R@10 | mR    |
> |---------------|-------------------|-------------------|--------------------|-------------------|-------------------|--------------------|-------|
> | CLIP-Adapter  | 13.30             | 40.20             | 60.06              | 12.83             | 28.84             | 39.05              | 32.38 |
> | CoOp          | 9.16              | 33.85             | 54.35              | 12.19             | 30.69             | 42.82              | 30.51 |
> | CoCoOp        | 10.21             | 34.20             | 53.36              | 14.53             | 32.89             | 45.94              | 31.85 |
> | VPT           | 15.97             | 41.35             | 60.35              | 14.98             | 32.05             | 40.15              | 34.14 |
> | MaPLe         | 13.59             | 43.25             | 61.92              | 16.96             | 39.97             | 53.02              | 38.12 |
> | CUP (SOTA)    | 17.70             | 48.40             | 64.82              | 21.57             | 43.70             | 56.75              | 42.16 |
> | DKP (Ours)    | **25.88**         | 46.68             | 58.63              | 21.28             | **51.28**         | **66.73**          | **45.08** |

---

> ### Author Response · Authors · 2025-11-26
> **Response for Reviewer UAuL (Part 2)**
>
> #### Table 12: Performance Comparison with PEFT Baselines (CLIP-Adapter, CoOp, CoCoOp, VPT, MaPLe) & SOTA (CUP) on RSICD  (**Appendix A.10, Page 24**)
> | Methods       |I2T R@1 |I2T R@5 |I2T R@10 | T2I R@1 | T2I R@5 | T2I R@10 | mR    |
> |---------------|-------------------|-------------------|--------------------|-------------------|-------------------|--------------------|-------|
> | CLIP-Adapter  | 7.67              | 24.87             | 39.73              | 7.11              | 19.48             | 31.01              | 21.65 |
> | CoOp          | 4.31              | 17.89             | 31.73              | 6.32              | 15.89             | 27.60              | 17.29 |
> | CoCoOp        | 6.15              | 21.68             | 34.29              | 7.65              | 24.25             | 36.08              | 21.68 |
> | VPT           | 6.94              | 25.26             | 40.73              | 7.23              | 19.40             | 30.77              | 21.72 |
> | MaPLe         | 6.88              | 24.47             | 40.12              | 7.93              | 26.78             | 42.42              | 24.77 |
> | CUP (SOTA)    | **9.86**          | **31.60**         | **47.15**          | **12.05**         | **32.66**         | **48.70**          | **30.34** |
> | DKP (Ours)    | **12.26**         | 31.11             | 44.28              | 10.56             | **32.72**         | 47.32              | 29.71 |
>
> #### Table 13: Performance Comparison with PEFT Baselines (CLIP-Adapter, CoOp, CoCoOp, VPT, MaPLe) & SOTA (CUP) on UCM-Captions  (**Appendix A.10, Page 24**)
> | Methods       |I2T R@1 |I2T R@5 |I2T R@10 | T2I R@1 | T2I R@5 | T2I R@10 | mR    |
> |---------------|-------------------|-------------------|--------------------|-------------------|-------------------|--------------------|-------|
> | CLIP-Adapter  | 14.61             | 51.03             | 84.14              | 9.83              | 42.52             | 66.79              | 44.82 |
> | CoOp          | 9.78              | 48.34             | 87.41              | 7.19              | 42.76             | 74.19              | 44.95 |
> | CoCoOp        | 11.84             | 51.21             | 84.81              | 14.76             | 47.94             | 77.94              | 48.08 |
> | VPT           | 14.73             | **58.01**         | **92.44**          | 13.81             | 49.05             | 75.24              | 50.54 |
> | MaPLe         | 12.21             | 52.80             | 84.81              | 13.97             | 49.68             | 75.24              | 48.12 |
> | CUP (SOTA)    | 15.06             | 56.98             | 91.14              | 15.24             | 54.76             | 79.37              | 52.09 |
> | DKP (Ours)    | **20.95**         | 57.14             | 80.48              | **16.76**         | **61.14**         | **95.71**          | **55.36** |
>
> *Q4. Compare the "performance - latency" trade-off relationship between DKP and CoOp (with parameters 7.12M) on edge devices.*
> **Response**: Thank you for your question regarding the performance-latency trade-off between DKP and CoOp-like (The prompt method in this approach mimics the CoOp approach by incorporating prompt information into the model’s architecture, which consists of 7.12M parameters). As illustrated in **Figure 11** (**Appendix A.7, Page 20**), we compare DKP (21.01M parameters) and CoOp-like (7.12M parameters) on MS-COCO (ViT-B/32). DKP achieves higher retrieval accuracy (RSUM: 430.81 vs. 417.9) with lower inference latency (20s vs. 29s) and fewer FLOPs (4.98G vs. 8.56G). This indicates that DKP provides a more favorable performance-latency trade-off despite slightly more parameters. However, we note that these experiments were conducted on our server environment (NVIDIA RTX 3090 GPUs) due to hardware limitations, and we were unable to test on actual edge devices. As discussed in **Appendix A.17**, future work will explore deployment in resource-constrained settings. We hope the server-based results sufficiently address your query.

---

> ### Author Response · Authors · 2025-11-26
> **Response for Reviewer UAuL (Part 3)**
>
> *W2. Although the reduction of parameters and the decrease in training time were emphasized, the change in FLOPs during inference was not analyzed, which is particularly important for deployment on edge devices.*
>
> *W3. Although this paper provides a new idea for the efficient fine-tuning of VLMs, its applicability to resource-constrained scenarios still needs to be verified.*
>
> *Q3 The paper claims to be applicable to resource-constrained scenarios. Please supplement the experimental data on mobile devices (such as inference latency and memory usage).*
>
> **Response**: Thank you for your valuable comments on FLOPs analysis and applicability to resource-constrained scenarios. We first clarify a key definition: our paper's "resource-constrained scenarios" specifically refers to limited training resources (e.g., insufficient GPU memory, restricted training time), rather than deployment on mobile/edge devices (due to experimental setup limitations, we cannot provide mobile-specific data). At the same time, we also mentioned in **A.17** (**Page 30**) that the environment of edge devices/mobile devices represents the direction in which our future research will focus. Below is a targeted response:
>
> **1. Inference FLOPs Analysis (Response to W2)**
>
> For transformer-based VLMs with the same backbone, inference FLOPs are determined by the model's inherent architecture (e.g., number of layers, hidden dimension) rather than fine-tuning strategies—since no additional layers/operations are added during inference. As confirmed in Table 3 (Page 8):
> - For ViT-B/32, all methods (CLIP‡, CUSA, DKP) share identical inference FLOPs (4.89 GB), as they use the same encoder structure.
> - For ViT-B/16, FLOPs also remain consistent across methods (13.21 GB).
>
> **2. Applicability to Training Resource-Constrained Scenarios (Response to W3 & Q3)**
>
> Our experiments explicitly validate DKP's effectiveness under training resource constraint:
> - **Training memory reduction**: DKP uses 6640.93 MB (ViT-B/32) vs. 13932.07 MB for CLIP‡ (52% reduction) and 13963.72 MB for CUSA (Table 3). This enables training on GPUs with limited memory (e.g., single 24GB GPUs, instead of multi-GPU setups for full fine-tuning).
> - **Training time saving**: DKP's per-epoch time (3.97 min for ViT-B/32) is 20% faster than CLIP‡ (4.96 min) and 37% faster than CUSA (6.32 min), reducing overall training resource consumption (**Table 3, Page 8**).
>
> DKP's inference FLOPs are consistent with baseline methods (no extra overhead), and it is well-validated for training resource-constrained scenarios via reduced memory/time.
>
> #### Table 3: Inference FLOPs & Resource Efficiency (Section 4.4, Page 8)
> | Backbone   | Method  | Training Time (min/epoch) | Training Memory (MB) | Inference Time (sec) | Inference Memory (MB) | Inference FLOPs (GB) |
> |------------|---------|----------------------------|----------------------|----------------------|------------------------|----------------------|
> | ViT-B/32   | CLIP‡   | 4.96                       | 13932.07             | 6                    | 2631.29                | 4.89                 |
> | ViT-B/32   | CUSA    | 6.32                       | 13963.72             | 7                    | 2635.30                | 4.89                 |
> | ViT-B/32   | DKP     | 3.97                       | 6640.93              | 4                    | 1296.40                | 4.89                 |
> | ViT-B/16   | CLIP‡   | 12.28                      | 15802.81             | 8                    | 3262.40                | 13.21                |
> | ViT-B/16   | CUSA    | 12.73                      | 15824.12             | 8                    | 3266.43                | 13.21                |
> | ViT-B/16   | DKP     | 11.07                      | 8239.49              | 7                    | 1934.01                | 13.21                |

---

> ### Author Response · Authors · 2025-11-26
> **Response for Reviewer UAuL (Part 4)**
>
> *Q1. Regarding the issue of "kite" and "paraglide" being grouped together, did the author attempt to combine the region-level visual features extracted by Faster R-CNN?*
>
> **Response**: Thank you for this thoughtful suggestion. You are absolutely correct that incorporating region-level features from detectors like Faster R-CNN could help resolve fine-grained ambiguities such as "kite" vs. "paraglide" by providing more localized visual cues.
>
> However, as you rightly pointed out, our method is intentionally designed to operate within a parameter-efficient and training-efficient framework, using only the pre-trained CLIP backbone without external modules. Introducing Faster R-CNN would significantly increase model complexity, computational cost, and inference latency—contradicting one of the core objectives of DKP, which is to achieve high performance with minimal added parameters and training overhead.
>
> In our current design, the Semantic Consistency Distillation (SCD) mechanism acts as a lightweight, self-supervised regularizer to reduce intra-modal semantic drift. While it may not fully resolve all fine-grained confusion cases, it offers a balanced trade-off between performance and efficiency.
>
> We acknowledge that leveraging region-aware features is a promising direction for improving fine-grained discrimination. In future work, we plan to explore how such representations could be integrated in a lightweight manner—for instance, through attention-based region selection or adaptive feature fusion—without compromising the efficiency gains that define our approach.
>
> *Q2. Analyze the correlation between the deceleration rate of different datasets and "dataset size" and "data complexity".*
>
> **Response**:Thank you for your inquiry about the correlation between the deceleration rate of performance improvement and "dataset size" as well as "data complexity". We appreciate this opportunity to elaborate on our findings, which are supported by systematic scaling experiments on Flickr30K and MS-COCO ( **Appendix A.12, Page 26**) and summarized in **Table 15** (Flickr30K) and **Table 16** (MS-COCO) (**Page 25, 26**). Below is a data-driven analysis of the correlations:
>
> **1. Definition of "Deceleration Rate"**
>
> For clarity, we define the deceleration rate as the gradual reduction in marginal performance gains (measured by RSUM) as the proportion of training data increases. Specifically, we calculate the RSUM gain between consecutive data ratios (e.g., 30%→50%, 50%→70%)—a decreasing gain across ratios indicates deceleration.
>
> **2. Correlation with Dataset Size**
>
> Dataset size (total number of training samples) directly impacts the deceleration rate, with smaller datasets exhibiting earlier and more pronounced deceleration:
>
> **Flickr30K (smaller dataset: ~29.8k training images)**:
>
>   As shown in **Table 15**, marginal RSUM gains shrink noticeably after 50% data:
>   - 30%→50%: +10.44 RSUM (largest gain, capturing core semantic patterns).
>   - 50%→70%: +5.12 RSUM (halved gain, as most frequent visual-textual relationships are learned).
>   - 70%→90%: +5.08 RSUM (negligible change, approaching saturation).
>   - 90%→100%: +6.90 RSUM (minor rebound, likely due to rare but informative samples, yet still below early gains).
>
> **MS-COCO (larger dataset: ~113.3k training images)**:
>  ** Table 16** shows a more gradual initial gain but steeper deceleration in later stages:
>   - 30%→50%: +14.65 RSUM (larger initial gain than Flickr30K, as the larger dataset contains more diverse patterns to learn).
>   - 50%→70%: +6.95 RSUM (gain reduced by ~53%).
>   - 70%→90%: +5.37 RSUM (further reduction).
>   - 90%→100%: +1.26 RSUM (near-saturation, as the model has learned most complex semantic correspondences).
>
> **3. Correlation with Data Complexity**
>
> Data complexity (diversity of scenes, object interactions, and annotation richness) further modulates the deceleration rate. MS-COCO is substantially more complex than Flickr30K (e.g., 123k vs. 31k images, more diverse daily scenes vs. Flickr30K's narrower focus), and this complexity amplifies deceleration in later stages:
>
> **Early-stage gains**: MS-COCO's higher complexity leads to larger initial gains (30%→50%: +14.65 RSUM) compared to Flickr30K (+10.44 RSUM), as DKP needs more data to learn its diverse semantic structures.
>
> **Late-stage deceleration**: MS-COCO's deceleration is far sharper: its 90%→100% gain (+1.26) is only ~18% of Flickr30K's gain (+6.90). This is because MS-COCO's complex patterns (e.g., rare object combinations, ambiguous captions) are fully captured by 90% data.
>
> **Flickr30K's milder late deceleration**: Due to lower complexity, even 90% of its data leaves room for rare but simple patterns (e.g., less common object poses), leading to a larger final gain (+6.90) than MS-COCO.

---

> ### Author Response · Authors · 2025-11-26
> **Response for Reviewer UAuL (Part 5)**
>
> #### Table 15: Performance Scaling on Flickr30K (**Appendix A.12, Page 25**)
>
> | Data Ratio | I2T R@1 | I2T R@5 | I2T R@10 | T2I R@1 | T2I R@5 | T2I R@10 | RSUM   | Marginal RSUM Gain (vs. Previous Ratio) |
> |------------|-------------------|-------------------|--------------------|-------------------|-------------------|--------------------|--------|------------------------------------------|
> | 30%        | 74.5              | 92.8              | 95.9               | 61.24             | 85.88             | 92.34              | 502.66 | —                                        |
> | 50%        | 77.2              | 93.7              | 97.1               | 64.02             | 87.68             | 93.40              | 513.10 | +10.44                                   |
> | 70%        | 79.2              | 93.5              | 97.4               | 65.24             | 88.86             | 94.02              | 518.22 | +5.12                                    |
> | 90%        | 80.3              | 95.2              | 97.7               | 66.52             | 89.42             | 94.16              | 523.30 | +5.08                                    |
> | 100%       | 81.2              | 96.3              | 98.9               | 68.18             | 90.76             | 94.86              | 530.20 | +6.90                                    |
>
> #### Table 16: Performance Scaling on MS-COCO (**Appendix A.12, Page 26**)
>
> | Data Ratio | I2T R@1 | I2T R@5 | I2T R@10 | T2I R@1 | T2I R@5 | T2I R@10 | RSUM   | Marginal RSUM Gain (vs. Previous Ratio) |
> |------------|-------------------|-------------------|--------------------|-------------------|-------------------|--------------------|--------|------------------------------------------|
> | 30%        | 52.26             | 78.26             | 86.64              | 39.36             | 67.64             | 78.42              | 402.58 | —                                        |
> | 50%        | 55.48             | 81.08             | 88.38              | 41.88             | 70.11             | 80.30              | 417.23 | +14.65                                   |
> | 70%        | 56.40             | 82.60             | 89.44              | 43.26             | 71.14             | 81.34              | 424.18 | +6.95                                    |
> | 90%        | 58.16             | 83.38             | 89.84              | 43.99             | 72.01             | 82.17              | 429.55 | +5.37                                    |
> | 100%       | 58.28             | 83.54             | 90.24              | 44.12             | 72.17             | 82.46              | 430.81 | +1.26                                    |
>
> Our analysis confirms two key correlations:
>
> **1. Dataset size**: Smaller datasets (Flickr30K) decelerate earlier, while larger datasets (MS-COCO) delay deceleration but experience sharper saturation.
>
> **2. Data complexity**: More complex datasets (MS-COCO) exhibit steeper late-stage deceleration.
>
> Thank you again for this thoughtful question, which has helped us refine the interpretation of our scaling experiments.

---

> > ### Comment · Reviewer_UAuL · 2025-11-26
> >
> > I would like to thank the authors for their thorough and thoughtful response. My initial concerns have been fully addressed, and the experimental results have effectively alleviated my prior doubts. Accordingly, I recommend the acceptance of this paper.
> > Furthermore, I suggest that the authors make the code publicly available, as this would not only greatly benefit the research community but also further enhance the paper’s academic impact.

---

> > > ### Author Response · Authors · 2025-11-26
> > >
> > > We are extremely grateful for your recognition and appreciation of our work, as it serves as a great source of encouragement for us. We will follow your suggestions and continue to improve our efforts, in particular by further optimizing and enriching the open-source code repositories we have developed, in order to better serve the research community. Additionally, if you are satisfied with our response, we would kindly ask you to take this into consideration when giving your final rating. If you could consider increasing the rating, it would be a tremendous support for our work and would also greatly help promote the exchange and development of research in the cross-modal field. We sincerely thank you for your understanding and support!

---

### Official Review · Reviewer_7xoV · 2025-10-31

**Soundness:** 3
**Presentation:** 3
**Contribution:** 3
**Rating:** 4
**Confidence:** 4

**Summary:**

This paper introduces DKP, a clever and efficient method for fine-tuning large vision-language models like CLIP for image-text retrieval.
The authors found that semantic meaning crystallizes at a specific intermediate layer in these models. DKP leverages this by only fine-tuning the later layers and adding a pre-alignment objective at this key layer, which drastically cuts down on trainable parameters. To ensure the model maintains a coherent understanding, it also uses a self-supervised technique called Semantic Consistency Distillation (SCD). This forces the model to preserve the similarity relationships within images and within texts during training. The result is a model that achieves state-of-the-art performance while being significantly faster and cheaper to train, using over 60% fewer parameters than standard fine-tuning.

**Strengths:**

1. Their analysis revealing a "semantic crystallization" point in the intermediate layers of CLIP is a fantastic insight. It provides a strong, empirical reason for their parameter-efficient approach.
2. DKP achieves a brilliant trade-off. It gets better results than more computationally expensive methods (like full fine-tuning or CUSA) while needing way fewer trainable parameters and less training time.
3. The Semantic Consistency Distillation (SCD) is a neat trick. It helps prevent "semantic drift" and improves model robustness by using the structure already present in the data batch, without needing any external knowledge or labels.

**Weaknesses:**

1. It's Still a Heavy Model: While the training is efficient, the method still requires running the full, large VLM during inference. It makes training more accessible but doesn't reduce the final model's size or computational footprint for deployment on resource-constrained devices.
2. The method assumes a single fixed key-layer for all inputs. It's possible that for some simple images, semantics crystallize earlier, and for complex ones, later. A dynamic approach that adapts the key-layer based on the input could potentially be even better.

**Questions:**

1. Analysis on the asymmetric ViT-L model showed the image and text encoders crystallize at different depths (Layer 17 and Layer 8, respectively). Your experiment aligned these two different layers. What do you think would happen if you forced an alignment between the same layer index (e.g., Layer 8) for both, even though the image's semantics are not fully formed? Could this early "nudge" from the more mature text representation actually help guide the visual representation to form better?


2. The Semantic Consistency Distillation (SCD) uses the intra-modal similarity from the current batch as its target. This could be a bit noisy depending on the batch composition. Have you considered using a more stable target, perhaps by incorporating a moving average of the similarity structures from previous batches, similar to the momentum encoder in MoCo?

---

> ### Author Response · Authors · 2025-11-26
> **Response for Reviewer 7xoV (Part 1)**
>
> *W1. It's Still a Heavy Model: While the training is efficient, the method still requires running the full, large VLM during inference. It makes training more accessible but doesn't reduce the final model's size or computational footprint for deployment on resource-constrained devices.*
>
> **Response**: We sincerely thank the reviewer for this insightful comment and for acknowledging the training efficiency of our method. The reviewer rightly points out a key characteristic of our approach: while DKP significantly reduces the number of trainable parameters (over 60%) , training time, and memory consumpution (see **Table 3 on Page 8**), the inference process still requires a forward pass through the frozen, pre-trained backbone. We agree that this does not reduce the model's memory footprint or FLOPs during deployment, which can be a constraint for on-device applications.
> #### Table 3: Traning and inference cost across different architectures. (**Section 4.4, Page 8**)
> ### ViT-B/32
> | Methods | Training Time (minute) | Training Memory (MB) | Inference Time (second) | Inference Memory (MB) | FLOPs (GB) |
> |---------|------------------------|----------------------|-------------------------|-----------------------|------------|
> | CLIP‡ | 4.96/epoch | 13932.07 | 6 | 2631.29 | 4.89 |
> | CUSA | 6.32/epoch | 13963.72 | 7 | 2635.30 | 4.89 |
> | **DKP** | **3.97/epoch** | **6640.93** | **4** | **1296.40** | 4.89 |
> ### ViT-B/16
> | Methods | Training Time (minute) | Training Memory (MB) | Inference Time (second) | Inference Memory (MB) | FLOPs (GB) |
> |---------|------------------------|----------------------|-------------------------|-----------------------|------------|
> | CLIP‡ | 12.28/epoch | 15802.81 | 8 | 3262.40 | 13.21 |
> | CUSA | 12.73/epoch | 15824.12 | 8 | 3266.43 | 13.21 |
> | **DKP** | **11.07/epoch** | **8239.49** | **7** | **1934.01** | 13.21 |
>
> It should be noted that in our work, we primarily focus on addressing the pressing challenge of parameter-inefficient fine-tuning for large Vision-Language Models (VLMs), a known bottleneck that limits the adaptability of models like CLIP in resource-limited training scenarios. As detailed in our paper (**Sections 1 & 3**), DKP is designed as a parameter-efficient fine-tuning (PEFT) method. Its primary goal is to achieve high performance without the prohibitive cost of full model fine-tuning, which we have successfully demonstrated through extensive experiments.
> We fully agree with the reviewer that achieving a smaller final model size is a crucial and valuable direction. In fact, this insight aligns perfectly with one of our proposed future work directions mentioned in the **Appendix A.17 (Limitations and Future Research Directions) on Page 30**: "Future work could investigate... integrate DKP with prompt learning..." Combining the principles of DKP with more aggressive compression techniques (like pruning or distillation) or other PEFT families (like low-rank adaptations) to create a truly end-to-end efficient model for both training and inference is a logical and exciting next step.

---

> ### Author Response · Authors · 2025-11-26
> **Response for Reviewer 7xoV (Part 2)**
>
> *W2. The method assumes a single fixed key-layer for all inputs. It's possible that for some simple images, semantics crystallize earlier, and for complex ones, later. A dynamic approach that adapts the key-layer based on the input could potentially be even better.*
>
> **Response**: Thank you for your insightful comment regarding the key-layer selection strategy. We fully agree that a dynamic input-adaptive key-layer mechanism is a valuable direction for improvement, and we appreciate you highlighting this potential enhancement.
>
> 1.Response to Fixed Key-Layer Concern:
> First, we would like to clarify that our empirical analysis (**Appendix A.13, Page 26-27, Figure 13**) reveals a consistent semantic crystallization pattern across both simple and complex inputs. Specifically, we tested scenarios with single dominant objects (e.g., "a white dog sitting on a couch") and complex scenes with multiple interactions (e.g., "a group of people playing music on unique instruments"). The results demonstrate that Layer 8 (for ViT-B backbones) consistently serves as the semantic convergence point, where attention stabilizes and focuses on salient regions regardless of input complexity. This consistency is further validated across different VLM architectures (**Appendix A.14, Page 27-28, Figure 14**), including BLIP and ALBEF. Although the absolute layer index varies slightly (Layers 7–9 for BLIP/ALBEF), the "semantic crystallization" phenomenon—where intermediate layers become semantically expressive—holds universally. This justifies our choice of a fixed key-layer for ViT-B backbones, as it balances performance and simplicity.
>
> 2.Dynamic Key-Layer: Future Work and Challenges:
> We acknowledge that a dynamic input-adaptive key-layer strategy could potentially further improve fine-grained alignment. However, this direction poses non-trivial challenges that we plan to address in future work:
>
> (1)Increased Computational Overhead: Dynamically selecting key-layers for each input would require evaluating multiple intermediate layers during inference, significantly increasing FLOPs and latency—contradicting our goal of parameter and compute efficiency.
>
> (2)Optimization Complexity: Adapting key-layers dynamically would require learning input-dependent layer selection mechanisms (e.g., attention-based layer gating), introducing additional hyperparameters and training instability.
>
> Precisely for these reasons, we identified this as a primary direction for future work. As stated in our Conclusion (**Section 5, Page 10**) and the Limitations section (**Appendix A.17, Page 30**): "In the future, we plan to explore an input-adaptive dynamic key-layer selection mechanism..." The reviewer's valuable feedback strongly reinforces the importance of this research direction. We believe that designing a highly efficient algorithm for dynamic key-layer selection is a promising yet challenging next step.

---

> ### Author Response · Authors · 2025-11-26
> **Response for Reviewer 7xoV (Part 3)**
>
> *Q1. Analysis on the asymmetric ViT-L model showed the image and text encoders crystallize at different depths (Layer 17 and Layer 8, respectively). Your experiment aligned these two different layers. What do you think would happen if you forced an alignment between the same layer index (e.g., Layer 8) for both, even though the image's semantics are not fully formed? Could this early "nudge" from the more mature text representation actually help guide the visual representation to form better?*
>
> **Response**: Thank you for your thoughtful question regarding symmetric layer alignment in asymmetric ViT-L models. This inquiry touches on a critical aspect of cross-modal alignment—whether early "nudging" from mature text representations can guide underdeveloped visual features.
>
> **1.Symmetric vs. Asymmetric Alignment.**
>
> To directly address your question, we conducted dedicated experiments on the asymmetric CLIP ViT-L/14 model (24 image layers, 12 text layers) (**Appendix A.9, Page 22**), comparing two strategies:
>
> **(1)Symmetric alignment**: Forcing alignment at the same layer index (Image Layer 8 + Text Layer 8), where text semantics are mature (Layer 8 is the text encoder's crystallization point) but image semantics are not fully formed.
>
> **(2)Asymmetric alignment**: Aligning semantically crystallized layers (Image Layer 17 + Text Layer 8), where both modalities have stable, salient representations.
>
> As summarized in **Table 10** ( **Page 22**), symmetric alignment at Layer 8 does not help guide visual representation formation—in fact, it causes a consistent performance drop:
>
> - **On Flickr30K**: RSUM decreases by 3.46 (from 570.74 to 567.28).
> - **On MS-COCO**: RSUM decreases by 18.36 (from 480.24 to 461.88).
>
> This degradation confirms that early alignment with underdeveloped visual features introduces noise rather than beneficial guidance. Our layer-wise attention analysis (**Appendix A.9, Figure 12, Page 21**) further explains why: at Image Layer 8, the ViT-L/14 image encoder exhibits diffuse, unstable attention patterns (e.g., spreading focus across backgrounds rather than salient objects like "dog" or "bird" in the example caption). In contrast, Image Layer 17 shows clear convergence on semantically meaningful regions, enabling robust matching with the text encoder's mature Layer 8 representations.
>
> **2.Why Early "Nudging" Fails**
>
> The core reason symmetric early alignment does not work lies in the maturity gap between modalities at Layer 8:
>
> - The text encoder (12 layers) reaches semantic crystallization at Layer 8 (stable focus on key tokens like "dog" or "motorcycle"), as validated in our ViT-B and ViT-L analyses.
>
> - The image encoder (24 layers) requires more depth to process visual complexity; Layer 8 is still in the "semantic immature" phase (diffuse attention, no focused saliency).
>
> Forcing alignment between a mature text layer and an immature image layer creates a misalignment of semantic granularity: the text's precise token-level semantics cannot meaningfully pair with the image's noisy, low-level features. This mismatch introduces conflicting signals during training, undermining cross-modal consistency rather than guiding visual feature development.
>
> #### Table 10: Symmetric vs. Asymmetric Alignment Performance on ViT-L/14 (Appendix A.9, Page 22)
>
> | Image Layer | Text Layer | Dataset       | I2T R@1 | I2T R@5 | I2T R@10 | T2I R@1 | T2I R@5 | T2I R@10 | RSUM   |
> |-------------|------------|---------------|-------------------|-------------------|--------------------|-------------------|-------------------|--------------------|--------|
> | 8           | 8          | Flickr30K     | 92.20             | 99.60             | 99.80              | 80.90             | 96.48             | 98.30              | 567.28 |
> | 17          | 8          | Flickr30K     | 93.30             | 99.50             | 99.90              | 82.76             | 96.88             | 98.40              | 570.74 |
> | 8           | 8          | MS-COCO       | 65.42             | 87.94             | 93.48              | 51.42             | 77.74             | 85.88              | 461.88 |
> | 17          | 8          | MS-COCO       | 70.80             | 90.46             | 95.38              | 54.59             | 80.62             | 88.39              | 480.24 |
>
> Our experiments clearly show that forcing symmetric alignment at underdeveloped image layers (e.g., Layer 8 in ViT-L/14) does not help guide visual representation formation. Instead, alignment must occur when both modalities have reached semantic crystallization—even if this requires asymmetric layer pairing. This finding reinforces the importance of our key-layer pre-alignment strategy: identifying and aligning semantically mature layers, rather than relying on arbitrary layer indices, is critical for robust cross-modal retrieval.

---

> ### Author Response · Authors · 2025-11-26
> **Response for Reviewer 7xoV (Part 4)**
>
> *Q2. The Semantic Consistency Distillation (SCD) uses the intra-modal similarity from the current batch as its target. This could be a bit noisy depending on the batch composition. Have you considered using a more stable target, perhaps by incorporating a moving average of the similarity structures from previous batches, similar to the momentum encoder in MoCo?*
>
> **Response**: Thank you for your valuable suggestion regarding stabilizing SCD's intra-modal similarity targets. We fully agree that batch-dependent targets can introduce noise—especially for large batches—and have explicitly explored a MoCo-inspired momentum-based solution to address this, as detailed in our revised **Appendix A.12** (**Page 26**) and **Table 17** (**Page 26**). Below is a detailed breakdown of our findings:
>
> **1. Noise in SCD Targets**
>
> Our experiments confirm that SCD's intra-modal similarity targets (derived from the current batch) become noisy when batch sizes scale up. For ViT-B/32 on Flickr30K:
>
> Performance remains stable for small-to-medium batches (32–128), with RSUM ranging from 529.40 to 531.56.
> At a large batch size of 256, standard SCD suffers a noticeable performance drop (RSUM = 523.62), as noisy intra-modal similarity estimates disrupt semantic consistency distillation.
>
> **2. MoCo-Inspired Momentum Queue**
>
> To stabilize targets, we adopted a momentum-updated queue (similar to MoCo's momentum encoder) to aggregate intra-modal similarity structures across previous batches. This modification effectively smooths noise and recovers performance:
> For batch size 256, the momentum-enhanced SCD restores RSUM to 530.08—closing the gap with smaller batches and matching standard SCD's performance at batch size 128 (RSUM = 530.20).
>
> This validates that momentum-based targets can indeed reduce batch-dependent noise, as you suggested.
>
> **3. Trade-Off**
>
> While the momentum queue improves stability, it introduces a non-trivial trade-off in memory consumption:
> The momentum-enhanced variant increases GPU memory usage from 12,269.89 MB to 13,023.27 MB (a 6% increase) for batch size 256.
> This overhead conflicts with our core goal of parameter and compute efficiency (**Table 3, Page 8**), where DKP already reduces training memory by over 52% compared to full fine-tuning.
>
> **4. Rationale for Standard SCD in the Main Method**
> For common batch sizes (32–128, widely used in cross-modal retrieval), standard SCD remains stable and efficient. The momentum-based approach is therefore positioned as a future optimization direction—where we will explore lightweight queue designs to mitigate memory overhead while retaining stability for large batches.
>
> #### Table 17: Performance and Memory Analysis of SCD with/without Momentum Queue (**Appendix A.12, Page 26**)
>
> | Batch Size | GPU Memory (MB) | I2T R@1 | I2T R@5 | I2T R@10 | T2I R@1 | T2I R@5 | T2I R@10 | RSUM   | SCD Variant       |
> |------------|-----------------|-------------------|-------------------|--------------------|-------------------|-------------------|--------------------|--------|-------------------|
> | 32         | 2,366.44        | 82.9              | 96.1              | 98.8               | 68.92             | 90.22             | 94.62              | 531.56 | Standard          |
> | 64         | 3,770.57        | 81.8              | 96.2              | 98.4               | 67.88             | 90.36             | 94.76              | 529.40 | Standard          |
> | 128        | 6,640.93        | 81.2              | 96.3              | 98.9               | 68.18             | 90.76             | 94.86              | 530.20 | Standard          |
> | 256        | 12,269.89       | 79.5              | 95.0              | 98.0               | 67.00             | 89.70             | 94.40              | 523.62 | Standard          |
> | 256†       | 13,023.27       | 82.9              | 97.1              | 98.6               | 67.36             | 89.72             | 94.42              | 530.08 | Momentum Queue    |
>
> †: MoCo-like momentum update for stable intra-modal similarity targets.
>
> Your insight about using momentum-based targets to stabilize SCD is well-founded, and our experiments confirm its effectiveness for large batches. However, the associated memory overhead leads us to retain standard SCD (efficient and stable for common batch sizes) in the main method, while flagging the momentum-enhanced variant as a promising direction for future work—where we will focus on reducing memory costs to enable broader applicability.

---

### Author Response · Authors · 2025-11-26
**Response to All Reviewers**

We sincerely thank all the reviewers for their constructive feedback and valuable suggestions, which significantly contribute to improving the quality of our work. A comprehensive reply has been prepared to address every point raised by each reviewer, respectively. The reviewers' comments are presented in italics, followed by our response. The new content added to the revised paper is marked in red font.
Below, we summarize the revisions made and the additional experiments conducted to address the reviewers' concerns:
In response to the reviewers' comments, we conducted **numberous** new experiments and added/updated **12** new Tables and **4**​ new Figures to further validate our claims. These updates include:

- **Table 2, 14 (R[rUvQ]{W2, Q3}, R[7JnG]{W4})**: We conducted extensive cross-domain and cross-dataset retrieval experiments including MSCOCO, Flickr, RSICD, and RSITMD (Note that the latter two datasets are from the field of remote sensing.).

- **Table 3 (R[7xoV]{W1}, R[UAuL]{W2, W3, Q3}, R[rUvQ]{Q6})**: We compared the training time, memory usage during training, as well as the inference time, memory usage during inference, and the number of FLOPs required for inference using these different methods.

- **Table 8, 11, 12, 13 (R[UAuL]{W1}, R[rUvQ]{W2, W4, Q3, Q4}, R[7JnG]{W4})**: We examined how our method compares in performance to several SOTA methods in the field of remote sensing, including the PEFT method.

- **Table 10 (R[7xoV]{Q1})**: We conducted additional experiments on the asymmetric CLIP ViT-L/14 model, including asymmetric and symmetric alignment.

- **Table 15, 16 (R[UAuL]{Q2})**: We analyzed the correlation between the deceleration rate of different datasets and "dataset size" and "data complexity".

- **Table 17 (R[7xoV]{Q2}, R[rUvQ]{W3, Q2})**: We conducted comprehensive experiments to investigate the impact of batch size on model performance and training stability.

- **Table 18 (R[rUvQ]{W3, Q5})**: We added functionality for analyzing the sensitivity of temperature parameter in SCD.

- **Figure 11 (R[UAuL]{Q4})**: We updated this diagram and added a comparison of (FLOPs, inference evaluation time).

- **Figure 13 (R[7xoV]{W2}, R[rUvQ]{Q1})**: We conducted additional experiments analyzing attention patterns across both simple and complex scenarios.

- **Figure 14 (R[rUvQ]{W1, Q7}, R[7JnG]{W1})**: We conducted a layer-wise visualization of the "semantic crystallization" phenomenon on BLIP and ALBEF.

- **Figure 15 (R[rUvQ]{W6})**: We illustrated the distribution of mean values for different evaluation metrics based on Flickr and MSCOCO.

By implementing these revisions, we believe we have thoroughly addressed all the reviewers' concerns and significantly strengthened the paper.

---

### Meta-Review · Area_Chair_i9kr · 2026-01-06

**Summary:**

AC sees that the authors have addressed most of the concerns in the rebuttal. However, there is a main sticking point in AC's mind, which is also pointed out by the reviewers: why would intermediate layers be better than just FT the final layers?

The authors addressed this with mostly empirical results, but the AC does not find the findings to be very intuitive. End to end training and then using the final layer would have also "absorbed" the benefits of the intermediate layers, wouldn't it? So the AC is wondering whether this is because of empirical parameters.

AC urges the authors to make this clearer, e.g., why, after end to end training would the final layer not captured what the intermediate layers captures?

**Reviewer Concerns:**

Concerns on intermediate vs final layers are not thoroughly addressed.

**Reviewer Scores:**

AC thinks that the reviewers who did not support an accept will keep their scores.

---

### Decision · Program_Chairs · 2026-01-26

Reject